# ATF4 selectively regulates heat nociception and contributes to kinesin-mediated TRPM3 trafficking

Man-Xiu Xie[1,6], Xian-Ying Cao[2,3,6], Wei-An Zeng[1,6], Ren-Chun Lai[1,6], Lan Guo[2], Jun-Chao Wang[1], Yi-Bin Xiao[4], Xi Zhang[4], Di Chen[2], Xian-Guo Liu [4✉] & Xiao-Long Zhang [5✉]

Effective treatments for patients suffering from heat hypersensitivity are lacking, mostly due to our limited understanding of the pathogenic mechanisms underlying this disorder. In the nervous system, activating transcription factor 4 (ATF4) is involved in the regulation of synaptic plasticity and memory formation. Here, we show that ATF4 plays an important role in heat nociception. Indeed, loss of ATF4 in mouse dorsal root ganglion (DRG) neurons selectively impairs heat sensitivity. Mechanistically, we show that ATF4 interacts with transient receptor potential cation channel subfamily M member-3 (TRPM3) and mediates the membrane trafficking of TRPM3 in DRG neurons in response to heat. Loss of ATF4 also significantly decreases the current and KIF17-mediated trafficking of TRPM3, suggesting that the KIF17/ATF4/TRPM3 complex is required for the neuronal response to heat stimuli. Our findings unveil the non-transcriptional role of ATF4 in the response to heat stimuli in DRG neurons.

[1] Department of Anesthesiology, Sun Yat-sen University Cancer Center, State Key Laboratory of Oncology in South China, Collaborative Innovation Center for Cancer Medicine, 651 Dongfeng East Road, Guangzhou, China. [2] College of Food Science and Technology, Hainan University, 58 Renmin Avenue, Haikou, China. [3] State Key Laboratory of Marine Resources Utilization of South China Sea, 58 Renmin Avenue, Haikou, China. [4] Pain Research Center and Department of Physiology, Zhongshan School of Medicine of Sun Yat-sen University, 74 Zhongshan Road 2, Guangzhou, China. [5] Medical Research Center of Guangdong Provincial People's Hospital, Guangdong Academy of Medical Sciences, 106 Zhongshan Rd. 2, Guangzhou, China. [6]These authors contributed equally: Man-Xiu Xie, Xian-Ying Cao, Wei-An Zeng and Ren-Chun Lai. ✉email: liuxg@mail.sysu.edu.cn; zhangxiaolong@gdph.org.cn

Activating transcription factor 4 (ATF4) is a member of the ATF/cAMP response element binding protein (CREB) family[1]. ATF4 regulates transcription by forming dimers with partners via its basic leucine zipper (bZIP) domain[2]. ATF4 increases the presentation of a related transcription factor, activating transcription factor 3 (ATF3), which, along with ATF4, is conducive to stress adaptation by mediating the change of expression levels of genes involved in metabolism, cellular redox status, and cell apoptosis[3,4]. ATF4 plays roles in a variety of tissues. ATF4 knockout mice exhibit severe impairment of skeletal and lens development[5,6]. In the nervous system, ATF4 plays a critical role in synaptic plasticity and memory formation[7]. However, whether and how ATF4 participates in nociception is unclear.

Distinctive membrane proteins in neurons are sensitive to particular stimuli that mediate the transduction of nociceptive signals. TRP channels form various cation channel clusters involved in diverse physiological processes, including stimulus detection in sensory neurons, transcellular cation transport and electrogenesis[8]. Among these proteins, transient receptor potential cation channel subfamily M member-3 (TRPM3), a non-selective cation channel, can be activated by heat and membrane depolarisation[9,10]. TRPM3 is expressed at high levels in nociceptive dorsal root ganglion (DRG) neurons and has recently been shown to contribute to painful reactions to heat[10,11]. Several TRP isoforms, including TRPM3, are localised mainly to the cytoplasm in primary sensory neurons, and modulation of intracellular TRP sorting and trafficking may significantly modify channel function and neuronal excitability[12]. Currently, the mechanisms of TRPM3 membrane trafficking are unclear. It is well known that kinesin superfamily proteins (KIFs) constitute 15 kinesin families, which are termed kinesin 1 to kinesin 14B and a large fraction of these motor proteins mediate the microtubule-dependent transport of cargo proteins in various cell types[13]. The

motor protein KIF13B has been shown to mediate TRPV1 membrane transport and to not affect total TRPV1 expression in DRG neurons[14]. A previous study demonstrated that KIF5B overexpression improves the cell surface and axonal distribution of Nav1.8 and that KIF5B knockdown reduces the current density of Nav1.8 in DRG neurons; these results indicate that the anterograde axonal transport of Nav1.8 occurs via a mechanism involving motor proteins[15]. Our previous study showed that KIF3A-mediated Nav1.6 trafficking in DRG neurons is involved in chronic pain[16]. The kinesin 2 family protein KIF17 is localised predominantly in the somata and dendrites of neurons[17]. It has been indicated that KIF17 plays a significant role in the distribution of NR2B, GluR5 and Kv4.2 in distal dendrites[17–19]. In addition, KIF17 interacts with nuclear RNA export factor 2 (NXF2), enabling the bidirectional transport of mRNA in dendrites[20]. KIF17 also mediates intraflagellar transport of cargo to the distal tips of flagella or cilia, thereby regulating the function of flagella[20]. The role of KIF motor proteins in TRPM3 membrane trafficking, however, is unclear.

In the present study, we revealed that ATF4 controls thermal, but not mechanical, sensitivity. Furthermore, ATF4 mediates the membrane localisation of TRPM3 in DRG neurons by interacting with the motor protein KIF17 and thus contributes to thermal sensitivity. Ultimately, our work reveals a cellular mechanism essential for heat nociception.

## Results

**ATF4 is expressed in primary sensory neurons.** We first evaluated the profile of ATF4 expression in the peripheral sensory nervous system by immunoblotting and found that ATF4 was abundant in the DRG, dorsal root and sciatic nerve (Supplementary Fig. 1a). Immunostaining and in situ hybridisation showed that ATF4 protein and mRNA were expressed in most DRG neurons (Fig. 1a; Supplementary Fig. 1b, d). ATF4, a

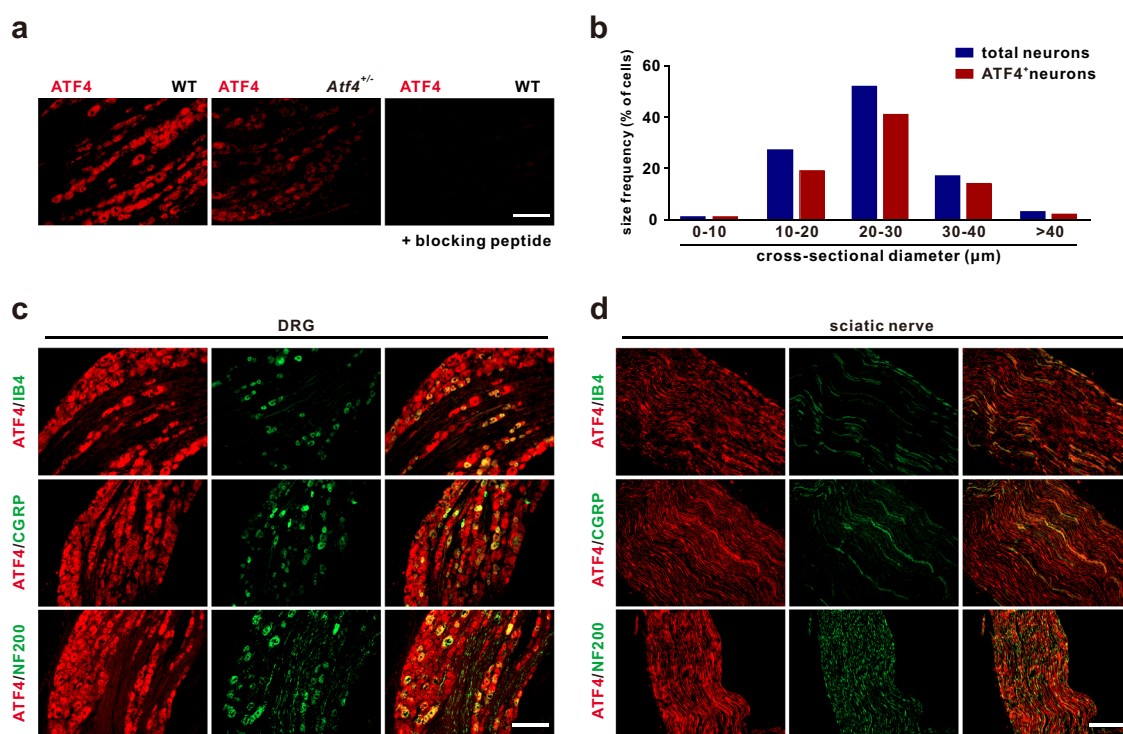

**Fig. 1 Distribution of ATF4 in primary sensory neurons. a** Left, image of immunostaining showing broad ATF4 expression in mouse DRG neurons. Middle, ATF4 expression was decreased in *Atf4*+/− mice. Right, absence of ATF4 immunostaining upon treatment with a blocking peptide. Scale bar, 200 μm. **b** Size frequency distribution of ATF4-positive and total neurons in the mouse DRG. A total of 1489 neurons from *n* = 4 mice were analysed. **c, d** Colocalization of ATF4 and cell markers (IB4, CGRP and NF200) in the DRG (**c**) and sciatic nerve (**d**). Scale bar, 200 μm.

transcription factor, was mainly expressed in the cytoplasm and at low levels in the nuclei of mouse DRG neurons (Supplementary Fig. 1b, c). Size frequency analysis showed broad expression of ATF4 in small, medium, and large DRG neurons (Fig. 1b). Double immunostaining confirmed that ATF4 was expressed in nonpeptidergic IB4-positive, calcitonin gene-related peptide (CGRP)-positive and neurofilament-200 (NF200)-positive DRG neurons (Fig. 1c). Colocalization analysis showed that 35.1% of ATF4-positive cells expressed IB4, 35.0% expressed CGRP and 19.6% expressed NF200 (Supplementary Fig. 1e). ATF4 was also colocalized with IB4, CGRP and NF200 in the sciatic nerve (efferent C, Aδ and Aβ fibres) and spinal dorsal horn (afferent C, Aδ and Aβ fibres) (Fig. 1d; Supplementary Fig. 1f). Together, these results demonstrated that ATF4 is expressed in primary sensory neurons, including their peripheral and central axons.

**Loss of ATF4 selectively impairs noxious heat sensation**. To determine the function of ATF4, we intrathecally injected ATF4 siRNA to knockdown the expression of ATF4 in DRGs (Supplementary Fig. 1g, h). We found that there were no differences in mechanical sensitivity between the ATF4 siRNA-injected and scrambled siRNA-injected mice in the von Frey (Fig. 2a), dynamic (Fig. 2b), tape (Fig. 2c), tail clip (Fig. 2d) and pinprick (Fig. 2e) tests. However, the thermal withdrawal latencies of the ATF4 siRNA-injected mice were significantly longer than those of the scrambled siRNA-injected mice in the Hargreaves (radiant heat test) (Fig. 2f), tail-flick (Fig. 2g) and hot plate (Fig. 2h) tests. No differences in responses in the evaporative cooling test were observed between the scrambled siRNA-injected and ATF4 siRNA-injected mice (Fig. 2i). Knockdown of ATF4 did not change the locomotor activity of mice in the rotarod test (Fig. 2j). We also investigated the role of ATF4 in pathologic pain. The behavioural data showed that knockdown of ATF4 significantly alleviated thermal hyperalgesia (Fig. 2k), but not mechanical allodynia (Fig. 2l), induced by spinal nerve ligation (SNL), which was used to simulate neuropathic pain. Importantly, SNL also increased the expression of ATF4 in DRG tissues (Supplementary Fig. 1i); this increase started on day 4 and continued to the end of the experiment (day 14). The changes in ATF4 expression were accompanied by SNL-induced neuropathic pain (Supplementary Fig. 1j, k). Consistently, knockdown of ATF4 alleviated heat hyperalgesia (Fig. 2m), but not mechanical allodynia (Fig. 2n), induced by complete Freund's adjuvant (CFA), which was used to induce inflammatory pain. Therefore, ATF4 selectively regulates heat nociception. TRPM3 regulates heat transduction and is especially critical for the development of heat nociception[10,21]. We tested the effects of ATF4 siRNA on spontaneous pain induced by the TRPM3 agonists pregnenolone sulfate (PS) and CIM2016[10,22]. Intraplantar injection of PS (2.5 nmol/paw) or CIM0216 (2.5 nmol/paw) induced rapid spontaneous pain (duration or number of licking or flinching behaviours within 2 min) in scrambled siRNA-injected mice, but this spontaneous pain was abated in ATF4 siRNA-injected mice (Fig. 2o, p). Similarly, knockdown of ATF4 reduced spontaneous pain (duration of licking or flinching behaviour within 5 min) induced by intrathecal injection of PS (1.25 nmol/mouse) or CIM0216 (1.25 nmol/mouse) (Fig. 2q). Furthermore, to investigate the specific role of ATF4 in pain regulation, we constructed Atf4 knockout mice. Considering that homozygous Atf4 mice exhibit postnatal lethality or functional defects, we used heterozygous mice for the experiments[23]. The protein expression of ATF4 in DRG neurons was lower in $Atf4^{+/-}$ mice than in WT mice (Fig. 1a). Notably, $Atf4^{+/-}$ mice showed no developmental defects in sensory neurons or their innervations. The central innervations of primary afferents in the spinal dorsal horn were

comparable in WT and heterozygote mice (Supplementary Fig. 2). The distribution patterns of primary sensory neurons, including small nociceptive (CGRP$^+$ peptidergic and IB4$^+$ non-peptidergic) neurons and large A-fibre DRG (NF200$^+$) neurons, as well as the total population of sensory neurons, were also unaltered in the DRG tissues of heterozygous mice (Supplementary Fig. 3). $Atf4^{+/-}$ mice displayed normal mechanical sensitivity in the von Frey (Fig. 3a), dynamic (Fig. 3b), tape (Fig. 3c), tail clip (Fig. 3d) and pinprick (Fig. 3e) tests. However, $Atf4^{+/-}$ mice showed lower thermal sensitivity than WT mice in the Hargreaves (Fig. 3f), tail-flick (Fig. 3g) and hot plate (Fig. 3h) tests. $Atf4^{+/-}$ mice showed normal evaporative cooling and locomotor function (Fig. 3i, j). The behavioural data showed that loss of ATF4 markedly relieved heat hyperalgesia (Fig. 3k, m), but not mechanical allodynia (Fig. 3l, n), induced by SNL or CFA. Spontaneous pain induced by intraplantar or intrathecal injection of PS or CIM0216 was also alleviated in $Atf4^{+/-}$ mice (Fig. 3o–q). Therefore, loss of ATF4 in mice results in defects in heat nociception, which may be due to compromised TRPM3 signalling.

**Re-expression of ATF4 rescues heat nociception in $Atf4^{+/-}$ mice**. To further confirm the role of ATF4 in heat nociception, we intrathecally injected adeno-associated virus (AAV) (rAAV-CMV-Atf4-2A-EGFP-WPRE-PA) to re-express ATF4 in the DRGs of $Atf4^{+/-}$ mice (rescued mice) (Supplementary Fig. 4a) and then subjected the rescued mice to behavioural tests. The behavioural data showed that re-expression of ATF4 prominently rescued heat sensitivity (Fig. 3f–h, k, m) but did not increase mechanical (Fig. 3a–e, l, n) or cold (Fig. 3i) sensitivity in $Atf4^{+/-}$ mice. Importantly, PS- and CIM0216-induced spontaneous pain was also rescued in $Atf4^{+/-}$ mice after re-expression of ATF4 (Fig. 3o–q). In addition, we intrathecally injected AAV (rAAV-CMV-Atf4-2A-EGFP-WPRE-PA) in normal mice to overexpress ATF4 in the DRGs (Supplementary Fig. 4b, c). We found that there were no differences in mechanical sensitivity between the ATF4 overexpression and control mice in the von Frey (Fig. 4a), dynamic (Fig. 4b), tape (Fig. 4c), tail clip (Fig. 4d) and pinprick (Fig. 4e) tests. However, the thermal withdrawal latencies of ATF4-overexpressing mice were significantly shorter than those of the control mice in the Hargreaves (radiant heat test) (Fig. 4f), tail-flick (Fig. 4g) and hot plate (Fig. 4h) tests. No differences in responses to the evaporative cooling test were observed between the control and ATF4-overexpressing mice (Fig. 4i). ATF4-overexpressing mice showed normal locomotor activity in the rotarod test (Fig. 4j). Furthermore, overexpression of ATF4 also increased spontaneous pain induced by intraplantar or intrathecal injection of PS or CIM0216 (Fig. 4k–m). Therefore, the results further confirm that ATF4 may control heat pain by regulating the TRPM3 signalling pathway.

**ATF4 mediates TRPM3 trafficking in DRG neurons**. We next aimed to identify the underlying mechanism through which ATF4 controls noxious heat sensation. TRPV1, TRPA1 and TRPM3 play critical roles in thermal nociception[10,24]. Thus, we examined whether TRPV1, TRPA1 and TRPM3 participate in the regulation of heat nociception by ATF4. Compared with scrambled siRNA, ATF4 siRNA (intrathecal) significantly decreased the membrane accumulation of TRPM3 (Supplementary Fig. 5a), but not of TRPA1 (Supplementary Fig. 5e) or TRPV1 (Supplementary Fig. 5f) in the DRG. Knockdown of ATF4 also did not alter spontaneous pain (the duration or number of licking or flinching behaviours within 2 min) induced by intraplantar injection of capsaicin (1 nmol/paw) (Supplementary Fig. 5g). Interestingly, compared with scrambled siRNA administration, ATF4 knockdown increased TRPM3 accumulation in the cytoplasm of DRG

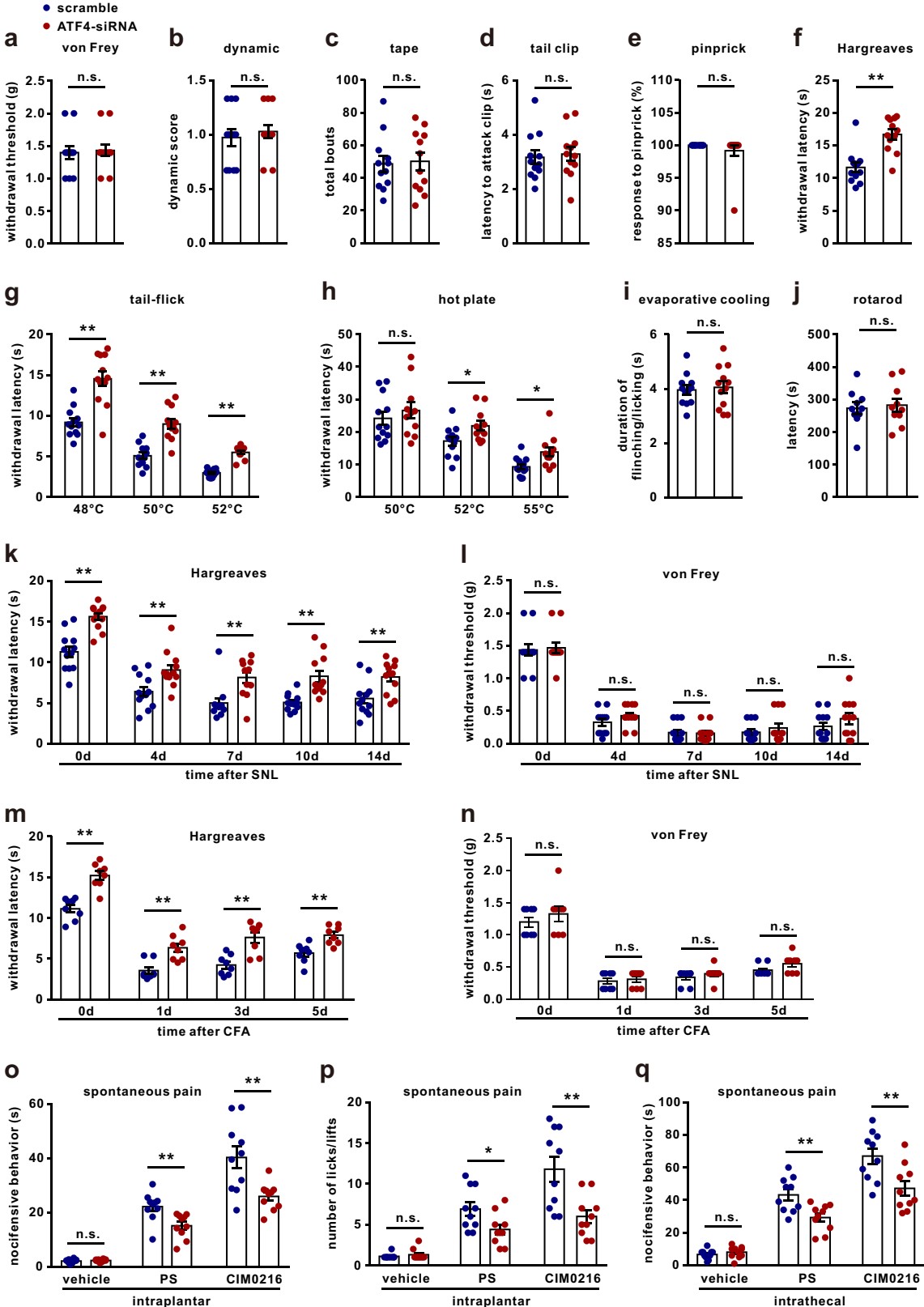

neurons (Supplementary Fig. 5b) but did not change total TRPM3 expression (Supplementary Fig. 5c). Notably, the proportion of TRPM3 trafficking significantly decreased in the DRGs of mice after ATF4 siRNA treatment (Supplementary Fig. 5d). Consistently, a decrease in the level of membrane TRPM3 (Fig. 5a) and an increase in the level of cytoplasm TRPM3

(Fig. 5b) but no change in total TRPM3 levels (Fig. 5c) were observed in DRG tissues from $Atf4^{+/-}$ mice compared to those from WT mice. The ratio of TRPM3 trafficking was also significantly decreased in the DRGs of $Atf4^{+/-}$ mice compared to those of WT mice (Supplementary Fig. 5h). Surface biotinylation analysis revealed a reduction in TRPM3 surface expression in the

**Fig. 2 Knockdown of ATF4 decreases thermal sensitivity. a–j** The behaviours of ATF4 siRNA- and scrambled siRNA-injected mice were evaluated by the von Frey (**a**), dynamic (**b**), tape (**c**), tail clip (**d**), pinprick (**e**), Hargreaves (**f**), tail-flick (**g**), hot plate (**h**), evaporative cooling (**i**) and rotarod (**j**) tests. $n = 12$ mice per group in **a–g**, **i**. $n = 12$ mice in scrambled and $n = 11$ mice in ATF4 siRNA group in **h**. $n = 10$ mice per group in **j**. $t_{22} = 4.783$, $P < 0.0001$ in **f**. $t_{22} = 5.163$, $P < 0.0001$ in 48 °C; $t_{22} = 5.661$, $P < 0.0001$ in 50 °C; $t_{22} = 10.30$, $P < 0.0001$ in 52 °C in **g**. $t_{21} = 2.357$, $P = 0.0282$ in 52 °C; $t_{21} = 2.804$, $P = 0.0106$ in 55 °C in **h**. **k**, **l** ATF4 siRNA abated SNL-induced thermal hyperalgesia (**k**) but not mechanical allodynia (**l**). $n = 12$ mice per group. $F_{(1,22)} = 56.9$, $P < 0.0001$ in day 0, $P = 0.0095$ in day 4, $P = 0.0013$ in day 7, $P = 0.0009$ in day 10, $P = 0.0089$ in day 14 in **k**. **m**, **n** ATF4 siRNA abated CFA-induced heat hyperalgesia (**m**) but not mechanical allodynia (**n**). $n = 8$ mice per group. $F_{(1,14)} = 121.6$, $P < 0.0001$ in day 0, $P = 0.0007$ in day 1, $P < 0.0001$ in day 3, $P = 0.0098$ in day 5 in **m**. **o**, **p** Spontaneous pain: total duration (**o**) or number (**p**) of nocifensive behaviour (paw licking or flinching within 2 min) in response to intraplantar injection of vehicle, PS (2.5 nmol/paw) or CIM0216 (2.5 nmol/paw) into ATF4 siRNA and scrambled mice. $n = 10$ mice per group. $t_{18} = 3.231$, $P = 0.0046$ in PS, $t_{18} = 3.286$, $P = 0.0041$ in CIM0216 in **o**. $t_{18} = 2.429$, $P = 0.0258$ in PS, $t_{18} = 3.327$, $P = 0.0038$ in CIM0216 in **p. q** Spontaneous pain: total duration of nocifensive behaviour (within 5 min) in response to intrathecal injection of vehicle, PS (1.25 nmol/paw) or CIM0216 (1.25 nmol/paw) into ATF4 siRNA and scrambled mice. $n = 10$ mice per group. $t_{18} = 3.281$, $P = 0.0042$ in PS, $t_{18} = 3.024$, $P = 0.0073$ in CIM0216. **a–j**, **o–q** Two.tailed Independent Student's $t$ test; **k–n** Two-way ANOVA followed by Bonferroni's multiple comparisons test. $*P < 0.05$, $**P < 0.01$, n.s. means not significant. The error bars indicate the SEMs.

DRG neurons of $Atf4^{+/-}$ mice compared to those of WT mice (Fig. 5d). Co-immunoprecipitation (co-IP) revealed that ATF4 potentially interacted with TRPM3 in DRG protein extracts (Fig. 5e). Furthermore, we performed high-resolution imaging by structured illumination microscopy (SIM) and found that 62.0 ± 5.4% of ATF4 was colocalized with TRPM3 and that 52.0 ± 5.3% of TRPM3 was colocalized with ATF4 in DRG neurons (Fig. 5f; Supplementary Fig. 5i). In addition, the GST pull-down assay showed a direct interaction between purified ATF4 and TRPM3 (Fig. 5g). To determine how ATF4 interacts with TRPM3, we constructed three expression vectors, namely, ATF4-His (aa 1-116), ATF4-His (aa 117-232), and ATF4-His (aa 233-349), and co-expressed them with the TRPM3-Flag construct in HEK293T cells. The region 233-349 aa region of ATF4 was found to be the major binding site for TRPM3 (Fig. 5h). Additionally, ATF4 overexpression increased membrane TRPM3 levels (Fig. 5i) and decreased cytoplasm TRPM3 levels (Fig. 5j) without changing total TRPM3 levels (Fig. 5k) in DRG tissues. A higher proportion of TRPM3 trafficking was observed in the DRGs of ATF4-overexpressing mice than in those of control mice (Supplementary Fig. 5j). The membrane expression of TRPM3 was also increased in the DRGs of SNL mice (Supplementary Fig. 5k). The specificity of the TRPM3 antibodies were validated by loss of TRPM3 immunostaining in the DRG neurons of $Trpm3^{-/-}$ mice (Supplementary Fig. 5l, m) and further confirmed by the absence of staining in the DRG after co-incubation of the antibody with a blocking peptide (Supplementary Fig. 5l). To further study the effect of the transcriptional function of ATF4, a transcription factor, on TRPM3 channels, we performed cytoplasmic and nuclear separation experiments. As ISRIB can significantly inhibit the expression of ATF4[25], we intrathecally injected ISRIB (300 ng)[26] and measured ATF4 levels in the cytoplasm and nuclei of DRG neurons at different time points. A significant decrease in ATF4 was detected in the cytoplasm 30 min after injection (Supplementary Fig. 6a), whereas ATF4 expression was decreased in the nuclei 90 min after injection but not 30 min (Supplementary Fig. 6b). That is, 30 min after intrathecal injection of ISRIB, it may have inhibited the function of cytoplasmic ATF4 without affecting the nuclear transcription function of ATF4. Furthermore, 30 min after intrathecal injection of ISRIB reduced the expression of TRPM3 on the membrane (Supplementary Fig. 6c) but did not change the total expression of TRPM3 in sensory neurons (Supplementary Fig. 6d). And 30 min after intrathecal injection of ISRIB also significantly decreased the heat sensitivity of mice (Supplementary Fig. 6e). Then, we overexpressed a transcriptionally inactive form of ATF4 (rAAV-CMV-Atf4-ΔbZIP-2A-EGFP-WPRE-PA) in $Atf4^{+/-}$ mice and the expression of TRPM3 in sensory neurons were tested. The transcriptionally inactive form of ATF4 (rAAV-CMV-Atf4-ΔbZIP-2A-EGFP-

WPRE-PA) was generated by site-directed mutagenesis with 6 amino acid substitutions within the DNA-binding domain ($^{292}$RYRQKKR$^{298}$ to $^{292}$GYLEAAA$^{298}$)[27–29]. The data showed that overexpression of a transcriptionally inactive form of ATF4 in $Atf4^{+/-}$ mice increased the membrane expression of TRPM3 but did not altered the total expression of TRPM3 in sensory neurons (Supplementary Fig. 6f, g). And overexpressing the transcriptionally inactive form of ATF4 also significantly enhanced the heat sensitivity of $Atf4$ heterozygote mice (Supplementary Fig. 6h). To further investigate whether ATF4 has a direct transcriptional regulatory effect on the $Trpm3$ gene, we performed a luciferase reporter assay to determine the effects of ATF4 on $Trpm3$ promoter activity. The data showed that knockdown of ATF4 did not alter the luciferase activity of the $Trpm3$ promoter (Supplementary Fig. 6i). These results suggest that ATF4 may have no transcriptional regulatory effect on the TRPM3. To assess TRPM3 activity, we recorded TRPM3 currents in DRG neurons. TRPM3 agonist (CIM0216, 3 μM) and inhibitor (isosakuranetin, 10 μM) were used to verify the specificity of the recorded TRPM3 currents[22] (Fig. 5l–q). The results showed that 60%, 55% and 67% of the neurons were activated by CIM0216 in the WT ($n = 10$), $Atf4^{+/-}$ ($n = 11$) and rescued ($n = 9$) mice, respectively. The outward and inward TRPM3 currents induced by CIM0216 were decreased in $Atf4^{+/-}$ mice compared to WT mice (Fig. 5r). Interestingly, re-expression of ATF4 reversed the decrease of TRPM3 currents in $Atf4^{+/-}$ mice (Fig. 5r). Thus, ATF4 interacts with TRPM3 and increases its trafficking to the surface of DRG neurons to control heat sensitivity.

**ATF4/TRPM3 interaction is facilitated by heat stimulation and maintains TRPM3 levels in the cell membrane.** It has been shown that heat stimulation (43 °C) of DRG neurons can change the molecular interactions and membrane expression of ion channels within 30 s[30]. We further investigated the effects of heat stimulation on the interaction between ATF4 and TRPM3 and on the surface expression of TRPM3. The hind paws of mice were soaked in a 43 °C water bath for 30 s, and the mice were kept at room temperature (RT) for 2, 5, 10, 20 or 40 min. Then, behavioural and immunoblotting experiments were performed (Fig. 6a). The results showed that the interaction between ATF4 and TRPM3 in DRG tissues was enhanced at 2, 5, 10 and 20 min and returned to normal levels at 40 min after heat stimulation (Fig. 6b). The membrane abundance of TRPM3 in DRG tissues was also increased at 2, 5, 10 and 20 min but not 40 min after heat stimulation (Fig. 6c). Heat stimulation of the paws did not change the expression of ATF4 in DRG tissues at any time point (Supplementary Fig. 7a). Furthermore, we detected the effects of heat stimulation of the paws on the expression of TRPM3 in nerve

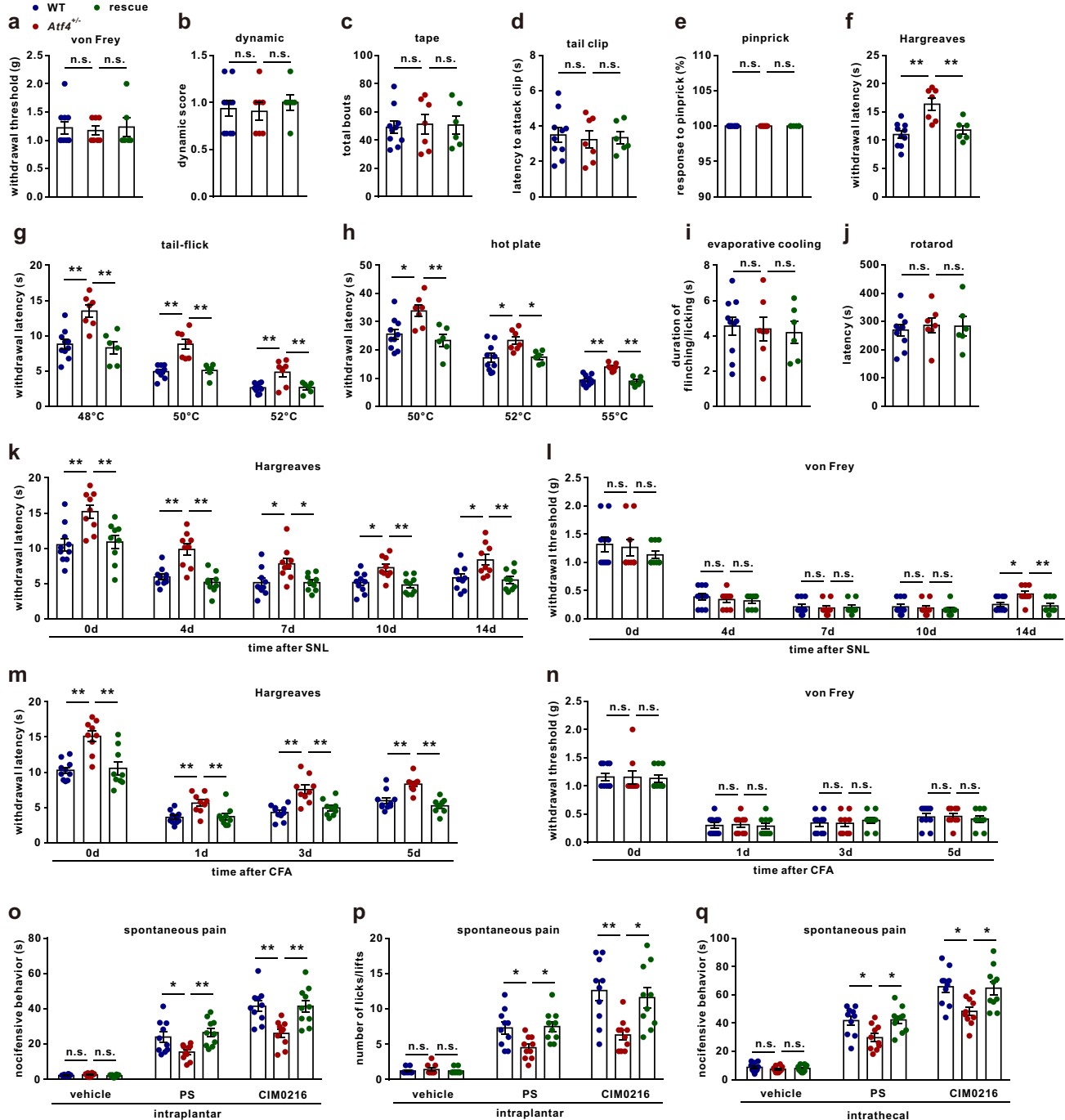

endings. The results showed that heat stimulation increased the expression of TRPM3 in nerve endings (Supplementary Fig. 7b, c). Furthermore, a TRPM3 agonist (CIM0216, 2.5 nmol/paw) was intraplantarly injected into the hindpaws of mice kept at RT and mice subjected to heat stimulation. We found that the decrease in thermal withdrawal thresholds was more predominant in heat-stimulated mice than in mice kept at RT (Supplementary Fig. 7d, e). We further studied the effects of direct heat stimulation of isolated DRG neurons on the membrane expression of TRPM3 (Fig. 6d). The results showed that the membrane TRPM3 was increased at 2, 5, 10 and 20 min, but not at 40 min after direct heat stimulation of isolated DRG neurons (Fig. 6e). Consistently, the thermal withdrawal threshold was significantly decreased in naïve mice at 2, 5, 10 and 20 min, but not at 40 min after paws heat stimulation (Fig. 6f). Importantly, the threshold was not

changed in ATF4 knockdown or $Atf4^{+/-}$ mice at 2, 5, 10, 20 or 40 min after paws heat stimulation (Fig. 6g, h). And the decreased of threshold in mice after paws heat stimulation was also abolished by intrathecal injection of ISRIB for 30 min (Supplementary Fig. 7f), indicated that the non-transcriptional function of ATF4 is involved in this process. These results revealed that the heat-facilitated ATF4/TRPM3 interaction may stabilise TRPM3 channels on the cell membrane and contribute to heat nociception.

**KIF17 is involved in ATF4-dependent TRPM3 membrane trafficking.** We next investigated the underlying mechanism of ATF4-dependent TRPM3 membrane trafficking. Studies have shown that motor protein kinesins play a crucial role in neuronal

**Fig. 3 Loss of ATF4 impaired heat nociception, and re-expression of ATF4 rescued heat nociception. a–j** The behaviours of WT, $Atf4^{+/-}$ and rescued mice were evaluated by the von Frey (**a**), dynamic (**b**), tape (**c**), tail clip (**d**), pinprick (**e**), Hargreaves (**f**), tail-flick (**g**), hot plate (**h**), evaporative cooling (**i**) and rotarod (**j**) tests. $n = 10$ mice in WT, $n = 7$ mice in $Atf4^{+/-}$ and $n = 6$ mice in rescue group. $F_{(2,20)} = 12.11$, $P = 0.0004$ in WT vs. $Atf4^{+/-}$, $P = 0.0050$ in $Atf4^{+/-}$ vs. rescue in **f**. $F_{(2,20)} = 12.59$, $P = 0.0008$ in WT vs. $Atf4^{+/-}$, $P = 0.0009$ in $Atf4^{+/-}$ vs. rescue in 48 °C; $F_{(2,20)} = 20.67$, $P < 0.0001$ in WT vs. $Atf4^{+/-}$, $P = 0.0002$ in $Atf4^{+/-}$ vs. rescue in 50 °C; $F_{(2,20)} = 8.924$, $P = 0.0024$ in WT vs. $Atf4^{+/-}$, $P = 0.0076$ in $Atf4^{+/-}$ vs. rescue in 52 °C in **g**. $F_{(2,20)} = 7.097$, $P = 0.0159$ in WT vs. $Atf4^{+/-}$, $P = 0.0066$ in $Atf4^{+/-}$ vs. rescue in 50 °C; $F_{(2,20)} = 5.678$, $P = 0.0135$ in WT vs. $Atf4^{+/-}$, $P = 0.0375$ in $Atf4^{+/-}$ vs. rescue in 52 °C; $F_{(2,20)} = 21.31$, $P < 0.0001$ in WT vs. $Atf4^{+/-}$, $P < 0.0001$ in $Atf4^{+/-}$ vs. rescue in 55 °C in **h**. **k, l** Knockout and rescue of ATF4 impaired and rescued SNL-induced heat hyperalgesia (**k**) but not mechanical allodynia (**l**), respectively. $n = 10$ mice in WT, $n = 9$ mice in $Atf4^{+/-}$ and $n = 9$ mice in rescue group. $F_{(2,25)} = 36$; $P = 0.0038$ in WT vs. $Atf4^{+/-}$, $P = 0.0098$ in $Atf4^{+/-}$ vs. rescue in day 0; $P = 0.0002$ in WT vs. $Atf4^{+/-}$, $P < 0.0001$ in $Atf4^{+/-}$ vs. rescue in day 4; $P = 0.0145$ in WT vs. $Atf4^{+/-}$, $P = 0.0170$ in $Atf4^{+/-}$ vs. rescue in day 7; $P = 0.0105$ in WT vs. $Atf4^{+/-}$, $P = 0.0034$ in $Atf4^{+/-}$ vs. rescue in day 10; $P = 0.0174$ in WT vs. $Atf4^{+/-}$, $P = 0.0087$ in $Atf4^{+/-}$ vs. rescue in day 14 in **k**. $F_{(2,25)} = 1.112$; $P = 0.0171$ in WT vs. $Atf4^{+/-}$, $P = 0.0074$ in $Atf4^{+/-}$ vs. rescue in day 14 in **l. m, n** Knockout and rescue of ATF4 impaired and rescued CFA-induced heat hyperalgesia (**m**) but not mechanical allodynia (**n**), respectively. $n = 10$ mice in WT, $n = 9$ mice in $Atf4^{+/-}$ and $n = 9$ mice in rescue group. $F_{(2,25)} = 108.5$; $P = 0.0002$ in WT vs. $Atf4^{+/-}$, $P = 0.0005$ in $Atf4^{+/-}$ vs. rescue in day 0; $P = 0.0045$ in WT vs. $Atf4^{+/-}$, $P = 0.0086$ in $Atf4^{+/-}$ vs. rescue in day 1; $P = 0.0001$ in WT vs. $Atf4^{+/-}$, $P = 0.0017$ in $Atf4^{+/-}$ vs. rescue in day 3; $P = 0.0006$ in WT vs. $Atf4^{+/-}$, $P < 0.0001$ in $Atf4^{+/-}$ vs. rescue in day 5. **o, p** Spontaneous pain: total duration (**o**) or number (**p**) of nocifensive behaviour (paw licking or flinching within 2 min) in response to intraplantar injection of vehicle, PS (2.5 nmol/paw) or CIM0216 (2.5 nmol/paw) into WT, $Atf4^{+/-}$ and rescued mice. $n = 10$ mice per group. $F_{(2,27)} = 6.375$, $P = 0.0368$ in WT vs. $Atf4^{+/-}$, $P = 0.0056$ in $Atf4^{+/-}$ vs. rescue in PS; $F_{(2,27)} = 9.174$, $P = 0.0027$ in WT vs. $Atf4^{+/-}$, $P = 0.0026$ in $Atf4^{+/-}$ vs. rescue in CIM0216 in **o**. $F_{(2,27)} = 5.976$, $P = 0.0201$ in WT vs. $Atf4^{+/-}$, $P = 0.0123$ in $Atf4^{+/-}$ vs. rescue in PS; $F_{(2,27)} = 7.569$, $P = 0.0033$ in WT vs. $Atf4^{+/-}$, $P = 0.0138$ in $Atf4^{+/-}$ vs. rescue in CIM0216 in **p. q** Spontaneous pain: total duration of nocifensive behaviour (in 5 min) in response to intrathecal injection of vehicle, PS (1.25 nmol/paw) or CIM0216 (1.25 nmol/paw) into WT, $Atf4^{+/-}$ and rescued mice. $n = 10$ mice per group. $F_{(2,27)} = 5.849$, $P = 0.0194$ in WT vs. $Atf4^{+/-}$, $P = 0.0146$ in $Atf4^{+/-}$ vs. rescue in PS; $F_{(2,27)} = 5.986$, $P = 0.0127$ in WT vs. $Atf4^{+/-}$, $P = 0.0192$ in $Atf4^{+/-}$ vs. rescue in CIM0216. **a–j, o–q,** One-way ANOVA followed by Tukey's multiple comparisons test. **k–n** Two-way ANOVA followed by Bonferroni's multiple comparisons test. *$P < 0.05$, **$P < 0.01$, n.s. means not significant. The errors bars indicate the SEMs.

cargo trafficking[13]. To determine whether motor protein kinesins contribute to ATF4-dependent TRPM3 trafficking, we first examined the interaction between ATF4 and several motor proteins by co-IP. The results revealed that ATF4 showed a potent interaction with KIF17 and a relatively weak interaction with KIF5B, whereas ATF4 barely interacted with KIF3A, KIF3B, KIF5A or KIFC2 in mice DRG protein extracts (Fig. 7a–f). Furthermore, we performed high-resolution imaging by SIM and found that $35.0 \pm 6.6\%$ of ATF4 was colocalized with KIF17 and that $32.8 \pm 6.3\%$ of KIF17 was colocalized with ATF4 in DRG neurons (Fig. 7g; Supplementary Fig. 8a). In addition, the pull-down assay showed a direct interaction between purified ATF4 and KIF17 (Fig. 7h). To define how ATF4 interacts with KIF17, we constructed three expression vectors, namely, ATF4-His (aa 1-116), ATF4-His (aa 117-232), and ATF4-His (aa 233-349), and co-expressed them with the KIF17-Flag construct in HEK293T cells. The 117-232 aa region of ATF4 was found to be the major binding site for KIF17 (Fig. 7i). Next, we explored whether KIF17 participates in ATF4-dependent TRPM3 membrane trafficking. The co-IP results revealed that TRPM3 showed a potential interaction with KIF17 (Fig. 7j), and the interaction between TRPM3 and KIF17 was decreased in the DRGs of ATF4 knockdown mice (Fig. 7j). Additionally, knockdown of ATF4 decreased the colocalization of TRPM3 and KIF17 in SIM images (Fig. 7k), whereas the colocalization between TRPM3 and KIF17 in SIM images was increased after ATF4 overexpression (Fig. 7k). This suggested that ATF4 may function as an adaptor protein that mediates the interaction between KIF17 and TRPM3 in DRG neurons. Considering the potential interactions among KIF17, ATF4 and TRPM3, we performed triple fluorescence staining of DRG sections and found that the KIF17, ATF4 and TRPM3 proteins were expressed in the same subset of DRG neurons (Fig. 8a). In situ hybridisation showed that *KIF17*, *ATF4* and *TRPM3* mRNAs were co-expressed in the same subset of neurons (Fig. 8b). Furthermore, we performed high-resolution imaging by SIM. The data showed that $21.3 \pm 3.8\%$ of ATF4 was colocalized with TRPM3 and KIF17, $9.8 \pm 1.6\%$ of TRPM3 was colocalized with ATF4 and KIF17, and $28.2 \pm 2.1\%$ of KIF17 was colocalized with ATF4 and TRPM3 in cultured DRG neurons (Fig. 8c; Supplementary Fig. 8b;

Supplementary Movie 1). We next intrathecally injected KIF17-siRNA to knockdown the expression of KIF17 in DRG tissues (Supplementary Fig. 8c) and found that KIF17 siRNA significantly decreased the surface accumulation of TRPM3 in DRG neurons (Fig. 8d). Importantly, we intrathecally injected AAV (rAAV-CMV-*Kif17*-2A-EGFP-WPRE-PA) to overexpress KIF17 in DRG tissues (Supplementary Fig. 8d) and found that overexpression of KIF17 markedly increased TRPM3 membrane trafficking in DRG neurons (Fig. 8e). Consistently, knockdown and overexpression of KIF17 significantly increased and decreased thermal withdrawal thresholds in the Hargreaves test, respectively (Fig. 8f, g). Interestingly, ATF4 siRNA suppressed the increase in TRPM3 expression in the membrane induced by KIF17 overexpression (Fig. 8h). KIF17 siRNA also inhibited the increased expression of TRPM3 on the cell surface induced by ATF4 overexpression (Fig. 8i). These results suggested that the motor protein KIF17 plays a critical role in ATF4-dependent TRPM3 trafficking to the membrane.

**CXCL12 mediates ATF4 upregulation via CXCR4.** Our studies and those of our peers showed that chemokine C-X-C motif ligand 12 (CXCL12) is markedly increased in the DRG after induction of neuropathic pain by nerve injury or a chemotherapeutic agent[31–33]. Thus, to investigate the underlying mechanism of ATF4 up-regulation by nerve injury, we examined whether CXCL12 promotes ATF4 upregulation. The data showed that incubation with CXCL12 increased the expression of ATF4 in cultured DRG neurons (Fig. 9a). Intrathecal delivery of CXCL12 in vivo also increased ATF4 expression in the DRG (Fig. 9b), suggesting that CXCL12 stimulates ATF4 upregulation. Consistently, intrathecal injection of CXCL12 induced heat hyperalgesia in normal mice, whereas ATF4 siRNA inhibited this effect (Fig. 9c). The chemokines C-X-C motif receptor 4 (CXCR4) and C-X-C motif receptor 7 (CXCR7) are receptors of CXCL12 and participate in physiological and pathological processes regulated by CXCL12[34]. We further investigated the role of CXCR4 and CXCR7 in the CXCL12-mediated upregulation of ATF4. We intrathecally injected CXCR4 siRNA or CXCR7 siRNA to knock

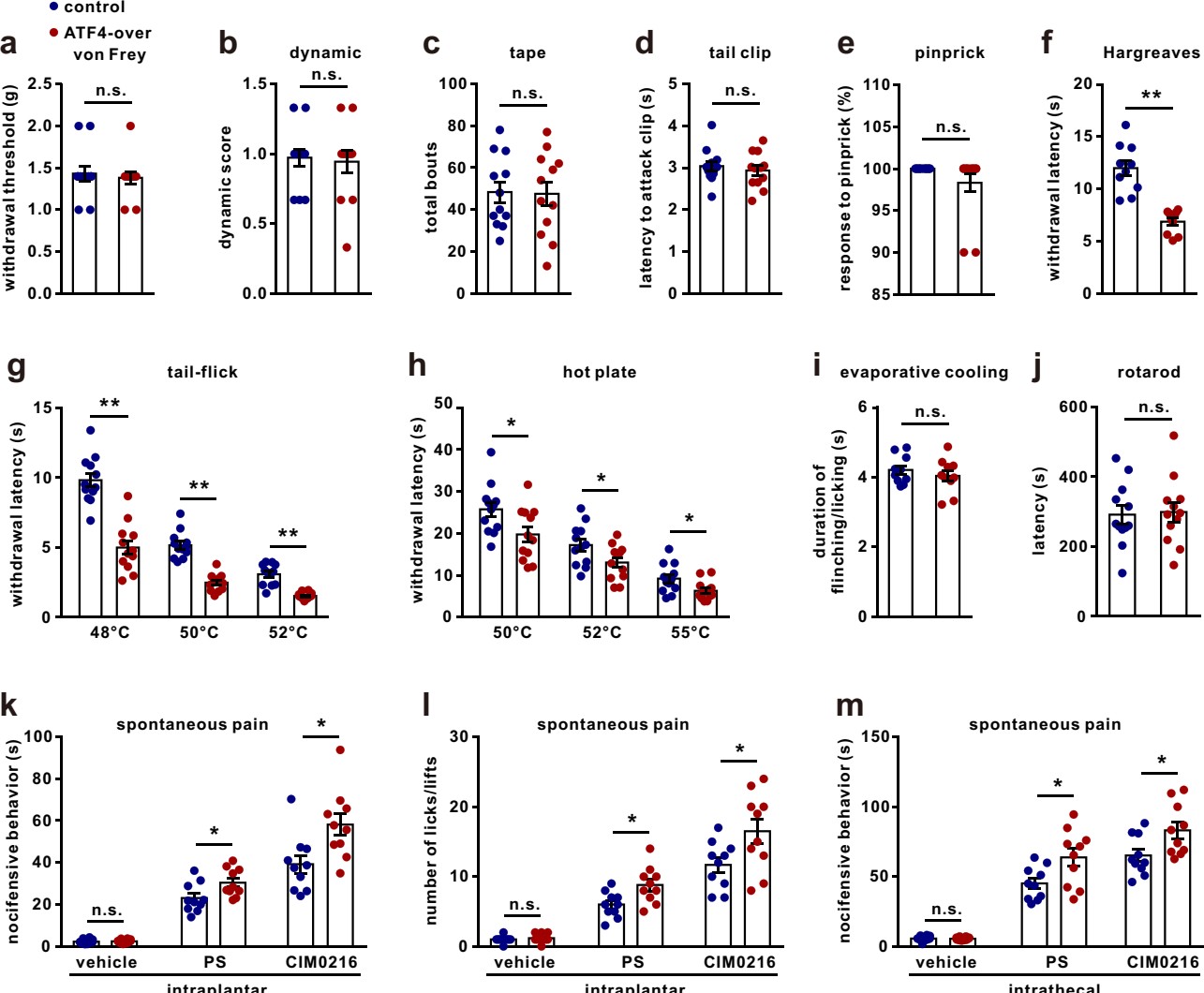

**Fig. 4 Overexpression of ATF4 increases heat sensitivity. a–j** The behaviours of ATF4-overexpressing and control mice were evaluated by the von Frey (**a**), dynamic (**b**), tape (**c**), tail clip (**d**), pinprick (**e**), Hargreaves (**f**), tail-flick (**g**), hot plate (**h**), evaporative cooling (**i**) and rotarod (**j**) tests. $n = 12$ mice per group in **a–e**, **g**, **h**, **j**. $n = 10$ mice per group in **f**, **i**. $t_{18} = 6.308$, $P < 0.0001$ in **f**. $t_{22} = 6.893$, $P < 0.0001$ in 48 °C; $t_{22} = 8.009$, $P < 0.0001$ in 50 °C; $t_{22} = 6.831$, $P < 0.0001$ in 52 °C in **g**. $t_{22} = 2.373$, $P = 0.0268$ in 50 °C; $t_{22} = 2.245$, $P = 0.0352$ in 52 °C; $t_{22} = 2.396$, $P = 0.0255$ in 55 °C in **h**. **k**, **l** Spontaneous pain: total duration (**k**) or number (**l**) of nocifensive behaviour (paw licking or flinching within 2 min) in response to intraplantar injection of vehicle, PS (2.5 nmol/paw) or CIM0216 (2.5 nmol/paw) into control and ATF4-overexpressing mice. $n = 10$ mice per group. $t_{18} = 2.411$, $P = 0.0268$ in PS; $t_{18} = 2.795$, $P = 0.0120$ in CIM0216 in **k**. $t_{18} = 2.775$, $P = 0.0125$ in PS; $t_{18} = 2.334$, $P = 0.0314$ in CIM0216 in **l**. **m** Spontaneous pain: total duration of nocifensive behaviour (in 5 min) in response to intrathecal injection of vehicle, PS (1.25 nmol/paw) or CIM0216 (1.25 nmol/paw) into control and ATF4-overexpressing mice. $n = 10$ mice per group. $t_{18} = 2.506$, $P = 0.0220$ in PS; $t_{18} = 2.46$, $P = 0.0242$ in CIM0216. **a–m**, Two-tailed Independent Student's $t$ test, *$P < 0.05$, **$P < 0.01$, n.s. means not significant. The error bars indicate the SEMs.

down the expression of CXCR4 (Supplementary Fig. 9a) or CXCR7 (Supplementary Fig. 9b) in DRG tissues, respectively. The results revealed that CXCR4 siRNA, but not CXCR7 siRNA, abolished CXCL12-induced upregulation of ATF4 in the DRG (Fig. 9d). Taken together, these results demonstrate that CXCL12-dependent ATF4 upregulation occurs via CXCR4 activation.

## Discussion
In the present study, we revealed that ATF4 mediated the cell membrane localisation of TRPM3 in DRG neurons by interacting with the motor protein KIF17 and thus contributed to thermal sensitivity (Fig. 9e). ATF4 plays a critical role in synaptic plasticity and memory formation in the hippocampus[7] and is increased in the DRG following facet joint distraction[35]. Moreover, ATF4 transcriptionally activates the expression of ATF3, which is a

cellular marker of nerve injury[36]. This evidence suggests that some of the effects of ATF4 on thermal nociception could be regulated by the ATF4/ATF3 axis, which needs further study. Our data showed that ATF4 knockdown increased the withdrawal latency in response to heat to a similar extent under baseline and SNL or CFA treatment conditions. This may suggest that the effects of knocking down ATF4 on the pain models (SNL or CFA) were not due to alterations in heat hyperalgesia but rather to its effects on the basal heat threshold.

TRPM3, a nonselective cation channel, has been recognised as a thermosensor in somatosensory neurons that detect noxious heat stimuli[10], as TRPM3 knockout mice exhibit deficits in their avoidance response to painful heat but not to a noxious cold stimulus[10]. In addition, TRPM3-deficient mice show significant alleviation of heat hyperalgesia induced by inflammation[10]. Flavanones, such as hesperetin, isosakuranetin and the

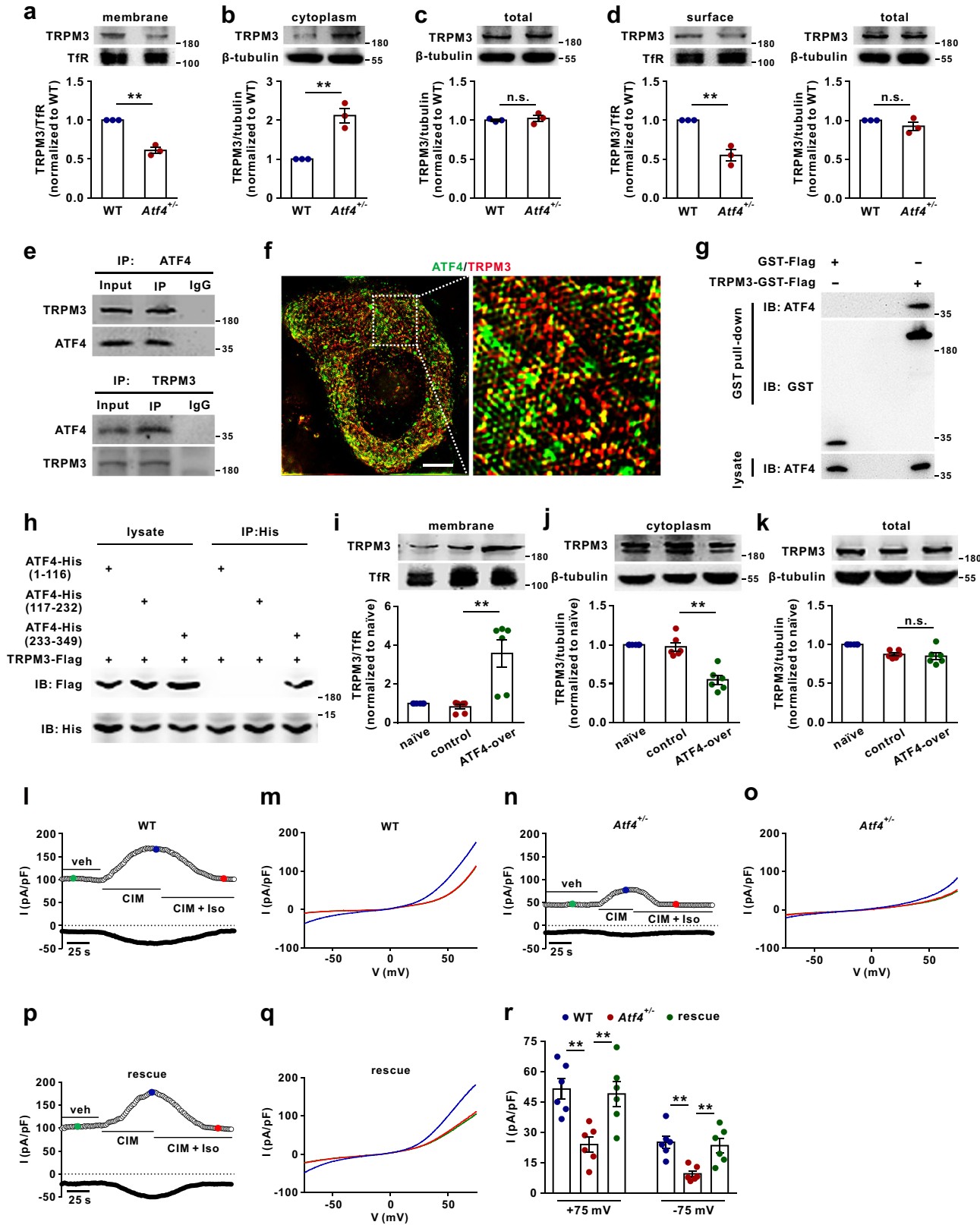

anticonvulsant primidone, are well recognised to reduce the TRPM3-mediated increase in intracellular Ca$^{2+}$ concentration and have been shown to decrease sensitivity to noxious thermal and PS-induced pain behaviours[21,37]. Our findings showed that loss of ATF4 significantly decreased the membrane expression of TRPM3 without changing its total expression in sensory

neurons, and this effect can be reversed by overexpression of a transcriptionally inactive form of ATF4. Knockdown of ATF4 did not affect the promoter activity of *Trpm3*. This is a surprising finding and indicate that ATF4, a transcription factor, regulates the TRPM3 not via gene expression regulation, but by directly influencing TRPM3 trafficking. This phenomenon may

**Fig. 5 ATF4 mediates TRPM3 trafficking in DRG neurons. a–c** TRPM3 expression in the DRG membrane fraction (**a**), cytoplasmic fraction (**b**) and total lysate (**c**) from WT and $Atf4^{+/-}$ mice. $n = 3$ mice per group. $t_4 = 10.3$, $P = 0.0005$ in **a**. $t_4 = 6.043$, $P = 0.0038$ in **b**. **d** TRPM3 surface levels were measured in cultured DRG neurons prepared from WT and $Atf4^{+/-}$ mice using a surface biotinylation assay. $n = 3$ cultures per group. $t_4 = 6.28$, $P = 0.0033$. **e** Co-IP showing the ATF4/TRPM3 interaction in DRGs. DRG lysates were immunoprecipitated with an ATF4 (top) or TRPM3 (bottom) antibody and immunoblotted with a TRPM3 or ATF4 antibody as indicated. This experiment was repeated three times. **f** The SIM images show that the colocalization between ATF4 and TRPM3 in DRG neurons. Scale bar, 5 μm. **g** The GST pull-down assay with two purified proteins, TRPM3-GST-Flag and ATF4, showed a direct interaction between ATF4 and TRPM3. This experiment was repeated three times. **h** Interaction between TRPM3 and ATF4 mutants. The ATF4 mutants was transiently co-expressed with TRPM3, and the cell lysates were immunoprecipitated with a His antibody and then immunoblotted with a His or Flag antibody as indicated. This experiment was repeated three times. **i–k** TRPM3 expression in the DRG membrane fraction (**i**), cytoplasmic fraction (**j**) and total lysate (**k**) after ATF4 overexpression. $n = 6$ mice per group. $F_{(2,15)} = 14.31$, $P = 0.0007$ in **i**. $F_{(2,15)} = 30.15$, $P < 0.0001$ in **j**. **l, n, p** Time course of a whole-cell patch-clamp recording showing the effect of CIM0216 (3 μM) and isosakuranetin (Iso; 10 μM) on the TRPM3 current in DRG neurons from WT (**l**), $Atf4^{+/-}$ (**n**) and rescued (**p**) mice. The open circle indicates the outward current recorded at + 75 mV, and the closed circle indicates the inward current recorded at −75 mV. **m, o, q** I–V relationship of the TRPM3 current at the time points indicated in **l**, **n** and **p**. **r** Histogram of the TRPM3 current density (outward and inward, current before CIM0216 was subtracted) of DRG neurons from WT, $Atf4^{+/-}$ and ATF4 rescued mice after CIM0216 treatment. $n = 6$ neurons per group. $F_{(2,15)} = 8.735$, $P = 0.0049$ in WT vs. $Atf4^{+/-}$, $P = 0.0097$ in $Atf4^{+/-}$ vs. rescue in +75 mV; $F_{(2,15)} = 9.158$, $P = 0.0038$ in WT vs. $Atf4^{+/-}$, $P = 0.0089$ in $Atf4^{+/-}$ vs. rescue in −75 mV. **a–d** Two-tailed Independent Student's t test. **i**, **j**, **k**, **r**, One-way ANOVA followed by Tukey's multiple comparisons test, *$P < 0.05$, **$P < 0.01$, n.s. means not significant. The error bars indicate the SEMs.

represent a function of transcription factors in addition to transcription but requires further study. At present, the study of the non-transcriptional function of transcription factors is an important field. It is well known that nuclear factor-kappaB (NF-κB) regulates neuronal structures and functions transcriptionally[38,39]. Our recent study showed that NF-κB interacts with Nav1.7 in DRG neurons to regulate neuropathic pain in a non-transcriptional manner[40]. Although ATF4 is a transcription factor, it also has functions other than transcription. Studies have shown that ATF4 directly interacts with GABA_B receptors in the soma and at the dendritic membrane surface of both cultured hippocampal neurons and retinal amacrine cells[41–43]. Research has also shown that knocking down ATF4 significantly decreases the membrane expression of GABA_B receptors but does not change the total abundance of GABA_B receptors in hippocampal neurons, indicating that ATF4 regulates GABA_B receptor trafficking[44]. Therefore, despite being a transcription factor, ATF4 may also be involved in non-transcriptional regulation. Our data showed that ATF4, a transcription factor, was mainly expressed in the cytoplasm and at low levels in the nuclei of mouse DRG neurons, which was similar to previous studies[35]. A research report also showed that ATF4 is mainly distributed in the cytoplasm and rarely found in the nuclei of retinal amacrine neurons[43]. These data suggest that ATF4, as a transcription factor, is weakly expressed in the nuclei of neurons under normal conditions. Many factors can activate ATF4 and promote its translocation to the nucleus, thus participating in the regulation of various biological functions. Activation of the TLR4-MyD88 signalling pathway can promote the translocation of ATF4 into the nucleus, where it regulates the secretion of various cytokines and participates in the innate immune response[45]. Activation of GABA_B receptors leads to marked translocation of ATF4 from the cytoplasm into the nucleus and accumulation of ATF4 in the nucleus in primary rat cortical cultures[46]. NMDA-induced long-term depression (LTD) also increases the translocation of ATF4 into the nucleus in rodent hippocampal neurons[47]. However, the factors that induce ATF4 translocation to the nucleus in DRG neurons are not clear and require further study. We know that changes in ion channel current depend on two factors: the membrane abundance of the channel and the characteristics of the individual channel. Our data showed that loss of ATF4 significantly decreased the membrane expression of TRPM3 and reduced the TRPM3 current. These results suggested that the effect of ATF4 on TRPM3 membrane expression may be one of the mechanisms by which ATF4 regulates the TRPM3 current. However,

whether ATF4 also regulates the TRPM3 current by altering the characteristics of individual TRPM3 channel is unclear and requires further study. Our results also revealed that heat stimulation promoted the ATF4/TRPM3 interaction and stabilised TRPM3 in the membrane of DRG neurons, contributing to thermal hypersensitivity. After noxious heat stimulation, the thermal withdrawal thresholds of mice significantly decreased within a few minutes, and this effect was blocked by ATF4 deficiency. This may be a potential protective mechanism of the body against external noxious stimuli. Considering that the skin nerve endings are several centimetres away from DRG cell bodies, how the heat stimulation signals generated by nerve endings affect molecular interactions in the cell body is unclear and needs further study. Our data showed that, 30 min after ISRIB was intrathecally administered, it may have inhibited the function of cytoplasmic ATF4 without affecting the nuclear transcription function of ATF4. Behavioural data showed that intrathecal injection of ISRIB for 30 min abolished the decrease of threshold in mice after paws heat stimulation, showed that only inhibiting the function of ATF4 in cytoplasm but not in nuclei can also block the decrease of threshold in mice after paws heat stimulation. Thus, our evidence further indicated that the effect of interfering with ATF4 on the thermal stimulation threshold is mainly derived from the non-transcriptional function of ATF4. A previous study showed that TRPM3 is expressed in the spinal cord[48]; thus, it should be investigated whether intrathecal injection of CIM0216 or PS (agonists of TRPM3) regulates pain behaviour by affecting the spinal dorsal horn.

The trafficking of thermo TRP channels towards the cell membrane is an essential step in their ability to mediate neuronal sensitivity to heat[12]. We further explored the critical role and the underlying mechanism of motor proteins of the kinesin superfamily (KIFs), i.e., KIF17, in the process of ATF4-dependent TRPM3 membrane trafficking in sensory neurons. Previous evidence has demonstrated that various subtypes of KIFs exhibit differential functions to form intracellular cargo apparatuses and regulate protein trafficking[13]. For example, upon binding with its target protein, KIF13B promotes the membrane translocation of vascular endothelial growth factor receptor 2 to the surface of endothelial cells[49]. Kinesin KIF13 also serves as a key regulator of anterograde glutamate receptor trafficking in the ventral nerve cord of *Caenorhabditis elegans*[50]. It has been reported that via direct binding with the N-terminal T1 domain, KIF5 transports the voltage-gated potassium channel Kv3.1 through the axon initial segment[51]. Similarly, KIF3A, a component of kinesin 2, is involved

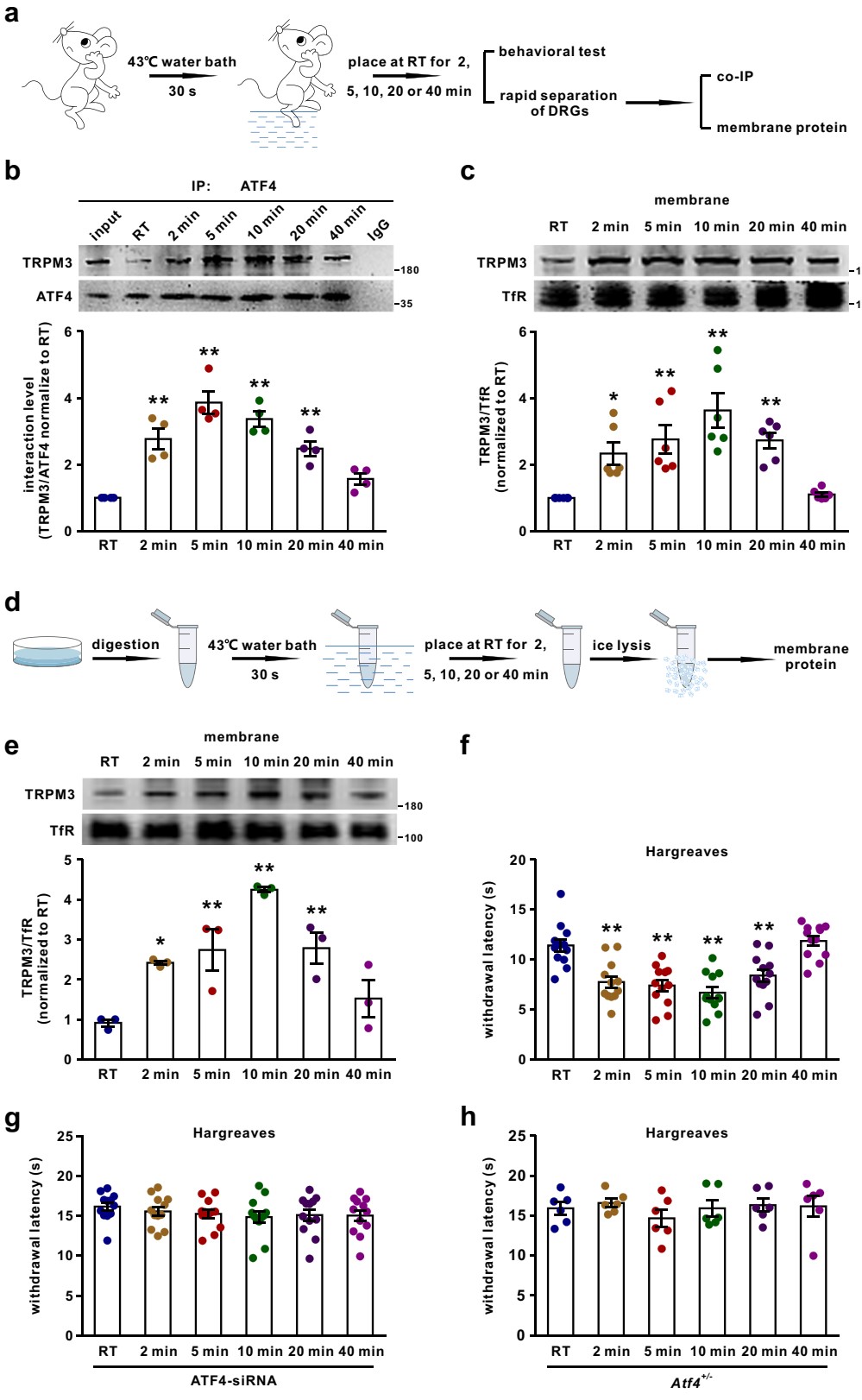

in the polarised trafficking of the Kv1.2 channel to axons[52,53]. Our previous study showed that KIF3A regulates Nav1.6 trafficking in DRG neurons and is thus involved in chronic pain[16]. Here, we found that ATF4 interacted with KIF17 and that knockdown or overexpression of KIF17 decreased or increased membrane TRPM3 levels, respectively, in sensory neurons. Interestingly, the increase in TRPM3 membrane expression induced by ATF4 overexpression was reversed by KIF17 siRNA in DRG neurons. These results suggested that the motor protein KIF17 plays a critical role in ATF4-dependent TRPM3 trafficking in DRG neurons.

The chemokine CXCL12 plays a critical role in regulating neural activity. For example, in the dorsal raphe nucleus and

**Fig. 6 Heat stimulation stabilises TRPM3 in the DRG membrane. a** The diagram shows the experimental procedure. The hindpaws of mice were soaked in a 43 °C water bath for 30 s, and the mice were placed at RT for 2, 5, 10, 20 or 40 min. Then, behavioural, co-IP and membrane protein immunoblotting experiments were performed. **b, c** ATF4/TRPM3 interactions (**b**) and TRPM3 membrane expression (**c**) in mice were evaluated at different time points after heat stimulation. The experiment was repeated four times in **b**. $n = 6$ mice per group in **c**. $F_{(5,18)} = 20.17$, $P = 0.0007$ in 2 min, $P < 0.0001$ in 5 min, $P < 0.0001$ in 10 min, $P = 0.0044$ in 20 min in **b**. $F_{(5,30)} = 10.59$, $P = 0.0277$ in 2 min, $P = 0.0022$ in 5 min, $P < 0.0001$ in 10 min, $P = 0.0027$ in 20 min in **c**. *$P < 0.05$, **$P < 0.01$ versus the RT group. **d** The diagram shows the experimental procedure. Isolated DRG neurons were placed in a 43 °C water bath for 30 s and placed at RT for 2, 5, 10, 20 or 40 min. The neurons were then lysed to detect the membrane abundance of TRPM3. **e** TRPM3 membrane expression in isolated DRG neurons was evaluated at different time points after direct heat stimulation. $n = 3$. $F_{(5,12)} = 12.3$, $P = 0.0358$ in 2 min, $P = 0.0100$ in 5 min, $P < 0.0001$ in 10 min, $P = 0.0085$ in 20 min. *$P < 0.05$, **$P < 0.01$ versus the RT group. **f–h** Naïve (**f**), ATF4 siRNA-injected (**g**) and $Atf4^{+/-}$ (**h**) mice were subjected to the Hargreaves test at different time points after heat stimulation. $n = 12$ mice per group in **f, g**. $n = 6$ mice per group in **h**. $F_{(5,66)} = 14.36$, $P = 0.0004$ in 2 min, $P < 0.0001$ in 5 min, $P < 0.0001$ in 10 min, $P = 0.0057$ in 20 min. **$P < 0.01$ versus RT group. One-way ANOVA followed by Tukey's multiple comparisons test. The error bars indicate the SEMs.

nigrostriatum, CXCL12 increases the excitability of dopaminergic (DAergic) neurons and serotonergic (5-HTergic) neurons[54,55]. Our studies and those of our peers have shown that CXCL12 expression is markedly increased in the DRG after neuropathic pain induced by nerve injury or a chemotherapeutic agent and that inhibiting CXCR4, a CXCL12 receptor, attenuates these abnormal pain behaviours[31–33]. In addition, another study suggested that CXCL12 synthesised in the somata of DRG neurons is transported to the dorsal horn to modulate nociceptive signalling[56]. All of these studies indicate that CXCL12 participates in neuropathic pain. Our results showed that CXCL12 application increased the presentation of ATF4 in DRG neurons in vivo and in vitro. Interestingly, knocking down ATF4 in the DRG attenuated the heat hyperalgesia induced by intrathecally delivered CXCL12. CXCR4 and CXCR7 are specific CXCL12 receptors that participate in CXCL12-regulated physiological and pathological processes[34]. We found that knocking down CXCR4, but not CXCR7, inhibited the increase in ATF4 induced by CXCL12 in the DRG. Taken together, these data suggested that CXCL12-dependent ATF4 upregulation occurs by activating CXCR4.

In summary, we revealed a previously unknown role for ATF4 in nociception. We also illustrated the underlying mechanism of TRPM3 channel membrane trafficking in sensory neurons. This information can be applied to develop further strategies to treat conditions related to heat sensation, most notably burn pain, and the parameters mentioned in this study should be measured when ATF4 modulators are used in clinical settings.

## Methods

**Animals**. C57BL/6 mice were obtained from the Institute of Experimental Animals of Sun Yat-sen University. $Atf4^{+/-}$ and $Trpm3^{-/-}$ mice on the C57BL/6 background were purchased from Cyagen Biosciences Inc. All animals were housed in separate cages in a temperature-controlled ($24 \pm 1$ °C) and humidity-controlled (50-60%) room under a 12/12-h light/dark cycle. The mice had ad libitum access to sterile water and standard laboratory chow. All animal experimental procedures were approved by Research Ethics Committee of Guangdong Provincial People's Hospital, Guangdong Academy of Medical Sciences and were carried out in accordance with the guidelines of the National Institutes of Health on animal care and ethics[57]. All animals were assigned randomly to different experimental or control groups.

**Animal pain models and intrathecal injection**. To produce inflammatory pain, CFA (20 μl) was injected into the plantar surface of the hindpaw. To produce neuropathic pain, the mice were anaesthetised, and the left L5 spinal nerve was isolated adjacent to the vertebral column and tightly ligated with 6-0 silk sutures distal to the DRG and proximal to the formation of the sciatic nerve. The L5 spinal nerves of sham-operated mice were identically exposed but not ligated. CIM0216 (Tocris Bioscience, catalogue no.: 5521), PS (R&D, catalogue no.: 5376) or capsaicin (Tocris Bioscience, catalogue no.: 0462) was injected intraplantarly (2.5 nmol/20 μl/paw, 2.5 nmol/20 μl/paw, or 1 nmol20 μl/paw, respectively), or intrathecally (1.25 nmol/10 μl/mouse) to induce spontaneous pain. Intrathecal administration was performed by a polyethylene-10 catheter that was inserted into the subarachnoid space of the mouse through the L5 to L6 intervertebral space so that the tip of the catheter was located at the L5 level, and 10 μl reagent was delivered into the cerebrospinal fluid.

**Behavioural tests**. The animals were habituated to the environment for at least 2 days before testing. All the behaviours tests were performed in a blinded manner.

**Von Frey test**. The mice were placed in plastic chambers on a mesh floor. Von Frey filaments with increasing grades of force were applied to the hindpaws of the mice. Each filament was applied 5 times during the test. The lowest filament force that elicited paw withdrawal more than 3 times during the test was defined as the mechanical threshold.

**Dynamic mechanical test**. Mice were acclimated to von Frey chambers for 1 h. The lateral side of the hindpaw was gently stroked with a 5/0 brush from heel to toe. Responses were scored as follows: 0 = no response; 1 = very short, fast movement/lifting of the paw; 2 = sustained lifting of the paw for more than 2 s towards the body or strong lateral lifting above the body level; and 3 = flinching, licking, or flicking of the affected paw. The average score of three trials per mouse was reported as the allodynia score.

**Tape response test**. The mice were conditioned to a round plexiglass container for at least 5 min. A 1-inch piece of laboratory tape was gently applied to the bottom centre of the mouse's back. The mice were observed for 5 min, and the total number of responses to the tape was recorded. Biting or grabbing the tape or making an obvious "wet dog shake" movement to remove the tape from the back was considered a response.

**Tail clip test**. A small alligator clip (force, 700 g) was applied 1 cm from the base of the tail. The latency to attack/bite the clip was measured with a stopwatch. Upon attack, the clip was removed, and the animals were returned to their cages.

**Pinprick test**. The mice were acclimated to a von Frey chamber for 30 min, and 27 needles were applied to the glabrous skin of the hindpaw, taking care not to pierce the skin. Each mouse was tested 10 times at an interval of 1 min. Paw withdrawal, shaking, or licking was scored as a response, and the percentage of responses to the total number of tests was recorded.

**Tail-flick test**. The mice were gently restrained inside a cloth/cardboard pocket with their tails outside the pocket. The distal half of the tail was immersed in a 48 °C, 50 °C or 52 °C water bath, and the latency to vigorous withdrawal of the tail from the water was measured.

**Hargreaves test**. Thermal hypersensitivity was measured using a plantar test according to the method described by Hargreaves[58]. Briefly, a radiant heat source beneath a glass floor was aimed at the fat pad on the plantar surface of the hindpaw. The latency to withdraw the hindpaw was measured. The hind paw of each mouse was soaked in a 43 °C water bath for 30 s, and then the mice were immediately placed in a plastic chamber. Two, 5, 10, 20 or 40 min later, the behavioural experiment was performed. The mice were tested individually.

**Hot plate test**. The mice were placed into a clear Plexiglas cylinder on top of a metal surface maintained at 50 °C, 52 °C or 55 °C. The latency to lick or shake either hindpaw was measured.

**Evaporative cooling test**. The mice were habituated to a plastic chamber on a mesh floor. A drop (10-20 μl) of acetone was applied to the hindpaws of the mice. The duration of flinching or licking behaviours within 1 min was measured.

**Rotarod test**. The mice were tested on a rotarod with a velocity that increased from 4 rpm to 40 rpm within 5 min. The mice were pre-trained for 2 days for adaptation. Then, the amount of time that each mouse spent on the rotarod before it fell off was recorded.

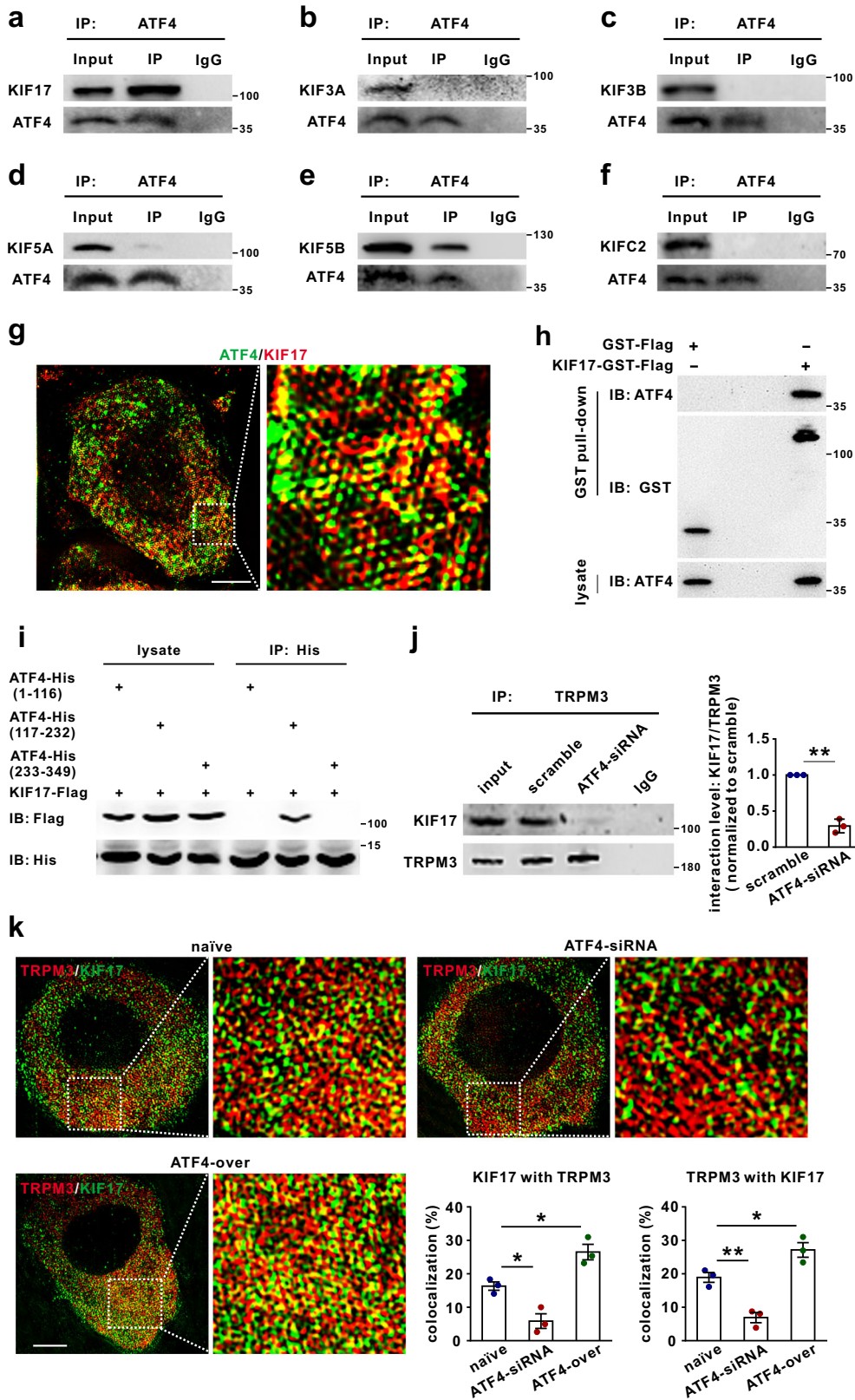

**Spontaneous pain test.** Spontaneous pain induced by intraplantar injection of CIM0216, PS or capsaicin was measured by counting the time and number of the mouse licked the affected paw or flinched within 2 min, spontaneous pain induced by intrathecal injection of CIM0216 or PS was assessed by counting the amount of time each mouse spent licking, flicking, biting, or flinching within 5 min after injection.

**Cell culture and transfection.** Mouse DRGs were freed from their connective tissue sheaths and broken into pieces with a pair of sclerotic scissors in DMEM/F12 (Gibco) at a low temperature. Following enzymatic and mechanical dissociation, the DRG neurons were plated on glass coverslips coated with poly-L-lysine (Sigma) in a humidified atmosphere (5% $CO_2$, 37 °C). The cells were used for electro-physiological recordings approximately 4 h to 24 h after plating. HEK293T cells

**Fig. 7 ATF4 interacted with KIF17 in the DRG tissues of mice. a–f** Co-IP showing the ATF4/KIF interaction in the DRG. DRG lysates were immunoprecipitated with an ATF4 antibody and immunoblotted with a KIF17, KIF3A, KIF3B, KIF5A, KIF5B, KIFC2 or ATF4 antibody as indicated. This experiment was repeated three times. **g** SIM images show that the colocalization between ATF4 and KIF17 in DRG neurons. Scale bar, 5 μm. **h** The GST pull-down assay with two purified proteins, KIF17-GST-Flag and ATF4, showed a direct interaction between ATF4 and KIF17. This experiment was repeated three times. **i** Interaction between KIF17 and ATF4 mutants. The ATF4 mutants were transiently co-expressed with KIF17, and the cell lysates were immunoprecipitated with a His antibody and then immunoblotted with a His or Flag antibody as indicated. This experiment was repeated three times. **j** The interaction level between TRPM3 and KIF17 was examined by co-IP in scrambled and ATF4 siRNA-treated mice DRG lysates. DRG lysates were immunoprecipitated with a TRPM3 antibody and immunoblotted with a KIF17 or TRPM3 antibody as indicated. This experiment was repeated three times. $t_4 = 12.88$, $P = 0.0002$. **k** SIM images showed the colocalization between TRPM3 and KIF17 in DRG neurons from naïve, ATF4 siRNA and ATF4-overexpresing mice. Quantification data showed the colocalization rates of KIF17 with TRPM3 (colocalized yellow spots/total KIF17 positive spots) and those of TRPM3 with KIF17 (colocalized yellow spots/total TRPM3 positive spots) in DRG neurons. $n = 3$ mice per group. $F_{(2,6)} = 27.17$, $P = 0.0224$ in naïve vs. ATF4-siRNA, $P = 0.0255$ in naïve vs. ATF4-over in KIF17 with TRPM3. $F_{(2,6)} = 33.5$, $P = 0.0069$ in naïve vs. ATF4-siRNA, $P = 0.0375$ in naïve vs. ATF4-over in TRPM3 with KIF17. Scale bar, 10 μm. **j** Two-tailed Independent Student's t test. **k** One-way ANOVA followed by Tukey's multiple comparisons test, *$P < 0.05$, **$P < 0.01$. The error bars indicate the SEMs.

were cultured in MEM with 10% foetal bovine serum. Full-length TRPM3-Flag, ATF4-His and KIF17-Flag plasmids were obtained from Synbio Technologies, and ATF4-His (aa 1–116, 117–232, 233–349) plasmids were modified from the full-length ATF4 plasmid. The Flag and His fusion proteins were co-expressed in HEK293T cells. The cells were transfected with 1–2 μg plasmid per 35-mm dish or 2–3 μg plasmid per 60-mm dish using Lipofectamine 2000 reagent (Invitrogen). The cells were used for subsequent experiments 24–48 h after transfection.

**Electrophysiological recordings.** Whole-cell patch-clamp recordings were performed with an EPC-10 amplifier and PULSE software (HEKA Electronics, Lambrecht) at RT[22,59]. Currents were recorded with glass pipettes (1–3 MΩ resistance) fabricated from borosilicate glass capillaries using a Sutter P-97 puller (Sutter Instruments, Novato, CA). Voltage errors were minimised using 80-90% series resistance compensation. TRPM3 currents in DRG neurons were recorded with a ramp potential of −75 mV to +75 mV. The voltage for current analysis was ±75 mV. Neurons with a leak current of >500 pA or a series resistance of >10 MOhm were excluded. The extracellular solution contained (in mM) 140 NaCl, 3.5 KCl, 1 MgCl$_2$, 1.25 NaH$_2$PO$_4$, 10 HEPES, and 10 D-glucose (adjusted to pH 7.4 with NaOH). The pipette solution contained (in mM) 140 CsCl, 1 CaCl$_2$, 2 MgATP, 2 Na$_2$ATP, 10 EGTA, and 5 TEA-Cl (adjusted to pH 7.2 with CsOH). The osmolality of all solutions was adjusted to 310 mOsm.

**Surface protein biotinylation.** Plasma membrane protein expression was detected by cell surface biotinylation using a Cell Surface Protein Isolation Kit (Pierce, catalogue no.: 89881) according to the manufacturer's instructions. Briefly, cells were washed with PBS and biotinylated with Sulfo-NHS-SS-Biotin in PBS for 30 min at 4 °C. After quenching, the cells were lysed, and labelled proteins were isolated by incubation with NeutrAvidin Agarose beads for 60 min at RT. After washing, the proteins were eluted by heating the beads for 5 min at 95 °C and prepared for immunoblotting.

**Extraction of plasma membrane proteins.** For membrane protein preparation, samples were homogenised on ice with a plasma membrane protein extraction kit (Invent Biotechnologies, catalogue no.: SM-005). The detailed protocol was as follows: DRG tissues or cultured DRG neurons were lysed with 200–500 μl buffer A. The filter cartridge was capped and centrifuged at 14,000 r.p.m. (16,000 g) for 30 s. The filter was discarded, and the pellet was resuspended by vortexing for 10 s and centrifuged at 3000 r.p.m. (700 g) for one min (the pellet contained the intact nuclei). The supernatant was transferred to a new tube and centrifuged at 4 °C for 10–30 min at 16,000 g. The supernatant (the cytosolic fraction) was removed, and the pellet (the total membrane protein fraction including organelles and plasma membranes) was saved. The total membrane protein fraction was resuspended in 200 μl buffer B by vortexing and centrifuged at 10,000 r.p.m. (7800 g) for 5 min at 4 °C. The pellet contained the organelle membrane proteins (in our study, cytoplasmic protein comprised the cytosolic fraction and organelle membrane fraction). The supernatant was carefully transferred to a fresh 2.0-ml microcentrifuge tube, and 1.6 ml cold PBS was added. The sample was mixed a few times by inverting and centrifuged at 14,000 r.p.m. (16,000 g) for 15–30 min. The supernatant was discarded, and the pellet (isolated plasma membrane proteins) was saved. Protein samples of different fractions were denatured and prepared for immunoblotting.

**Preparation of the GST-fused proteins.** GST-Fused Proteins were expressed in *E. coli* BL21. The bacteria grew in 2 × YTA media and the expression of proteins were induced by 1 mM Isopropyl β-D-thiogalactopyranoside (IPTG). The bacteria were then centrifuged, resuspended and ultrasonic treated to release protein. The proteins were purified by Glutathione-Sepharose beads (GE Healthcare), concentrated and quantified before use.

**Co-immunoprecipitation and GST pull-down.** Transfected HEK293T cells or DRG tissues were lysed in cold co-IP RIPA buffer [20 mM Tris-HCl (pH 7.5), 150 mM NaCl, 0.1% Triton X-100, 1% sodium deoxycholate, 10 mM NaF, 1 mM EDTA, 1 mM PMSF, and 1 mg/ml leupeptin]. The lysates were centrifuged, and 5% of each supernatant was taken for the input sample. The remaining supernatants were incubated with 5–10 μg ATF4, TRPM3 or His antibody at 4 °C overnight and then with protein A/G beads (GE Healthcare) at 4 °C for 4 h. The immunoprecipitated samples were denatured and prepared for immunoblotting. Immunoprecipitation was performed with antibodies against ATF4, TRPM3, KIF17, KIF3A, KIF3B, KIF5A, KIF5B, KIFC2, Flag and His. To evaluate the direct interaction between ATF4 and TRPM3 or KIF17, 5 μg purified ATF4 protein (Proteintech, Ag16665) was incubated with 5 μg TRPM3-GST-Flag or KIF17-GST-Flag purified protein in 400 μl RIPA buffer. The proteins were incubated with the cell lysate at 4 °C overnight. Then, the GST-fused protein was precipitated with 10 μl of Glutathione-Sepharose beads at 4 °C for 2 h. The precipitant was washed, denatured and prepared for immunoblotting.

**Western blotting.** Spinal dorsal horn, dorsal root, DRG, sciatic nerve and sural nerve tissues or cultured DRG neurons were lysed and homogenised in cold RIPA buffer [50 mM Tris-HCl (pH 7.4), 150 mM NaCl, 0.1% Triton X-100, 1% sodium deoxycholate, 0.1% SDS, 10 mM NaF, 1 mM EDTA, 1 mM PMSF, and 1 mg/ml leupeptin]. The protein samples were separated via gel electrophoresis (SDS-PAGE) and transferred onto PVDF membranes. The membranes were placed in blocking buffer for 1 h at RT and incubated in primary antibodies against ATF4 (goat, 1:1000, GeneTex, catalogue no.: GTX89973), TRPM3 (rabbit, 1:1000, Bioss, catalogue no.: bs-9046R; rabbit, 1:200, Alomone Labs, catalogue no.: ACC-050), TRPV1 (rabbit, 1:200, Alomone Labs, catalogue no.: ACC-030), TRPA1 (rabbit, 1:1000, ABclonal, catalogue no.: A12544), KIF17 (mouse, 1:100, Santa Cruz Biotechnology, catalogue no.: sc-137040), KIF3A (goat, 1:100, Santa Cruz Biotechnology, catalogue no.: sc-18745), KIF3B (rabbit, 1:100, Santa Cruz Biotechnology, catalogue no.: sc-50456), KIF5A (rabbit, 1:1000, Abcam, catalogue no.: ab5628), KIF5B (rabbit, 1:2000, Abcam, catalogue no.: ab167429), KIFC2 (rabbit, 1:100, Abcam, catalogue no.: ab3476), TfR (mouse, 1:1000, Thermo Fisher Scientific, catalogue no.: 13-6800), CXCR4 (rabbit, 1:1000, Abcam, catalogue no.: ab124824), CXCR7 (rabbit, 1:1000, Abcam, catalogue no.: ab72100), β-tubulin (mouse, 1:2000, Arigo, catalogue no.: ARG62347), Flag (rabbit, 1:1000, Cell Signaling Technology, catalogue no.: 14793) or His (rabbit, 1:1000, Cell Signaling Technology, catalogue no.: 12698) overnight at 4 °C. Next, the membranes were incubated with an HRP-conjugated secondary antibody (1:10,000). Enhanced chemiluminescence (ECL) solution (Millipore) was used to detect the immunocomplexes. Each band was quantified with computer-assisted imaging analysis software (Tanon Gis).

**In situ hybridisation.** To label KIF17, ATF4 and TRPM3, a *KIF17* mRNA probe labelled with CY5, an *ATF4* mRNA probe labelled with FITC, and a *TRPM3* mRNA probe labelled with CY3 were constructed. The probes were designed by BersinBio, and the assay was conducted according to the manufacturer's instructions[60,61]. Paraffin sections were dewaxed in xylene, dehydrated in 70%, 80%, 95% and 100% ethanol for 5 min each. And digested in proteinase K digested at 55 °C for 10 min. The sections were soaked in preheated (78 °C) denaturing solution and denatured for 8 min. Fluorescent dye-labelled probe and probe diluent were mixed to produce a probe hybridisation mixture and denatured at 78 °C for 8 min. A total of 20–40 μl hybridisation reaction solution was dripped onto each sample, and the samples were co-denatured at 73 °C for 5–8 min and then hybridised overnight (16 h–20 h) at 53 °C. The sections were washed and photographed with a laser confocal microscope.

**Luciferase assay.** Twenty thousand BV-2 cells were seeded in triplicate in 48-well plates and allowed to settle for 24 h. One hundred nanograms control luciferase plasmid or luciferase reporter plasmid containing a fragment (−2052/+558) of

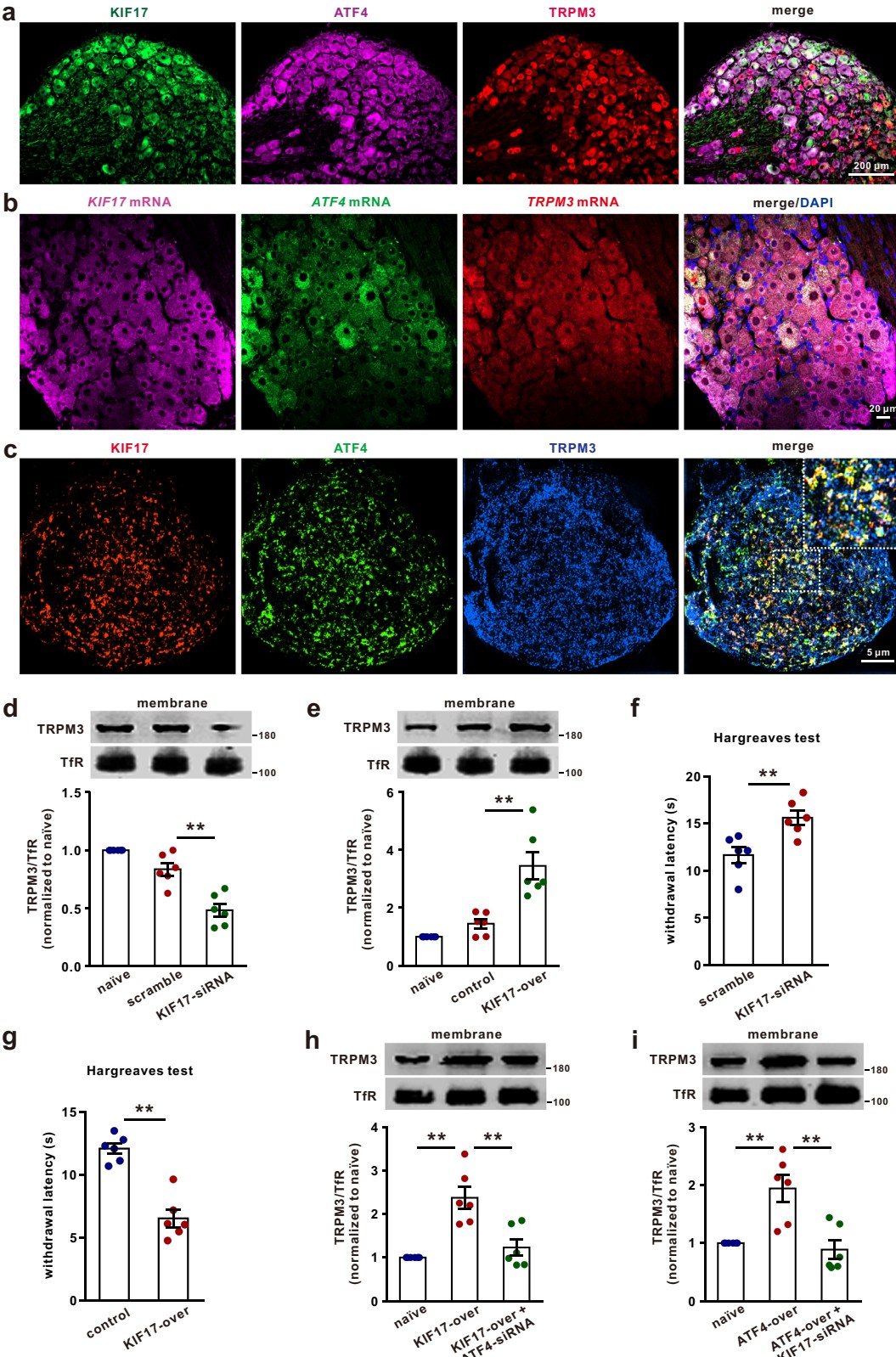

*Trpm3* promoter plus 1 ng pRL-TK Renilla plasmid (Promega) was transfected into the cells using Lipofectamine 2000 reagent (Invitrogen) according to the manufacturer's recommendations. Luciferase and Renilla signals were measured 36 h after transfection using the Dual Luciferase Reporter Assay Kit (Promega) according to a protocol provided by the manufacturer. To observe the effect of ATF4 on *Trpm3* luciferase activity, BV-2 cells were transfected with ATF4 siRNA (30 nM) using Lipofectamine 2000 (Invitrogen) according to the manufacturer's instructions 12 h before the experiments.

**Immunohistochemistry and SIM**. Mice were perfused with 4% paraformaldehyde (PFA). The spinal cord, DRG, skin and sciatic nerve were dissected and post-fixed in 4% PFA for 1 h. Next, the tissues were dehydrated in 30% sucrose and embedded for cryostat sectioning. The cryostat sections were incubated with primary antibodies against ATF4 (rabbit, 1:200, Abcam, catalogue no.: ab23760; rabbit, 1:100, Cell Signaling Technology, catalogue no.: 11815 S; goat, 1:100, GeneTex, catalogue no.: GTX89973), TRPM3 (rabbit, 1:100, Bioss, catalogue no.: bs-9046R; rabbit, 1:100, Alomone Labs, catalogue no.: ACC-050), IB4 (1:50, Sigma, catalogue no.:

**Fig. 8 KIF17 is involved in ATF4-dependent TRPM3 membrane trafficking. a** Colocalization of the KIF17, ATF4 and TRPM3 proteins in mouse DRG sections. Scale bar, 200 μm. **b** Colocalization of *KIF17* mRNA, *ATF4* mRNA and *TRPM3* mRNA in mouse DRG sections. Scale bar, 20 μm. **c** Colocalization of the KIF17, ATF4 and TRPM3 proteins in DRG neurons was detected by SIM. Scale bar, 5 μm. **d, e** Changes in the membrane expression of TRPM3 in the DRG after KIF17 knockdown (**d**) or KIF17 overexpression (**e**). $n = 6$ mice per group. $F_{(2,15)} = 34.36$, $P = 0.0002$ in **d**. $F_{(2,15)} = 20.48$, $P = 0.0005$ in **e. f** The behaviours of KIF17 siRNA- and scrambled siRNA-injected mice were evaluated by the Hargreaves test. $n = 6$ mice per group. $t_{10} = 3.485$, $P = 0.0059$. **g** The behaviours of KIF17-overexpressing and control mice were evaluated by the Hargreaves test. $n = 6$ mice per group. $t_{10} = 6.776$, $P < 0.0001$. **h** ATF4 knockdown suppressed the increased expression of TRPM3 in the membrane induced by KIF17 overexpression. $n = 6$ mice per group. $F_{(2,15)} = 16.19$, $P = 0.0002$ in naïve vs. KIF17-over, $P = 0.0014$ in KIF17-over vs. KIF17-over + ATF4-siRNA. **i** KIF17 knockdown inhibited the increase in TRPM3 expression on the cell surface induced by ATF4 overexpression. $n = 6$ mice per group. $F_{(2,15)} = 12.76$, $P = 0.0025$ in naïve vs. ATF4-over, $P = 0.0010$ in ATF4-over vs. ATF4-over + KIF17-siRNA. **P < 0.01. **d, e, h, i** One-way ANOVA followed by Tukey's multiple comparisons test. **f, g** Two-tailed Independent Student's $t$ test. The error bars indicate the SEMs.

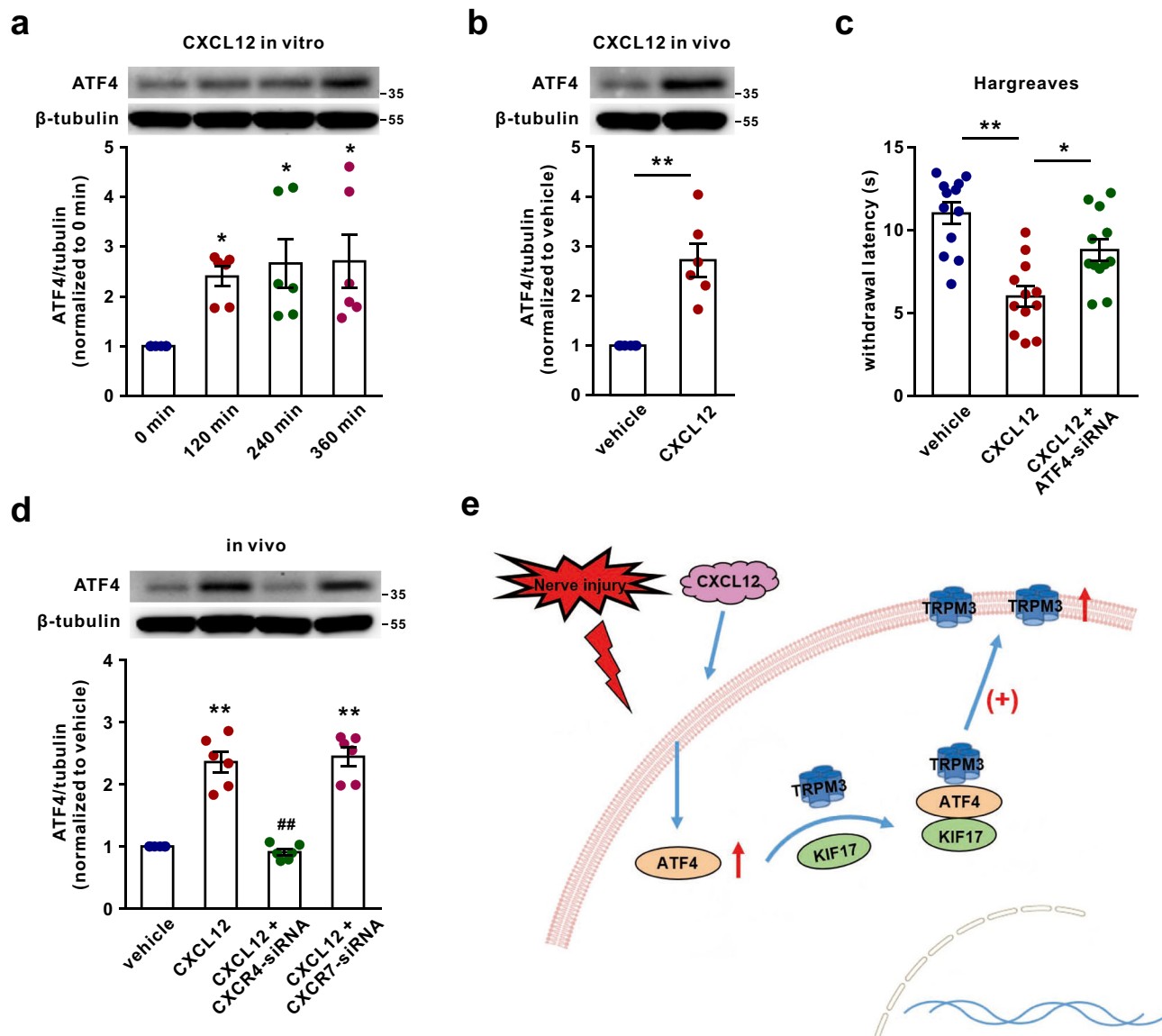

**Fig. 9 CXCL12 mediates ATF4 upregulation via CXCR4. a** Cultured DRG neurons were treated with CXCL12 (1 μg/mL) for 120, 240 or 360 min, and then ATF4 expression was measured. $n = 6$. $F_{(3,20)} = 4.642$, $P = 0.0462$ in 120 min, $P = 0.0154$ in 240 min, $P = 0.0128$ in 360 min. *$P < 0.05$ versus 0 min. **b** CXCL12 (1 μg in 10 μL PBS + 0.5% BSA) was intrathecally injected three times every 3 h, and ATF4 expression was measured in the DRG 2 h after the last injection. $n = 6$ mice per group. $t_{10} = 5.151$, $P = 0.0004$. **P < 0.01. **c** ATF4 siRNA significantly relieved CXCL12-induced thermal hyperalgesia. $n = 12$ mice per group. $F_{(2,33)} = 15.68$, $P < 0.0001$ in vehicle vs. CXCL12, $P = 0.0101$ in CXCL12 vs. CXCL12 + ATF4-siRNA. *$P < 0.05$, **$P < 0.01$. **d** CXCR4 siRNA but not CXCR7 siRNA abolished the increase in ATF4 expression in the DRG induced by CXCL12 in vivo. $n = 6$ mice per group. $F_{(3,20)} = 54.47$, $P < 0.0001$ in CXCL12, CXCL12 + CXCR4-siRNA and CXCL12 + CXCR7-siRNA. **P < 0.01 versus the vehicle group, ##$P < 0.01$ versus the CXCL12 group. **e** Hypothetical model illustrating that ATF4 interacts with TRPM3 and KIF17 to form a complex to regulate the membrane trafficking of TRPM3 in sensory neurons and thus contributes to thermal sensitivity. **a, c, d** One-way ANOVA followed by Tukey's multiple comparisons test. **b** Two-tailed Independent Student's $t$ test. The error bars indicate the SEMs.

L2895), CGRP (mouse, 1:200, Abcam, catalogue no.: ab81887; goat, 1:200, Abcam, catalogue no.: ab36001), NF200 (mouse, 1:200, Sigma, catalogue no.: N0142), NeuN (mouse, 1:200, Millipore, catalogue no.: MAB377; rabbit, 1:200, Abcam, catalogue no.: ab177487) or KIF17 (mouse, 1:50, Santa Cruz Biotechnology, catalogue no.: sc-137040) at 4 °C overnight and then incubated with secondary antibodies (1:400) for 1 h at RT. Three-dimensional super-resolution images were captured using a three-dimensional structured illumination microscope with the N-SIM System and a CFI SR oil immersion objective lens (Apochromat TIRF×100, 1.49 numerical aperture, Nikon, Japan), and images were post-processed with Nikon NIS-Elements software. We used Image-Pro Plus 6.0 software to analyse the colocalization percentage of images. First, we recorded the intensity of colocalization spots and then measured the intensity of single-colour spots. The intensity of the colocalization spots divided by the intensity of the single-colour spots was considered the percentage of colocalization. To confirm the specificity of the ATF4 and TRPM3 antibodies, blocking experiments were conducted in DRG sections using a mixture of anti-ATF4 antibody and immunising blocking peptide (10 times the molar concentration of the antibody, GeneTex, catalogue no.: GTX89973-PEP, sequence: EEVRKARGKKRVP, species: human) or anti-TRPM3 antibody and immunising blocking peptide (10 times the molar concentration of the antibody, Bioss, catalogue no.: bs-9046P, sequence: KLFITDDELKKVH, species: human) based on an immunising peptide blocking protocol (https://www.abcam.com/protocols/blocking-with-immunizing-peptide-protocol-peptide-competition).

**Intrathecal siRNA and AAV injection.** siRNAs targeting mouse ATF4 (FITC-labelled, sense: GCUAGGCAGUGAAGUUGAU), KIF17 (sense: GCCACGCAU UAAUGAAGAC), CXCR4 (sense: GCUAACCCUUAUGCAAAGA), CXCR7 (sense: CCAUGCCUAACAAGAACGU) and a nontargeting siRNA (sense: CCT AAGGUUAAGUCGCCCTCG) were purchased from Thermo Fisher Scientific. AAV overexpression vectors for ATF4 (serotype 5, rAAV-CMV-*Atf4*-2A-EGFP-WPRE-PA), KIF17 (serotype 5, rAAV-CMV-*Kif17*-2A-EGFP-WPRE-PA) and a transcriptionally inactive form of ATF4 (serotype 5, rAAV-CMV-*Atf4*-ΔbZIP-2A-EGFP-WPRE-PA) were purchased from BrainVTA Biotechnology. The transcriptionally inactive form of ATF4 (rAAV-CMV-*Atf4*-ΔbZIP-2A-EGFP-WPRE-PA) was generated by site-directed mutagenesis with 6 amino acid substitutions within the DNA-binding domain ($^{292}$RYRQKKR$^{298}$ to $^{292}$GYLEAAA$^{298}$)[27–29]. A mixture of 10 μg siRNA and 7.5 μg transfection reagent (chimeric rabies virus glycoprotein fragment, RVG-9R, Anaspec, USA)[62,63] in 10 μl 5% dextrose in water (D5W) was injected intrathecally 48 h prior to the experiments. rAAV-CMV-*Atf4*-2A-EGFP-WPRE-PA (10 μl), rAAV-CMV-*Kif17*-2A-EGFP-WPRE-PA (10 μl) and rAAV-CMV-*Atf4*-ΔbZIP-2A-EGFP-WPRE-PA (10 μl) were injected intrathecally in mice and the experiments were performed after 21 days.

**Data analysis.** The grey values of the western bands were quantified by Tanon Gis software. The grey value of the images and colocalization were quantified by Image-Pro Plus 6.0 software. All western blotting, immunostaining, electrophysiological and behavioural data are expressed as the mean ± SEM and were analysed with GraphPad Prism 6. The threshold for statistical significance was $P < 0.05$.

**Reproducibility.** Experiments were repeated independently with similar results at least three times. Micrographic images presented in figures are representative ones from experiments repeated independently: Fig. 1a (three times), Fig. 1c, d (four times), Fig. 5f (three times), Fig. 7g (three times), Fig. 8a–c (three times), Supplementary Fig. 1b, d, h (three times), Supplementary Fig. 1f (four times), Supplementary Fig. 2,a, b (four times), Supplementary Fig. 5l, m (three times), Supplementary Fig. 7b (three times).

**Reporting summary.** Further information on research design is available in the Nature Research Reporting Summary linked to this article.

## Data availability

All data supporting the findings of this study are provided within the paper and its supplementary information. All additional information will be made available upon reasonable request to the authors. Source data are provided with this paper.

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

## Acknowledgements

This study was supported by National Natural Science Foundation of China (82001172 to X.L.Z., 81801112 to M.X.X.), National Key R&D Program of China (2017YFC1103800 to X.Y.C.), Guangdong Basic and Applied Basic Research Foundation (2019A1515110014 to X.L.Z.), The Natural Science Foundation of Hainan Province (2017CXTD001 to X.Y.C.), The Fundamental Research Funds for the Central Universities (KY051067 to M.X.X.), Science and Technology Special Foundation of Guangdong Provincial People's Hospital, Guangdong Academy of Medical Sciences (2020bq20 to X.L.Z.).

## Author contributions

M.X., X.C., W.Z., R.L. and X.Z. designed the experiments. M.X., L.G., J.W., Y.X., X.Z. and D.C. performed the experiments. M.X., X.C., W.Z. and R.L. acquired and analysed the data. X.L. and X.Z. wrote and revised the paper.

## Competing interests

All other authors declare no competing interests.
