## [Peer Review File · Nature Communications]

Reviewers' Comments:

Reviewer #1:

Remarks to the Author:

Summary: The authors investigate the mechanism by which neuropathic pain and inflammation causes heat hyperalgesia. Since the transcription factor ATF4 is involved in various plastic changes in the central nervous system, the authors state that they wanted to test if ATF4 is also involved in heat nociception. While this logic is not compelling, the authors succeed to demonstrate that ATF4 is expressed in the expected location for such a function. Carefully conducted immunofluorescence experiments reveal the expression of ATF4 in sensory dorsal root ganglion (DRG) neurons. A wealth of behavioral data conclusively demonstrates that the level of ATF4 expression (knockdown and overexpression) correlates with heat sensitivity, but not other sensory stimuli or motor neuron output. Importantly, work from other labs demonstrated that inflammation and conditions that cause neuropathic pain can increase ATF4 expression, corroborating the notion that it is indeed ATF4 that is responsible for the increased heat sensitivity under these conditions. This reasoning should be presented more clearly in the manuscript. Next, the authors set out to determine the downstream effector of ATF4. They focus on the heat activated, non-specific cation channel TRPM3, which is a known 'heat sensor'. Interestingly, they find that ATF4 does not affect the expression level of TRPM3, but the abundance of TRPM3 on the plasma membrane of the sensory DRG neurons. To find the protein that mediates the redistribution of TRPM3 to the plasma membrane, the authors test the interaction of various molecular motor proteins with ATF4. They find that the kinesin motor KIF17 interacts with a complex containing ATF4 and TRPM3. Since the motility of KIF17 is directed towards the plasma membrane, the authors suggest that ATF4 might serve as an adaptor to tether KIF17 to an ATF4 and TRPM3 containing complex, thereby mediating the transport of TRPM3 to the plasma membrane. The adaptor function of ATF4 remains unconfirmed.

To sum up, the findings presented in this manuscript are interesting, novel, and potentially impactful. However, there are also some concerns that somewhat reduce the enthusiasm for this work. If the authors are able to adequately address these concerns, which I have listed below, I recommend this manuscript for publication in Nature Communications.

Major comments

1) Logic and grammar: The manuscript contains many grammatical mistakes and imprecisions (see minor comments), which should be corrected. In contrast to the carefully prepared data, the manuscript is not well written. For instance, important findings are not identified and are not put into context. It seems very surprising that a transcription factor (ATF4) regulates a process not via gene expression regulation, but by directly influencing motor protein-mediated trafficking. While this is a surprising finding and should be discussed as such, the presented data certainly support the authors' interpretation, i.e. sensory neurons in ATF4^{+/-} mice show normal development and innervation, and the total expression level of the effector, TRPM3, is not altered. Furthermore, different parts of the manuscript are logically not well connected. To compensate for that and generally facilitate the conveyance of the overarching concept, the reviewer suggests the inclusion of a figure presenting a model of the function of ATF4 in this manuscript.

2) Precisely explain method: It is surprising that "re-expression" of ATF4 in ATF4^{+/-} mice rescues heat associated behavior pretty much exactly to the wildtype level, whereas "overexpression" of ATF4 in wildtype mice results in behavior that is strongly enhanced over the wildtype: Compare e.g. ATF4 "re-expression" in Fig. 3 f (Hargreaves), g (tail-flick), h (hot plate) with ATF4 "overexpression" in Fig. 4 f (Hargreaves), g (tail-flick), h (hot plate). If ATF4 expression is mediated via the adeno-associated virus system in both instances, as described in the methods, the reviewer would expect that the behavioral rescue by ATF4 "re-expression" would look more similar to the ATF4 "overexpression" than to the WT. Since this is not the case, I am wondering if the ATF4 expression level is regulated or if the data are selected by a specific method. Please describe in detail.

3) Overinterpretation: This work reveals a cellular mechanism, but NOT a molecular mechanism for how inflammatory conditions lead to heat nociception (claim made e.g. line 61). A molecular mechanism is not necessary for the publication of this work, but this should not be incorrectly stated. For a molecular mechanism, I would at least expect that direct interactions between proteins are mapped and that the work explains how these interactions are regulated, e.g. by conformational changes, phosphorylation, etc., which is not the case in this study. Similarly, the authors claim to show the interaction of proteins (e.g. line 164 and 201: ATF4 with TRPM3, line 230: ATF4 with KIF17, line 242: TRPM3 with KIF17). This claim is principally correct, however, it should be clearly pointed out that these are not direct interactions. When coimmunoprecipitations of tagged ATF4 fragments with TRPM3 (Fig. 5) and KIF17 (Fig. 7) are shown, many readers will interpret this as mapping of direct interactions between the tested proteins. This is not the case here. Direct interactions can only be mapped between purified proteins, which rules out that secondary proteins mediate the interaction. Furthermore, statements like "To further confirm the ... interaction, we performed high-resolution images with structure illumination microscopy (SIM)," (lines 165-169 and 233-236) are misleading. At the shown resolution (Figs. 5 and 7), all one can see is that proteins co-localize roughly in some regions of the cell. Only dynamic co-localization in live cells shown at much higher magnification would convincingly support the claim that the proteins interact. Please also identify how the percent of colocalization that is mentioned in the text was scored from the images. This should be described in the methods.

4) Adaptor function of ATF4: ATF4 is suggested to function as an adaptor which recruits the KIF17 motor to TRPM3. If this suggestion is incorporated in the manuscript it should be tested. To do this, the direct interaction between ATF4 and KIF17 should be verified via coimmunoprecipitation of purified proteins. Additionally, the colocalization between TRPM3 and KIF17 in SIM images should be correlated with the expression level of ATF4, which should be altered by overexpression and knockdown. Again, the latter experiment would not show interaction, but in combination with the former experiment would convincingly demonstrate that ATF4 functions as adaptor that recruits KIF17 to TRPM3 containing trafficking complexes, which are most likely vesicles.

Minor comments

Protein and treatment aliases should be introduced:

Line 99: What is CFA?

Line 270 and below: What are CXCL12, CXCR4, CXCR7?

Lines 278-279: "Consistently, i.t. injection of CXCL12...". What is i.t.?

Line 12: Clarification: "ATF4 increased TRPM3 currents and maintained TRPM3 membrane localization..." Please clearly describe that ATF4 increases TRPM3-dependent currents by increasing TRPM3 abundance in the plasma membrane and not by e.g. altering currents through individual channels.

Line 25: Precise language: "...conductive to stress adaptation by mediating genes involved in metabolism,...". Change to something like: conducive to stress adaptation by mediating the change of expression levels of genes involved in metabolism.

Line 44: Correction: "...; these motor proteins mediate the microtubule-dependent transport of ...". Change to something like: and a large fraction of these motor proteins mediate the microtubule-dependent transport of... . Not all kinesins are involved in cargo transport.

Lines 54-56: Correction: "In addition, KIF17, or its homologue OSM-3, regulates cargo transport to the distal end of the cilia or flagella, thereby promoting cilia formation²¹." While OSM-3 mediates cilium formation, the vertebrate motor KIF17 is only implicated in transporting specialized signaling molecules in cilia, but has no function in cilium formation. See e.g. Jiang et al., 2015,

FASEB J. (DOI:10.1096/fj.15-275677) and Engelke et al., 2019, Curr Biol. (DOI: 10.1016/j.cub.2019.02.043). Please correct.

Lines 144-146: "Strikingly, intrathecal injection of adeno associated virus (rAAV-CMV-Atf4-2A-EGFP-WPRE-PA) to re-expression of ATF4 in DRG of ATF4+/- mice (Supplementary Fig. 4c)." This sentence does not make sense.

Lines 146-151: Repetition of results from previous section is not necessary.

Lines 197-200: "The hind paws...(Fig. 6a)". Please use full sentences in the main text, not protocol style.

Line 215: Precise language: "We further studied the effects of cultured DRG neurons...". Please change "cultured" to "isolated" or something like this. Culturing neurons implies that the neurons were not only isolated, but also cultured in cell culture vessels for an extended period of time. If I understand the experiment correctly, that was not the case here.

Lines 270-288: "CXCL12 mediates ATF4 upregulation via CXCR4." etc. The mechanism of ATF4 upregulation by inflammation seems to be a major finding of the study. Could this finding be presented as a main instead of a supplementary figure?

Lines 313-314: "Considering that the heat stimulation in the mouse skin endings,...". What are mouse skin endings? Please specify.

Line 345 Insert word in brackets: "This information can be applied to develop [further] strategies to treat..." Some strategies to treat conditions related to heat sensation seem to exist already, see references 22 and 33 of this manuscript. Hence, this study can only increase the repertoire of available strategies.

Line 479: Typo: "Enhanced chemiluminescence (ECL) solution (Milipore)...". Correct to Millipore.

Line 502-507: Please use full sentences, even in the methods part.

Figure 1: Images at the bottom of panel c are not representative of the data presented in panel d. The percentage of NF200 expressing ATF4 positive neurons seems higher than ATF4 positive neurons expressing IB4 and CGRP, not lower. Select representative images.

Figures 5, 7, and 8. Image acquisition with high NA objectives in combination with structured illumination produces images with very high spatial resolution. This cannot be appreciated from the shown images. Please show highly magnified inserts for the images shown in panels Fig 5i, Fig 7g, Fig 8c.

Figure 5: What do the open and closed circles in Fig. 5p, r, and t, represent? Please describe in the figure legend.

Reviewer #2:

Remarks to the Author:

The manuscript studies the role of the ATF4 transcription factor in nociceptive neurons. The key findings of the study are the following:

1. Loss, or reduction of ATF4 in DRG leads to impaired sensitivity to noxious heat, and impaired heat hyperalgesia, but sensitivity to mechanical stimuli is unaltered. Overexpression of ATF4 increases sensitivity to heat, but not to cold and to mechanical stimuli (Figs. 2-4).
2. Knockdown of ATF4 increases nociceptive behavior induced by local or intrathecal injection of

agonists of the heat-activated TRPM3 channel (Fig. 2-3).

3. Knockdown of ATF4 reduces surface expression of TRPM3 and decreases currents evoked by the TRPM3 agonist CIM0216 in DRG neurons (Fig. 5).

4. A short (30s) exposure to 43C increases co-immunoprecipitation of TRPM with ATF4, and increases TRPM3 surface expression and heat sensitivity in an ATF4 dependent manner (Fig. 6).

5. TRPM3 and ATF4 also coimmunoprecipitates with the kinesin motor protein KIF17 (Fig. 7).

6. Knockdown of KIF17 reduces TRPM3 surface expression, while overexpressing KIF17 increases it (Figure 8).

7. Nerve injury (Suppl Fig. 1) and CXCL12 treatment (Suppl Fig 8) increases ATF4 levels.

Overall the authors postulate that ATF4 interacts with KIF17 and TRPM3, which leads to increased trafficking of TRPM3 to the plasma membrane. Acute exposure to heat, as well as CXCL12 treatment appears to turn on this pathway. The manuscript presents an intriguing model for the involvement of ATF4 in peripheral heat sensation, based on an enormous amount of data. I have the following comments.

1. ATF4 is a transcription factor. Direct interaction with an ion channel and a motor protein seems quite an unorthodox role for a transcription factor. The authors should provide some discussion on what ATF4 is doing in the cytoplasm, and if any non-nuclear role of ATF4 is known. On the same note, it seems from figure 5 and figure 7 that ATF4 is excluded from the nucleus; maybe the authors should comment on this and discuss what usually induces its translocation to the nucleus, which I assume is needed to fulfill its role as a transcription factor.

2. On a similar note, it seems that the three proteins, TRPM3, KIF17 and ATF4 form a complex, as they co-immunoprecipitate in all three combinations. These co-IP data in most cases are shown by single representatives. The existence of this complex is a key point of the manuscript; thus, these data need to be shown in a more convincing manner, with disclosing the number of replicates and providing statistics.

3. TRPM3 trafficking to the plasma membrane is a key finding supporting the model. The authors use a commercial kit to separate the plasma membrane from other fractions. I would like to see this result confirmed with an independent technique for example cell surface biotinylation. At the minimum, the authors should provide a more detailed description of this technique. For example, in Fig. 5D they show increased signal in the cytoplasm, after ATF4 siRNA treatment. Is this a soluble fraction? TRPM3 is membrane protein, if it is not trafficked to the plasma membrane it is not supposed to be in the cytoplasm, rather in intracellular membranes.

4. Both the co-IP and the trafficking experiments rely on a single TRM3 antibody. Antibodies against ion channels are generally not very reliable. While some validation of the antibody is shown in Supplementary figure 5 this part needs to be strengthened. The exact source, including catalogue number, of the antibody needs to be disclosed and references provided for its successful usage. I am somewhat concerned by the almost completely cytoplasmic localization of TRPM3 in the SIM pictures in Figures 5 and 8, which raises some concerns about this TRPM3 antibody. I would like to see additional validation of this antibody and ideally verification of the key results with a different TRPM3 antibody.

5. Abstract lines 9-10: Loss of ATF4 in mouse dorsal root ganglion (DRG) neurons selectively impaired heat hyperalgesia. I think this is an overstatement, and these data need to be presented more carefully. In Fig 2 panels K and M, ATF4 knockdown increased latency to withdrawal from heat both in baseline conditions and after SNL or CFA injection to a similar extent. Looking at the same data from a different angle, one can easily say that ATF4 knockdown had no effect on hyperalgesia, as both in the presence and absence of SNL and CFA induced a large increase in heat sensitivity, ATF4 knockdown just shifted the baseline.

6. The involvement of TRPM3 in the effect of ATF4 on heat sensitivity is shown very convincingly

by the CIM and PS local injection experiments. The role of TRPV1 is excluded by a single experiment, the lack of increased PM expression of TRPV1 in Fig 5a. Antibodies against ion channels are generally not very reliable. The authors may consider testing the effect of ATF4 knockdown on nocifensive responses elicited by local capsaicin injection, which would greatly increase the confidence in the lack of involvement of TRPV1.

7. The catalogue numbers of all antibodies need to be disclosed

Minor comments:

Figure 5 panel p,r and t: I would make the Y axis the same on all three panels that would make the inhibition much easier to notice. Also, TRPM3 is an outwardly rectifying channel, but the currents in panel p and r show almost no rectification. Also, it is not clear what is plotted at panel o, is it the current induced by CIM, i.e. current before CIM subtracted, or the total current during CIM treatment, given the large baseline in the representatives, this is a big difference, therefore it needs to be clarified.

The order of panels should be uniform, in Fig 1 the letters go vertically, most other figures it is horizontal, but in Figure 5 it is mixed

Figure 8a-c: I would label the scale bars in the figure itself, given that a and b are low magnification pictures of whole DRG, while c is a single neuron. At the first look, the figure is confusing and gives the impression that panel C is also whole DRG given the same round shape.

Given that CXCL12 is likely one of the physiological mechanisms turning on the pathway proposed by the authors, they should briefly discuss what CXCL12 is.

ATF4 should be defined in the abstract and it should be stated that it is a transcription factor.

There are numerous typos and awkward, or grammatically incorrect sentences, a careful editing of the manuscript for language is highly recommended. Some examples are below:

Page 5 line 98 and 106 knockdown of ATF4

Page 7 line 131: we further intrathecally injected - delete "further"

Page 7 line 144: the following sentence is incomplete: Strikingly, intrathecal injection of adeno associated virus (rAAV-CMV-Atf4-2A-EGFP-WPRE-PA) to re-expression of ATF4 in DRG of ATF4+/- mice (Supplementary Fig. 4c).

Page 7 line 147: prominently rescued heat sensitivity (Fig. 3f-h) but not mechanical or cold sensitivity – ATF4 knockdown did not alter cold and mechanosensitivity, therefore there was nothing to rescue. I would change the end of the sentence to : but did not increase mechanical or cold sensitivity

Page 7: The rescue data described in Figure 3 (lines 147 - 151) should be described before the overexpression data which is presented in Figure 4.

Page 8: line 160: scrambled siRNA

Page 8 line 166, we performed high resolution imaging (instead of images)

Page 8 line 171: with the TRPM3-Flag construct

Page 10, line 212 injected into the hindpaw

Page 11 line 239 KIF17-Flag construct

Page 13 line 271: To investigate the underlying mechanism of ATF4 up-regulation

Page 16 line 313: "Considering that the heat stimulation..." This sentence is both complicated and grammatically incorrect.

Reviewer #3:

Remarks to the Author:

The major message of this ms is that ATF4 plays a role in regulating the thermal sensitivity response of dorsal root ganglion neurons in mice. The authors present data to support a model that ATF4 serves as an adapter for a complex with TRPM3 and Kif17 that traffics TRPM3 to the DRG neuronal surface and thereby regulates thermal sensitivity. While I do not necessarily accept the authors' model that ATF4 serves to regulate thermal sensitivity only by serving as a local adapter for TRPM3 trafficking (and not in its usual role as a nuclear transcription factor), the findings are very interesting with copious data and advance the field. There are some issues that need to be addressed as follows.

As noted above, while the findings are consistent with a role for ATF4 as a local adapter, in my view these are mainly correlative and do not establish causality. What is absent is a clear experiment to distinguish between the well-studied transcriptional actions of ATF4 and the non-transcriptional role postulated in this ms. For instance, does over-expression of a transcriptionally inactive form of ATF4 reverse the effects of ATF4 knockdown or heterogeneity? While this point is not a demand for such experiments in this ms, it does urge the authors to more closely consider their interpretation of the data.

In a related point, the ms reports that thermal stimulation rapidly lowers the thermal withdrawal threshold of mice, but not with ATF4 knockdown. The authors interpret this finding to a changed ATF4-TRPM3 interaction. An alternative explanation however is that prolonged ATF4 knockdown affects transcription of key proteins involved in this response.

In light of such points, rather than referring to the role of ATF4 in membrane trafficking of TRPM3 as "ATF4-mediated" the authors might wish to refer to this role as "ATF4-dependent"

Additional points:

1. The authors cite Costa-Mattioli et al and Chen et al as showing that "ATF4 plays a critical role in synaptic plasticity and memory formation". However, close reading of the papers indicates that neither makes a direct case for such a role of ATF4.
2. The authors state (lines 68-69): "suggesting that ATF4 synthesized in somata of DRG neurons is transported to their central and peripheral terminals." However, the literature suggests that ATF4 can be locally synthesized in distal portions of neurons (at least in the CNS) via enhanced translation.
3. The sequences of the various siRNAs used in the work need to be specified.
4. The authors use "blocking peptides" to authenticate their immunostaining. However, the sequences and sources of these need to be given. Additionally, while helpful, such peptides do not necessarily rule out cross-reactivity for immunostaining.
5. Suppl Fig 1D. The legend describes "infection", but the text indicates siRNA treatment. The

sequence of "FITC-siRNA" not appear to be described in the ms.

6. For ATF4 siRNA treatment, what was the time that blotting, staining and experiments were performed after injection? For experiments such as shown in Fig 2, etc, was the time after treatment comparable to that for the blot and staining? That is, is the knockdown present at the time of experimentation?

7. It would have been useful if ATF4 het mice had been used as well as ATF4 knockdown to study TRPM3 localization from surface to cytoplasm.

8. While well-written, the ms needs to be edited for minor errors in English language that sometimes affect clarity.

REVIEWER COMMENTS

Reviewer #1 (Remarks to the Author):

Summary: The authors investigate the mechanism by which neuropathic pain and inflammation causes heat hyperalgesia. Since the transcription factor ATF4 is involved in various plastic changes in the central nervous system, the authors state that they wanted to test if ATF4 is also involved in heat nociception. While this logic is not compelling, the authors succeed to demonstrate that ATF4 is expressed in the expected location for such a function. Carefully conducted immunofluorescence experiments reveal the expression of ATF4 in sensory dorsal root ganglion (DRG) neurons. A wealth of behavioral data conclusively demonstrates that the level of ATF4 expression (knockdown and overexpression) correlates with heat sensitivity, but not other sensory stimuli or motor neuron output. Importantly, work from other labs demonstrated that inflammation and conditions that cause neuropathic pain can increase ATF4 expression, corroborating the notion that it is indeed ATF4 that is responsible for the increased heat sensitivity under these conditions. This reasoning should be presented more clearly in the manuscript. Next, the authors set out to determine the downstream effector of ATF4. They focus on the heat activated, non-specific cation channel TRPM3, which is a known 'heat sensor'. Interestingly, they find that ATF4 does not affect the expression level of TRPM3, but the abundance of TRPM3 on the plasma membrane of the sensory DRG neurons. To find the protein that mediates the redistribution of TRPM3 to the plasma membrane, the authors test the interaction of various molecular motor proteins with ATF4. They find that the kinesin motor KIF17 interacts with a complex containing ATF4 and TRPM3. Since the motility of KIF17 is directed towards the plasma membrane, the authors suggest that ATF4 might serve as an adaptor to tether KIF17 to an ATF4 and TRPM3 containing complex, thereby mediating the transport of TRPM3 to the plasma membrane. The adaptor function of ATF4 remains unconfirmed.

To sum up, the findings presented in this manuscript are interesting, novel, and potentially impactful. However, there are also some concerns that somewhat reduce the enthusiasm for this work. If the authors are able to adequately address these concerns, which I have listed below, I recommend this manuscript for publication in Nature Communications.

Major comments

1) Logic and grammar: The manuscript contains many grammatical mistakes and imprecisions (see minor comments), which should be corrected. In contrast to the carefully prepared data, the manuscript is not well written. For instance, important findings are not identified and are not put into context. It seems very surprising that a transcription factor (ATF4) regulates a process not via gene

expression regulation, but by directly influencing motor protein-mediated trafficking. While this is a surprising finding and should be discussed as such, the presented data certainly support the authors' interpretation, i.e. sensory neurons in ATF4^{+/-} mice show normal development and innervation, and the total expression level of the effector, TRPM3, is not altered. Furthermore, different parts of the manuscript are logically not well connected. To compensate for that and generally facilitate the conveyance of the overarching concept, the reviewer suggests the inclusion of a figure presenting a model of the function of ATF4 in this manuscript.

2) Precisely explain method: It is surprising that "re-expression" of ATF4 in ATF4^{+/-} mice rescues heat associated behavior pretty much exactly to the wildtype level, whereas "overexpression" of ATF4 in wildtype mice results in behavior that is strongly enhanced over the wildtype: Compare e.g. ATF4 "re-expression" in Fig. 3 f (Hargreaves), g (tail-flick), h (hot plate) with ATF4 "overexpression" in Fig. 4 f (Hargreaves), g (tail-flick), h (hot plate). If ATF4 expression is mediated via the adeno-associated virus system in both instances, as described in the methods, the reviewer would expect that the behavioral rescue by ATF4 "re-expression" would look more similar to the ATF4 "overexpression" than to the WT. Since this is not the case, I am wondering if the ATF4 expression level is regulated or if the data are selected by a specific method. Please describe in detail.

3) Overinterpretation: This work reveals a cellular mechanism, but NOT a molecular mechanism for how inflammatory conditions lead to heat nociception (claim made e.g. line 61). A molecular mechanism is not necessary for the publication of this work, but this should not be incorrectly stated. For a molecular mechanism, I would at least expect that direct interactions between proteins are mapped and that the work explains how these interactions are regulated, e.g. by conformational changes, phosphorylation, etc., which is not the case in this study.

Similarly, the authors claim to show the interaction of proteins (e.g. line 164 and 201: ATF4 with TRPM3, line 230: ATF4 with KIF17, line 242: TRPM3 with KIF17). This claim is principally correct, however, it should be clearly pointed out that these are not direct interactions. When coimmunoprecipitations of tagged ATF4 fragments with TRPM3 (Fig. 5) and KIF17 (Fig. 7) are shown, many readers will interpret this as mapping of direct interactions between the tested proteins. This is not the case here. Direct interactions can only be mapped between purified proteins, which rules out that secondary proteins mediate the interaction.

Furthermore, statements like "To further confirm the ... interaction, we performed high-resolution images with structure illumination microscopy (SIM)," (lines 165-169 and 233-236) are misleading. At the shown resolution (Figs. 5 and 7), all one can see is that proteins co-localize roughly in some

regions of the cell. Only dynamic co-localization in live cells shown at much higher magnification would convincingly support the claim that the proteins interact. Please also identify how the percent of colocalization that is mentioned in the text was scored from the images. This should be described in the methods.

4) Adaptor function of ATF4: ATF4 is suggested to function as an adaptor which recruits the KIF17 motor to TRPM3. If this suggestion is incorporated in the manuscript it should be tested. To do this, the direct interaction between ATF4 and KIF17 should be verified via coimmunoprecipitation of purified proteins. Additionally, the colocalization between TRPM3 and KIF17 in SIM images should be correlated with the expression level of ATF4, which should be altered by overexpression and knockdown. Again, the latter experiment would not show interaction, but in combination with the former experiment would convincingly demonstrate that ATF4 functions as adaptor that recruits KIF17 to TRPM3 containing trafficking complexes, which are most likely vesicles.

Minor comments

Protein and treatment aliases should be introduced:

Line 99: What is CFA?

Line 270 and below: What are CXCL12, CXCR4, CXCR7?

Lines 278-279: "Consistently, i.t. injection of CXCL12...". What is i.t.?

Line 12: Clarification: "ATF4 increased TRPM3 currents and maintained TRPM3 membrane localization..." Please clearly describe that ATF4 increases TRPM3-dependent currents by increasing TRPM3 abundance in the plasma membrane and not by e.g. altering currents through individual channels.

Line 25: Precise language: "...conducive to stress adaptation by mediating genes involved in metabolism,...". Change to something like: conducive to stress adaptation by mediating the change of expression levels of genes involved in metabolism.

Line 44: Correction: "...; these motor proteins mediate the microtubule-dependent transport of ...". Change to something like: and a large fraction of these motor proteins mediate the microtubule-dependent transport of... . Not all kinesins are involved in cargo transport.

Lines 54-56: Correction: "In addition, KIF17, or its homologue OSM-3, regulates cargo transport to the distal end of the cilia or flagella, thereby promoting cilia formation²¹." While OSM-3 mediates cilium formation, the vertebrate motor KIF17 is only implicated in transporting specialized signaling molecules in cilia, but has no function in cilium formation. See e.g. Jiang et al.,

2015, FASEB J. (DOI:10.1096/fj.15-275677) and Engelke et al., 2019, Curr Biol. (DOI: 10.1016/j.cub.2019.02.043). Please correct.

Lines 144-146: "Strikingly, intrathecal injection of adeno associated virus (rAAV-CMV-Atf4-2A-EGFP-WPRE-PA) to re-expression of ATF4 in DRG of ATF4+/- mice (Supplementary Fig. 4c)." This sentence does not make sense.

Lines 146-151: Repetition of results from previous section is not necessary.

Lines 197-200: "The hind paws...(Fig. 6a)". Please use full sentences in the main text, not protocol style.

Line 215: Precise language: "We further studied the effects of cultured DRG neurons...". Please change "cultured" to "isolated" or something like this. Culturing neurons implies that the neurons were not only isolated, but also cultured in cell culture vessels for an extended period of time. If I understand the experiment correctly, that was not the case here.

Lines 270-288: "CXCL12 mediates ATF4 upregulation via CXCR4." etc. The mechanism of ATF4 upregulation by inflammation seems to be a major finding of the study. Could this finding be presented as a main instead of a supplementary figure?

Lines 313-314: "Considering that the heat stimulation in the mouse skin endings,...". What are mouse skin endings? Please specify.

Line 345 Insert word in brackets: "This information can be applied to develop [further] strategies to treat..." Some strategies to treat conditions related to heat sensation seem to exist already, see references 22 and 33 of this manuscript. Hence, this study can only increase the repertoire of available strategies.

Line 479: Typo: "Enhanced chemiluminescence (ECL) solution (Milipore)...". Correct to Millipore.

Line 502-507: Please use full sentences, even in the methods part.

Figure 1: Images at the bottom of panel c are not representative of the data presented in panel d. The percentage of NF200 expressing ATF4 positive neurons seems higher than ATF4 positive neurons expressing IB4 and CGRP, not lower. Select representative images.

Figures 5, 7, and 8. Image acquisition with high NA objectives in combination with structured illumination produces images with very high spatial resolution.

This cannot be appreciated from the shown images. Please show highly magnified inserts for the images shown in panels Fig 5i, Fig 7g, Fig 8c.

Figure 5: What do the open and closed circles in Fig. 5p, r, and t, represent? Please describe in the figure legend.

Reviewer #2 (Remarks to the Author):

The manuscript studies the role of the ATF4 transcription factor in nociceptive neurons. The key findings of the study are the following:

1. Loss, or reduction of ATF4 in DRG leads to impaired sensitivity to noxious heat, and impaired heat hyperalgesia, but sensitivity to mechanical stimuli is unaltered. Overexpression of ATF4 increases sensitivity to heat, but not to cold and to mechanical stimuli (Figs. 2-4).
2. Knockdown of ATF4 increases nociceptive behavior induced by local or intrathecal injection of agonists of the heat-activated TRPM3 channel (Fig. 2-3).
3. Knockdown of ATF4 reduces surface expression of TRPM3 and decreases currents evoked by the TRPM3 agonist CIM0216 in DRG neurons (Fig. 5).
4. A short (30s) exposure to 43C increases co-immunoprecipitation of TRPM3 with ATF4, and increases TRPM3 surface expression and heat sensitivity in an ATF4 dependent manner (Fig. 6).
5. TRPM3 and ATF4 also coimmunoprecipitates with the kinesin motor protein KIF17 (Fig. 7).
6. Knockdown of KIF17 reduces TRPM3 surface expression, while overexpressing KIF17 increases it (Figure 8).
7. Nerve injury (Suppl Fig. 1) and CXCL12 treatment (Suppl Fig 8) increases ATF4 levels.

Overall the authors postulate that ATF4 interacts with KIF17 and TRPM3, which leads to increased trafficking of TRPM3 to the plasma membrane. Acute exposure to heat, as well as CXCL12 treatment appears to turn on this pathway. The manuscript presents an intriguing model for the involvement of ATF4 in peripheral heat sensation, based on an enormous amount of data. I have the following comments.

1. ATF4 is a transcription factor. Direct interaction with an ion channel and a motor protein seems quite an unorthodox role for a transcription factor. The authors should provide some discussion on what ATF4 is doing in the cytoplasm, and if any non-nuclear role of ATF4 is known. On the same note, it seems from figure 5 and figure 7 that ATF4 is excluded from the nucleus; maybe the authors should comment on this and discuss what usually induces its translocation to

the nucleus, which I assume is needed to fulfill its role as a transcription factor.

2. On a similar note, it seems that the three proteins, TRPM3, KIF17 and ATF4 form a complex, as they co-immunoprecipitate in all three combinations. These co-IP data in most cases are shown by single representatives. The existence of this complex is a key point of the manuscript; thus, these data need to be shown in a more convincing manner, with disclosing the number of replicates and providing statistics.

3. TRPM3 trafficking to the plasma membrane is a key finding supporting the model. The authors use a commercial kit to separate the plasma membrane from other fractions. I would like to see this result confirmed with an independent technique for example cell surface biotinylation. At the minimum, the authors should provide a more detailed description of this technique. For example, in Fig. 5D they show increased signal in the cytoplasm, after ATF4 siRNA treatment. Is this a soluble fraction? TRPM3 is membrane protein, if it is not trafficked to the plasma membrane it is not supposed to be in the cytoplasm, rather in intracellular membranes.

4. Both the co-IP and the trafficking experiments rely on a single TRM3 antibody. Antibodies against ion channels are generally not very reliable. While some validation of the antibody is shown in Supplementary figure 5 this part needs to be strengthened. The exact source, including catalogue number, of the antibody needs to be disclosed and references provided for its successful usage. I am somewhat concerned by the almost completely cytoplasmic localization of TRPM3 in the SIM pictures in Figures 5 and 8, which raises some concerns about this TRPM3 antibody. I would like to see additional validation of this antibody and ideally verification of the key results with a different TRPM3 antibody.

5. Abstract lines 9-10: Loss of ATF4 in mouse dorsal root ganglion (DRG) neurons selectively impaired heat hyperalgesia. I think this is an overstatement, and these data need to be presented more carefully. In Fig 2 panels K and M, ATF4 knockdown increased latency to withdrawal from heat both in baseline conditions and after SNL or CFA injection to a similar extent. Looking at the same data from a different angle, one can easily say that ATF4 knockdown had no effect on hyperalgesia, as both in the presence and absence of it SNL and CFA induced a large increase in heat sensitivity, ATF4 knockdown just shifted the baseline.

6. The involvement of TRPM3 in the effect of ATF4 on heat sensitivity is shown very convincingly by the CIM and PS local injection experiments. The role of TRPV1 is excluded by a single experiment, the lack of increased PM expression of TRPV1 in Fig 5a. Antibodies against ion channels are generally not very reliable. The authors may consider testing the effect of ATF4 knockdown on

nocifensive responses elicited by local capsaicin injection, which would greatly increase the confidence in the lack of involvement of TRPV1.

7. The catalogue numbers of all antibodies need to be disclosed

Minor comments:

Figure 5 panel p,r and t: I would make the Y axis the same on all three panels that would make the inhibition much easier to notice. Also, TRPM3 is an outwardly rectifying channel, but the currents in panel p and r show almost no rectification. Also, it is not clear what is plotted at panel o, is it the current induced by CIM, i.e. current before CIM subtracted, or the total current during CIM treatment, given the large baseline in the representatives, this is a big difference, therefore it needs to be clarified.

The order of panels should be uniform, in Fig 1 the letters go vertically, most other figures it is horizontal, but in Figure 5 it is mixed

Figure 8a-c: I would label the scale bars in the figure itself, given that a and b are low magnification pictures of whole DRG, while c is a single neuron. At the first look, the figure is confusing and gives the impression that panel C is also whole DRG given the same round shape.

Given that CXCL12 is likely one of the physiological mechanisms turning on the pathway proposed by the authors, they should briefly discuss what CXCL12 is.

ATF4 should be defined in the abstract and it should be stated that it is a transcription factor.

There are numerous typos and awkward, or grammatically incorrect sentences, a careful editing of the manuscript for language is highly recommended. Some examples are below:

Page 5 line 98 and 106 knockdown of ATF4

Page 7 line 131: we further intrathecally injected - delete "further"

Page 7 line 144: the following sentence is incomplete: Strikingly, intrathecal injection of adeno associated virus (rAAV-CMV-Atf4-2A-EGFP-WPRE-PA) to re-expression of ATF4 in DRG of ATF4+/- mice (Supplementary Fig. 4c).

Page 7 line 147: prominently rescued heat sensitivity (Fig. 3f-h) but not mechanical or cold sensitivity – ATF4 knockdown did not alter cold and mechanosensitivity, therefore there was nothing to rescue. I would change the

end of the sentence to : but did not increase mechanical or cold sensitivity

Page 7: The rescue data described in Figure 3 (lines 147 - 151) should be described before the overexpression data which is presented in Figure 4.

Page 8: line 160: scrambled siRNA

Page 8 line 166, we performed high resolution imaging (instead of images)

Page 8 line 171: with the TRPM3-Flag construct

Page 10, line 212 injected into the hindpaw

Page 11 line 239 KIF17-Flag construct

Page 13 line 271: To investigate the underlying mechanism of ATF4 up-regulation

Page 16 line 313: "Considering that the heat stimulation..." This sentence is both complicated and grammatically incorrect.

Reviewer #3 (Remarks to the Author):

The major message of this ms is that ATF4 plays a role in regulating the thermal sensitivity response of dorsal root ganglion neurons in mice. The authors present data to support a model that ATF4 serves as an adapter for a complex with TRMP3 and Kif17 that traffics TRMP3 to the DRG neuronal surface and thereby regulates thermal sensitivity. While I do not necessarily accept the authors' model that ATF4 serves to regulate thermal sensitivity only by serving as a local adapter for TRPM3 trafficking (and not in its usual role as a nuclear transcription factor), the findings are very interesting with copious data and advance the field. There are some issues that need to be addressed as follows.

As noted above, while the findings are consistent with a role for ATF4 as a local adapter, in my view these are mainly correlative and do not establish causality. What is absent is a clear experiment to distinguish between the well-studied transcriptional actions of ATF4 and the non-transcriptional role postulated in this ms. For instance, does over-expression of a transcriptionally inactive form of ATF4 reverse the effects of ATF4 knockdown or heterogeneity? While this point is not a demand for such experiments in this ms, it does urge the authors to more closely consider their interpretation of the data.

In a related point, the ms reports that thermal stimulation rapidly lowers the thermal withdrawal threshold of mice, but not with ATF4 knockdown. The authors interpret this finding to a changed ATF4-TRPM3 interaction. An alternative explanation however is that prolonged ATF4 knockdown affects transcription of key proteins involved in this response.

In light of such points, rather than referring to the role of ATF4 in membrane trafficking of TRPM3 as "ATF4-mediated" the authors might wish to refer to this role as "ATF4-dependent "

Additional points:

1. The authors cite Costa-Mattioli et al and Chen et al as showing that "ATF4 plays a critical role in synaptic plasticity and memory formation". However, close reading of the papers indicates that neither makes a direct case for such a role of ATF4.
2. The authors state (lines 68-69): "suggesting that ATF4 synthesized in somata of DRG neurons is transported to their central and peripheral terminals." However, the literature suggests that ATF4 can be locally synthesized in distal portions of neurons (at least in the CNS) via enhanced translation.
3. The sequences of the various siRNAs used in the work need to be specified.
4. The authors use "blocking peptides" to authenticate their immunostaining. However, the sequences and sources of these need to be given. Additionally, while helpful, such peptides do not necessarily rule out cross-reactivity for immunostaining.
5. Suppl Fig 1D. The legend describes "infection", but the text indicates siRNA treatment. The sequence of "FITC-siRNA" not appear to be described in the ms.
6. For ATF4 siRNA treatment, what was the time that blotting, staining and experiments were performed after injection? For experiments such as shown in Fig 2, etc, was the time after treatment comparable to that for the blot and staining? That is, is the knockdown present at the time of experimentation?
7. It would have been useful if ATF4 het mice had been used as well as ATF4 knockdown to study TRPM3 localization from surface to cytoplasm.
8. While well-written, the ms needs to be edited for minor errors in English language that sometimes affect clarity.

Reviewer #1:

1): Logic and grammar: The manuscript contains many grammatical mistakes and imprecisions (see minor comments), which should be corrected. In contrast to the carefully prepared data, the manuscript is not well written. For instance, important findings are not identified and are not put into context. It seems very surprising that a transcription factor (ATF4) regulates a process not via gene expression regulation, but by directly influencing motor protein-mediated trafficking. While this is a surprising finding and should be discussed as such, the presented data certainly support the authors' interpretation, i.e. sensory neurons in ATF4^{+/-} mice show normal development and innervation, and the total expression level of the effector, TRPM3, is not altered. Furthermore, different parts of the manuscript are logically not well connected. To compensate for that and generally facilitate the conveyance of the overarching concept, the reviewer suggests the inclusion of a figure presenting a model of the function of ATF4 in this manuscript.

Answer: We appreciate this comment from the reviewer. According to the reviewer's suggestion, we carefully corrected the grammatical errors and imprecisions throughout the manuscript. Our findings showed that loss of ATF4 significantly decreases the membrane expression of TRPM3 without changing its total expression in sensory neurons. This is a surprising finding and indicate that ATF4, a transcription factor, regulates the TRPM3 not via gene expression regulation, but by directly influencing TRPM3 trafficking. This phenomenon may represent a novel function of transcription factors in addition to transcription but requires further study. To better clarify our findings, we discussed this topic in the revised manuscript (Lines 346-353). In addition, to facilitate the conveyance of the overarching concept and make the manuscript more logical, we added a figure presenting a model of ATF4 function in the revised manuscript (Fig. 9e).

2): Precisely explain method: It is surprising that "re-expression" of ATF4 in ATF4+/- mice rescues heat associated behavior pretty much exactly to the wildtype level, whereas "overexpression" of ATF4 in wildtype mice results in behavior that is strongly enhanced over the wildtype: Compare e.g. ATF4 "re-expression" in Fig. 3 f (Hargreaves), g (tail-flick), h (hot plate) with ATF4 "overexpression" in Fig. 4 f (Hargreaves), g (tail-flick), h (hot plate). If ATF4 expression is mediated via the adeno-associated virus system in both instances, as described in the methods, the reviewer would expect that the behavioral rescue by ATF4 "re-expression" would look more similar to the ATF4 "overexpression" than to the WT. Since this is not the case, I am wondering if the ATF4 expression level is regulated or if the data are selected by a specific method. Please describe in detail.

Answer: Thank you for this comment. Our data showed that compared with that in control wild-type mice, the expression level of ATF4 was significantly increased in ATF4-overexpressing wild-type mice (Fig. S4b). However, the expression level of ATF4 in ATF4-rescued mice was not different from that in wild-type mice (Fig. S4a). Since the expression level of ATF4 is closely related to heat pain, the heat pain sensitivity of ATF4-rescued mice is similar to that of wild-type mice rather than overexpressed mice.

3): Overinterpretation: This work reveals a cellular mechanism, but NOT a molecular mechanism for how inflammatory conditions lead to heat nociception (claim made e.g. line 61). A molecular mechanism is not necessary for the publication of this work, but this should not be incorrectly stated. For a molecular mechanism, I would at least expect that direct interactions between proteins are mapped and that the work explains how these interactions are regulated, e.g. by conformational changes, phosphorylation, etc., which is not the case in this study.

Similarly, the authors claim to show the interaction of proteins (e.g. line 164

and 201: ATF4 with TRPM3, line 230: ATF4 with KIF17, line 242: TRPM3 with KIF17). This claim is principally correct, however, it should be clearly pointed out that these are not direct interactions. When coimmunoprecipitations of tagged ATF4 fragments with TRPM3 (Fig. 5) and KIF17 (Fig. 7) are shown, many readers will interpret this as mapping of direct interactions between the tested proteins. This is not the case here. Direct interactions can only be mapped between purified proteins, which rules out that secondary proteins mediate the interaction.

Furthermore, statements like "To further confirm the ... interaction, we performed high-resolution images with structure illumination microscopy (SIM)," (lines 165-169 and 233-236) are misleading. At the shown resolution (Figs. 5 and 7), all one can see is that proteins co-localize roughly in some regions of the cell. Only dynamic co-localization in live cells shown at much higher magnification would convincingly support the claim that the proteins interact. Please also identify how the percent of colocalization that is mentioned in the text was scored from the images. This should be described in the methods.

Answer: We appreciate this important comment. According to the reviewer's suggestion, we changed "molecular mechanism" to "cellular mechanism" in our revised manuscript (Line 65). Furthermore, we used purified proteins to study whether a direct interaction exists between proteins (ATF4 with TRPM3 and ATF4 with KIF17). GST pull-down assay data showed that ATF4 directly interacted with TRPM3 (Fig. 5g) and KIF17 (Fig. 7h). Considering that the resolution of SIM images does not prove the interaction between proteins, we corrected this description (to further confirm the ... interaction) in the revised manuscript (Lines 180, 267 and 287). We used IPWIN60 software to analyse the colocalization percentage of images. First, we recorded the intensity of colocalization spots and then measured the intensity of single-colour spots. The intensity of the colocalization spots divided by the intensity of the single-colour

spots was considered the percentage of colocalization. According to the reviewer's suggestion, we added the method used to determine the percentage of colocalization in the revised manuscript (Lines 688-691).

4): Adaptor function of ATF4: ATF4 is suggested to function as an adaptor which recruits the KIF17 motor to TRPM3. If this suggestion is incorporated in the manuscript it should be tested. To do this, the direct interaction between ATF4 and KIF17 should be verified via coimmunoprecipitation of purified proteins. Additionally, the colocalization between TRPM3 and KIF17 in SIM images should be correlated with the expression level of ATF4, which should be altered by overexpression and knockdown. Again, the latter experiment would not show interaction, but in combination with the former experiment would convincingly demonstrate that ATF4 functions as adaptor that recruits KIF17 to TRPM3 containing trafficking complexes, which are most likely vesicles.

Answer: We appreciate this important comment. To further confirm the adaptor function of ATF4, we used purified proteins to study whether a direct interaction exists between proteins (ATF4 with TRPM3 and ATF4 with KIF17). GST pull-down assay data showed that ATF4 directly interacted with TRPM3 (Fig. 5g) and KIF17 (Fig. 7h). Furthermore, TRPM3 also showed a potential interaction with KIF17, and the interaction level between TRPM3 and KIF17 was decreased in the DRGs after ATF4-siRNA treatment (Fig. 7j). Additionally, knockdown of ATF4 decreased the colocalization of TRPM3 and KIF17 in SIM images (Fig. 7k), whereas the colocalization between TRPM3 and KIF17 in SIM images was increased after ATF4 overexpression (Fig. 7k). Thus, ATF4 is suggested to function as an adaptor that recruits the KIF17 to TRPM3.

5): Protein and treatment aliases should be introduced:

Line 99: What is CFA?

Line 270 and below: What are CXCL12, CXCR4, CXCR7?

Lines 278-279: "Consistently, i.t. injection of CXCL12...". What is i.t.?

Answer: Thank you for this comment. CFA is complete Freund's adjuvant (Lines 105-106). Intraplantar injection of CFA can induce an inflammatory pain model in rodents. Chemokines have been classified into CXC, CC, C, and CX3C subfamilies based on the arrangement of the two cysteine residues near the N-terminus¹. Stromal cell-derived factor-1a (SDF-1a), now also named chemokine C-X-C motif ligand 12 (CXCL12), belongs to the CXC chemokine subfamily (Lines 306-307). The chemokine C-X-C motif receptor 4 (CXCR4) and chemokine C-X-C motif receptor 7 (CXCR7) are receptors for CXCL12 and belong to the G-protein coupled receptor family^{2,3} (Lines 315-316). In addition, i.t. injection is short for intrathecal injection (Line 314). To make our article more readable, we further explained these abbreviations in the revised manuscript.

6): Line 12: Clarification: "ATF4 increased TRPM3 currents and maintained TRPM3 membrane localization..." Please clearly describe that ATF4 increases TRPM3-dependent currents by increasing TRPM3 abundance in the plasma membrane and not by e.g. altering currents through individual channels.

Answer: We appreciate this important comment. We know that changes in ion channel current depend on two factors: the membrane abundance of the channel and the characteristics of the individual channel. Our data showed that loss of ATF4 significantly decreased the membrane expression of TRPM3 and reduced the TRPM3 current. These results suggested that the effect of ATF4 on TRPM3 membrane expression may be one of the mechanisms by which ATF4 regulates the TRPM3 current. However, whether ATF4 also regulates the TRPM3 current by altering the characteristics of individual TRPM3 channel is unclear and requires further study. We corrected the relevant descriptions to make them more accurate (Lines 12-14) and discussed it in the revised manuscript (Lines 384-392).

7): Line 25: Precise language: "...conducive to stress adaptation by mediating genes involved in metabolism,...". Change to something like: conducive to stress adaptation by mediating the change of expression levels of genes involved in metabolism.

Answer: Thank you for this comment. According to the reviewer's suggestion, we changed "...conducive to stress adaptation by mediating genes involved in metabolism..." to "...conducive to stress adaptation by mediating the change of expression levels of genes involved in metabolism..." in our revised manuscript (Lines 27-28).

8): Line 44: Correction: "...; these motor proteins mediate the microtubule-dependent transport of ...". Change to something like: and a large fraction of these motor proteins mediate the microtubule-dependent transport of... . Not all kinesins are involved in cargo transport.

Answer: Thank you for this comment. According to the reviewer's suggestion, we changed "...; these motor proteins mediate the microtubule-dependent transport of..." to "...and a large fraction of these motor proteins mediate the microtubule-dependent transport of..." in our revised manuscript (Line 46).

9): Lines 54-56: Correction: "In addition, KIF17, or its homologue OSM-3, regulates cargo transport to the distal end of the cilia or flagella, thereby promoting cilia formation²¹." While OSM-3 mediates cilium formation, the vertebrate motor KIF17 is only implicated in transporting specialized signaling molecules in cilia, but has no function in cilium formation. See e.g. Jiang et al., 2015, FASEB J. (DOI:10.1096/fj.15-275677) and Engelke et al., 2019, Curr Biol. (DOI: 10.1016/j.cub.2019.02.043). Please correct.

Answer: We appreciate this important comment. According to the reviewer's suggestion, we corrected this mistake and reorganized it in the revised manuscript (Lines 58-60).

10): Lines 144-146: "Strikingly, intrathecal injection of adeno associated virus (rAAV-CMV-Atf4-2A-EGFP-WPRE-PA) to re-expression of ATF4 in DRG of ATF4+/- mice (Supplementary Fig. 4c)." This sentence does not make sense.

Answer: Thank you for this comment. To make the sentence clearer, we rewrote it in the revised manuscript (Lines 137-140).

11): Lines 146-151: Repetition of results from previous section is not necessary.

Answer: Thank you for the comment. According to the reviewer's suggestion, we rewrote the results of this section (Lines 140-144).

12): Lines 197-200: "The hind paws...(Fig. 6a)". Please use full sentences in the main text, not protocol style.

Answer: We appreciate this important comment. According to the reviewer's suggestion, we used full sentences to describe this section in the revised manuscript (Lines 231-234).

13): Line 215: Precise language: "We further studied the effects of cultured DRG neurons...". Please change "cultured" to "isolated" or something like this. Culturing neurons implies that the neurons were not only isolated, but also cultured in cell culture vessels for an extended period of time. If I understand the experiment correctly, that was not the case here.

Answer: Thank you for the comment. According to the reviewer's suggestion, we changed "cultured" to "isolated" in the revised manuscript (Line 247).

14): Lines 270-288: "CXCL12 mediates ATF4 upregulation via CXCR4." etc. The mechanism of ATF4 upregulation by inflammation seems to be a major finding of the study. Could this finding be presented as a main instead of a supplementary figure?

Answer: We appreciate this important comment. According to the reviewer's suggestion, we presented the finding of "CXCL12 mediates ATF4 upregulation via CXCR4" as a main figure (Fig. 9).

15): Lines 313-314: "Considering that the heat stimulation in the mouse skin endings,...". What are mouse skin endings? Please specify.

Answer: Thank you for the comment. The skin endings are nerve endings distributed in the skin. To improve the readability of the article, we changed "skin endings" to "skin nerve endings" (Line 397).

16): Line 345 Insert word in brackets: "This information can be applied to develop [further] strategies to treat..." Some strategies to treat conditions related to heat sensation seem to exist already, see references 22 and 33 of this manuscript. Hence, this study can only increase the repertoire of available strategies.

Answer: Thank you for this comment. To make our language more accurate, we inserted the word "further" into the manuscript according to the reviewer's suggestion (Line 453).

17): Line 479: Typo: "Enhanced chemiluminescence (ECL) solution (Milipore)...". Correct to Millipore.

Answer: We appreciate this comment from the reviewer. According to the reviewer's suggestion, we corrected the word (Line 644) and re-checked the whole manuscript carefully.

18): Line 502-507: Please use full sentences, even in the methods part.

Answer: We appreciate this comment. According to the reviewer's suggestion, we rewrote this section in the revised manuscript (Lines 656-659).

19): Figure 1: Images at the bottom of panel c are not representative of the data presented in panel d. The percentage of NF200 expressing ATF4 positive neurons seems higher than ATF4 positive neurons expressing IB4 and CGRP, not lower. Select representative images.

Answer: Thank you for this comment. According to the reviewer's suggestion, we updated the representative images in Fig. 1c.

20): Figures 5, 7, and 8. Image acquisition with high NA objectives in combination with structured illumination produces images with very high spatial resolution. This cannot be appreciated from the shown images. Please show highly magnified inserts for the images shown in panels Fig 5i, Fig 7g, Fig 8c.

Answer: We appreciate this comment from the reviewer. According to the reviewer's suggestion, we showed highly magnified inserts for the images shown in panels Fig. 5f, Fig. 7g and Fig. 8c.

21): Figure 5: What do the open and closed circles in Fig. 5p, r, and t, represent? Please describe in the figure legend.

Answer: Thank you for this comment. The open circle indicates the outward current recorded at + 75 mV, and the closed circle indicates the inward current recorded at - 75 mV. According to the reviewer's suggestion, we described this information in the revised figure legend (Lines 1021-1023).

Reviewer #2:

1): ATF4 is a transcription factor. Direct interaction with an ion channel and a motor protein seems quite an unorthodox role for a transcription factor. The authors should provide some discussion on what ATF4 is doing in the cytoplasm, and if any non-nuclear role of ATF4 is known. On the same note, it seems from figure 5 and figure 7 that ATF4 is excluded from the nucleus; maybe the authors should comment on this and discuss what usually induces its translocation to the nucleus, which I assume is needed to fulfill its role as a transcription factor.

Answer: We appreciate this comment from the reviewer. This is an important question. At present, the study of the non-transcriptional function of transcription factors is an important field. It is well known that nuclear factor-kappaB (NF- κ B) regulates neuronal structures and functions transcriptionally^{4, 5}. Our recent study showed that NF- κ B interacts with Nav1.7 in DRG neurons to regulate neuropathic pain in a non-transcriptional manner⁶. Although ATF4 is a transcription factor, it also has functions other than transcription. Studies have shown that ATF4 directly interacts with GABA_B receptors in the soma and at the dendritic membrane surface of both cultured hippocampal neurons and retinal amacrine cells^{7, 8 9}. Research has also shown that knocking down ATF4 significantly decreases the membrane expression of GABA_B receptors but does not change the total abundance of GABA_B receptors in hippocampal neurons, indicating that ATF4 regulates GABA_B receptor trafficking¹⁰. A study showed that ATF4 can be locally synthesized in axons and regulate the occurrence and development of Alzheimer's disease¹¹. Therefore, despite being a transcription factor, ATF4 may also be involved in non-transcriptional regulation. We discussed this information in the revised manuscript (Lines 356-370). To further determine the distribution of ATF4 in sensory neurons, we performed immunostaining experiments with three ATF4 antibodies from different companies (Abcam, catalogue no.: ab23760; Cell Signaling Technology, catalogue no.: 11815S and GeneTex, catalogue no.: GTX89973). The data showed that ATF4 was expressed in the cytoplasm of all stained neurons

(Supplementary Fig. 1b). ATF4 was distributed in the nuclei of most stained DRG neurons (Supplementary Fig. 1b, white arrow), but a small number of neurons did not express ATF4 in the nucleus (Supplementary Fig. 1b, yellow arrow). To further investigate the expression of ATF4 in the cytoplasm and nuclei, cytoplasmic and nucleic separation tests were carried out. Consistent with a previous study¹², ATF4 was abundantly expressed in the cytoplasm, and a small amount was expressed in the nuclei of DRG neurons (Supplementary Fig. 1c). A research report also showed that ATF4 was mainly distributed in the cytoplasm but barely distributed in the nuclei of retinal amacrine neurons⁹. These data suggested that ATF4, as a transcription factor, is weakly expressed in the nuclei of neurons under normal conditions. Many factors can activate ATF4 and promote its translocation to the nucleus and participate in the regulation of various biological functions. Activation of the TLR4-MyD88 signalling pathway can promote the translocation of ATF4 into the nucleus, thus regulating the secretion of various cytokines and participating in the innate immune response¹³. Activation of the GABA_B receptor leads to a marked translocation and accumulation of ATF4 from the cytoplasm into the nucleus in primary rat cortical cultures¹⁴. NMDA-induced long-term depression (LTD) also increases the translocation of ATF4 into the nucleus in rodent hippocampal neurons¹⁵. However, the factors resulting in ATF4 translocation to the nucleus in DRG neurons are not clear and require further study. According to the reviewer's suggestion, we commented on and discussed this information in the revised manuscript (Lines 370-384).

2): On a similar note, it seems that the three proteins, TRPM3, KIF17 and ATF4 form a complex, as they co-immunoprecipitate in all three combinations. These co-IP data in most cases are shown by single representatives. The existence of this complex is a key point of the manuscript; thus, these data need to be

shown in a more convincing manner, with disclosing the number of replicates and providing statistics.

Answer: We appreciate this comment from the reviewer. All co-IP experiments were performed three replicates, and we added this information to the figure legends. According to the reviewer's suggestion, we provided the statistics of co-IP data in the revised Figures (Fig. 7j). To confirm the interaction between proteins, we used purified proteins to study the direct interaction between proteins (ATF4 with TRPM3 and ATF4 with KIF17). GST pull-down assay data showed that ATF4 directly interacted with TRPM3 (Fig. 5g) and KIF17 (Fig. 7h).

3): TRPM3 trafficking to the plasma membrane is a key finding supporting the model. The authors use a commercial kit to separate the plasma membrane from other fractions. I would like to see this result confirmed with an independent technique for example cell surface biotinylation. At the minimum, the authors should provide a more detailed description of this technique. For example, in Fig. 5D they show increased signal in the cytoplasm, after ATF4 siRNA treatment. Is this a soluble fraction? TRPM3 is membrane protein, if it is not trafficked to the plasma membrane it is not supposed to be in the cytoplasm, rather in intracellular membranes.

Answer: We appreciate this important comment from the reviewer. According to the reviewer's suggestion, we performed cell surface biotinylation experiments, and the data showed that the surface expression of TRPM3 was decreased in DRG neurons of ATF4 heterozygous mice (Fig. 5d). To improve the repeatability of the experiment, we added a detailed description of the technique used to separate the plasma membrane from other fractions (Lines 583-600). Considering that TRPM3 is a membrane protein, if it is not trafficked to the plasma membrane and is mainly present in the intracellular membrane. To study the regulation of ATF4 on the process of TRPM3 membrane trafficking more comprehensively, intracellular membrane proteins should also be

considered. Therefore, in our study, cytoplasmic proteins were composed of two parts: a cytosolic fraction protein (soluble) and a cytoplasmic intracellular membrane fraction protein. We have described this information in detail in the revised manuscript (Lines 594-595).

4): Both the co-IP and the trafficking experiments rely on a single TRPM3 antibody. Antibodies against ion channels are generally not very reliable. While some validation of the antibody is shown in Supplementary figure 5 this part needs to be strengthened. The exact source, including catalogue number, of the antibody needs to be disclosed and references provided for its successful usage. I am somewhat concerned by the almost completely cytoplasmic localization of TRPM3 in the SIM pictures in Figures 5 and 8, which raises some concerns about this TRPM3 antibody. I would like to see additional validation of this antibody and ideally verification of the key results with a different TRPM3 antibody.

Answer: We appreciate this important comment from the reviewer. Two TRPM3 antibodies from different companies (Bioss, rabbit, catalogue no.: bs-9046R; Alomone labs, rabbit, catalogue no.: ACC-050) were used in our study. The TRPM3 antibody from Alomone labs was successfully used in a previous study¹⁶. We added the sources and catalogue numbers of the antibodies in the revised manuscript (Lines 630-632; 677-678). To further strengthen the validation of the TRPM3 antibodies, we used *Trpm3*^{-/-} mice to verify the specificity of the TRPM3 antibodies. The specificity of the TRPM3 antibodies was validated by loss of TRPM3 immunostaining in DRG neurons of *Trpm3*^{-/-} mice (Supplementary Fig. 5l, m). Although ion channels function on the cell membrane, they are mainly expressed in the cytoplasm before membrane trafficking. Because immunostaining of ion channels is extensive in the cytoplasm, distinguishing the membrane and cytoplasm by immunostaining is difficult^{16, 17, 18}. This may give the impression that ion channels seem to be

expressed almost entirely in the cytoplasm. The key results of TRPM3 trafficking (Fig. 5a, d in Alomone, ACC-050; Fig. 5i and Supplementary Fig. 5a in Bioss, bs-9046R) and ATF4/TRPM3 interaction (Fig. 5e top in Bioss, bs-9046R; Fig. 5e bottom in Alomone, ACC-050) were verified by two antibodies. To further confirm the key results of the interaction between ATF4 and TRPM3 or ATF4 and KIF17, we used purified proteins to study the direct interaction between them. GST pull-down assay data showed that ATF4 directly interacted with TRPM3 (Fig. 5g) and KIF17 (Fig. 7h).

5): Abstract lines 9-10: Loss of ATF4 in mouse dorsal root ganglion (DRG) neurons selectively impaired heat hyperalgesia. I think this is an overstatement, and these data need to be presented more carefully. In Fig 2 panels K and M, ATF4 knockdown increased latency to withdrawal from heat both in baseline conditions and after SNL or CFA injection to a similar extent. Looking at the same data from a different angle, one can easily say that ATF4 knockdown had no effect on hyperalgesia, as both in the presence and absence of it SNL and CFA induced a large increase in heat sensitivity, ATF4 knockdown just shifted the baseline.

Answer: Thank you for your comment. This is an important question. To further explore whether the effects of knocking down ATF4 on heat sensitivity in pain models were achieved by changing the baseline, we investigated the rate change in threshold with time of scramble and ATF4-siRNA mice after SNL or CFA. The data showed that the rate change in threshold was not different between scramble and ATF4-siRNA mice at different time points after SNL (Response Fig. 1a) or CFA (Response Fig. 1b). This result suggested that the effects of knocking down ATF4 on the pain models (SNL and CFA) were not due to alterations in heat hyperalgesia but rather to its effects on the basal heat threshold. According to the reviewer's suggestion, we corrected the relevant

descriptions (Lines 9-10) and discussed them in the revised manuscript (Lines 334-338).

Response Fig. 1

6): The involvement of TRPM3 in the effect of ATF4 on heat sensitivity is shown very convincingly by the CIM and PS local injection experiments. The role of TRPV1 is excluded by a single experiment, the lack of increased PM expression of TRPV1 in Fig 5a. Antibodies against ion channels are generally not very reliable. The authors may consider testing the effect of ATF4 knockdown on nocifensive responses elicited by local capsaicin injection, which would greatly increase the confidence in the lack of involvement of TRPV1.

Answer: We appreciate this important comment from the reviewer. According to the reviewer's suggestion, we observed the effect of ATF4 knockdown on spontaneous pain (duration or number of licking or flinching behaviours in 2 min) induced by intraplantar injection of capsaicin (1 nmol/paw). The behavioural data showed that knockdown of ATF4 did not change the spontaneous pain elicited by local capsaicin injection (Supplementary Fig. 5g). This result indicated that TRPV1 may not be involved in the regulation of ATF4 on heat nociception.

7): The catalogue numbers of all antibodies need to be disclosed

Answer: Thank you for your comment. We added the catalogue numbers of all antibodies in the revised manuscript (Lines 630-643; 675-683) and reporting summary.

8): Figure 5 panel p,r and t: I would make the Y axis the same on all three panels that would make the inhibition much easier to notice. Also, TRPM3 is an outwardly rectifying channel, but the currents in panel p and r show almost no rectification. Also, it is not clear what is plotted at panel o, is it the current induced by CIM, i.e. current before CIM subtracted, or the total current during CIM treatment, given the large baseline in the representatives, this is a big difference, therefore it needs to be clarified.

Answer: We appreciate this important comment from the reviewer. According to the reviewer's suggestion, we made the Y axis the same in all three panels (Fig. 5l, n, p). As the reviewer said, TRPM3 is an outwardly rectifying channel. To make our results more clearly, we updated the representative pictures (Fig. 5l, n, p) in the revised manuscript. The TRPM3 current we analysed was the CIM0216-induced current, i.e. current before CIM0216 was subtracted; we clarified this information in the figure legend (Line 1025).

9): The order of panels should be uniform, in Fig 1 the letters go vertically, most other figures it is horizontal, but in Figure 5 it is mixed

Answer: We appreciate this comment from the reviewer. According to the reviewer's suggestion, we unified the panel orders in all the figures.

10): Figure 8a-c: I would label the scale bars in the figure itself, given that a and b are low magnification pictures of whole DRG, while c is a single neuron. At the first look, the figure is confusing and gives the impression that panel C is also whole DRG given the same round shape.

Answer: We appreciate this comment from the reviewer. According to the reviewer's suggestion, we labelled the scale bars in Fig. 8a-c.

11): Given that CXCL12 is likely one of the physiological mechanisms turning on the pathway proposed by the authors, they should briefly discuss what CXCL12 is.

Answer: Thank you for this important comment. Chemokine C-X-C motif ligand 12 (CXCL12), belonging to the C-X-C chemokine subfamily, plays a critical role in regulating neural activity. For example, in the dorsal raphe nucleus or nigrostriatum, CXCL12 increases the excitability of dopaminergic (DAergic) neurons and serotonergic (5-HTergic) neurons^{19, 20}. Our studies and those of our peers showed that CXCL12 was markedly increased in the DRG after neuropathic pain was induced by nerve injury or a chemotherapeutic agent, and inhibiting CXCR4, a CXCL12 receptor, attenuated abnormal pain behaviours^{21, 22, 23}. In addition, another study suggested that CXCL12 synthesized in the somata of DRG neurons is transported to the dorsal horn to modulate nociceptive signalling²⁴. All of these studies indicate that CXCL12 participates in neuropathic pain. Our results showed that CXCL12 application increased the presentation of ATF4 in DRG neurons in vivo and in vitro. Interestingly, knocking down ATF4 in the DRG attenuated the heat hyperalgesia induced by intrathecally delivered CXCL12. CXCR4 and CXCR7 are specific CXCL12 receptors, and they participate in CXCL12-regulated physiological and pathological processes²⁵. We found that knocking down CXCR4, but not CXCR7, could inhibit the increase in ATF4 induced by CXCL12 in the DRG. Taken together, these data suggested that CXCL12-dependent ATF4 upregulation occurs via CXCR4 activation. According to the reviewer's suggestion, we added a discussion about CXCL12 in the revised manuscript (Lines 434-450).

12): ATF4 should be defined in the abstract and it should be stated that it is a transcription factor.

Answer: We appreciate this important comment from the reviewer. According to the reviewer's suggestion, we defined the transcription factor ATF4 in the abstract of the revised manuscript (Lines 6-7).

13): There are numerous typos and awkward, or grammatically incorrect sentences, a careful editing of the manuscript for language is highly recommended. Some examples are below:

Page 5 line 98 and 106 knockdown of ATF4

Page 7 line 131: we further intrathecally injected - delete "further"

Page 7 line 144: the following sentence is incomplete: Strikingly, intrathecal injection of adeno associated virus (rAAV-CMV-Atf4-2A-EGFP-WPRE-PA) to re-expression of ATF4 in DRG of ATF4+/- mice (Supplementary Fig. 4c).

Page 7 line 147: prominently rescued heat sensitivity (Fig. 3f-h) but not mechanical or cold sensitivity – ATF4 knockdown did not alter cold and mechanosensitivity, therefore there was nothing to rescue. I would change the end of the sentence to : but did not increase mechanical or cold sensitivity

Page 7: The rescue data described in Figure 3 (lines 147 - 151) should be described before the overexpression data which is presented in Figure 4.

Page 8: line 160: scrambled siRNA

Page 8 line 166, we performed high resolution imaging (instead of images)

Page 8 line 171: with the TRPM3-Flag construct

Page 10, line 212 injected into the hindpaw

Page 11 line 239 KIF17-Flag construct

Page 13 line 271: To investigate the underlying mechanism of ATF4 up-regulation

Page 16 line 313: "Considering that the heat stimulation..." This sentence is both complicated and grammatically incorrect.

Answer: We appreciate this comment from the reviewer. This is a very important and urgent issue that needs to be addressed. According to the reviewer's suggestion, we carefully corrected the spelling and grammatical errors throughout the manuscript.

Reviewer #3

1): As noted above, while the findings are consistent with a role for ATF4 as a local adapter, in my view these are mainly correlative and do not establish causality. What is absent is a clear experiment to distinguish between the well-studied transcriptional actions of ATF4 and the non-transcriptional role postulated in this ms. For instance, does over-expression of a transcriptionally inactive form of ATF4 reverse the effects of ATF4 knockdown or heterogeneity? While this point is not a demand for such experiments in this ms, it does urge the authors to more closely consider their interpretation of the data.

Answer: We appreciate this important comment from the reviewer. This is an interesting question. To further study the effect of the transcriptional function of ATF4, a transcription factor, on TRPM3 channels, we performed cytoplasmic

and nuclear separation experiments. As ISRIB can significantly inhibit the expression of ATF4²⁶, we intrathecally injected ISRIB (300 ng)²⁷ and measured ATF4 levels in the cytoplasm and nuclei of DRG neurons at different time points. A significant decrease in ATF4 was detected in the cytoplasm 30 min after injection (Supplementary Fig. 6a), whereas ATF4 expression was decreased in the nuclei 90 min after injection but not 30 min (Supplementary Fig. 6b). That is, 30 min after ISRIB was intrathecally administered, it may have inhibited the function of cytoplasmic ATF4 without affecting the nuclear transcription function of ATF4. Furthermore, 30 min after intrathecal injection of ISRIB reduced the expression of TRPM3 on the membrane (Supplementary Fig. 6c) but did not change the total expression of TRPM3 in sensory neurons (Supplementary Fig. 6d). And 30 min after intrathecal injection of ISRIB also significantly decreased the heat sensitivity of mice (Supplementary Fig. 6e). To further investigate whether ATF4 has a direct transcriptional regulatory effect on the *Trpm3* gene, we performed a luciferase reporter assay to determine the effects of ATF4 on *Trpm3* promoter activity. The data showed that knockdown of ATF4 did not alter the luciferase activity of the *Trpm3* promoter (Supplementary Fig. 6f). These results suggest that ATF4 may have no transcriptional regulatory effect on the *Trpm3* gene. As a transcription factor, ATF4 can regulate the expression of many genes. Although our data show that ATF4 has no transcriptional effect on the *Trpm3* gene, it may participate in the regulation of heat sensitivity by mediating the transcription of other genes. Therefore, we cannot exclude the effect of ATF4 transcriptional function on heat sensitivity. We discussed it in the revised manuscript (Lines 353-356).

2): In a related point, the ms reports that thermal stimulation rapidly lowers the thermal withdrawal threshold of mice, but not with ATF4 knockdown. The authors interpret this finding to a changed ATF4-TRPM3 interaction. An

alternative explanation however is that prolonged ATF4 knockdown affects transcription of key proteins involved in this response.

Answer: We appreciate this comment from the reviewer. This is an important question. Our data showed that, 30 min after ISRIB was intrathecally administered, it may have inhibited the function of cytoplasmic ATF4 without affecting the nuclear transcription function of ATF4 (Supplementary Fig. 6a, b). Behavioural data showed that intrathecal injection of ISRIB for 30 min abolished the decrease of threshold in mice after paws heat stimulation in 2 min (Supplementary Fig. 7f), showed that only inhibiting the function of ATF4 in cytoplasm but not in nuclei can also block the decrease of threshold in mice after paws heat stimulation. This result may suggest that the effect of interfering with ATF4 on the thermal stimulation threshold is mainly derived from the non-transcriptional function of ATF4. Although our evidence further indicated that the effect of interfering with ATF4 on the thermal stimulation threshold is mainly derived from the non-transcriptional function of ATF4, we cannot completely rule out the possibility that ATF4 may participate in this process through transcriptional regulation of other key proteins. We discussed this issue in the revised manuscript (Lines 400-410).

3): In light of such points, rather than referring to the role of ATF4 in membrane trafficking of TRPM3 as "ATF4-mediated" the authors might wish to refer to this role as "ATF4-dependent "

Answer: Thank you for your comment. According to the reviewer's suggestion, we changed "ATF4-mediated" to "ATF4-dependent" in our revised manuscript.

4): The authors cite Costa-Mattioli et al and Chen et al as showing that "ATF4 plays a critical role in synaptic plasticity and memory formation". However, close reading of the papers indicates that neither makes a direct case for such a role of ATF4.

Answer: We appreciate this important comment. According to the reviewer's suggestion, we updated the citation in our revised manuscript (Line 31).

5): The authors state (lines 68-69): "suggesting that ATF4 synthesized in somata of DRG neurons is transported to their central and peripheral terminals." However, the literature suggests that ATF4 can be locally synthesized in distal portions of neurons (at least in the CNS) via enhanced translation.

Answer: We appreciate this important comment from the reviewer. To make our description more accurate and reasonable, we deleted the description "suggesting that ATF4 synthesized in somata of DRG neurons is transported to their central and peripheral terminals" in the revised manuscript.

6): The sequences of the various siRNAs used in the work need to be specified.

Answer: Thank you for your comment. According to the reviewer's suggestion, we added the sequences of siRNAs used in our work to the revised manuscript (Lines 702-706).

7): The authors use "blocking peptides" to authenticate their immunostaining. However, the sequences and sources of these need to be given. Additionally, while helpful, such peptides do not necessarily rule out cross-reactivity for immunostaining.

Answer: Thank you for this comment. The sequence of the ATF4 blocking peptide (GeneTex, catalogue: GTX89973-PEP), which is a synthetic peptide derived from humans, is EEVRKARGKKRVP. The sequence of the TRPM3 blocking peptide (Bioss, catalogue: bs-9046P), which is also a synthetic peptide derived from humans, is KLFITDDELKKVH. We added the sequences and sources of the blocking peptides used in our work to the revised manuscript (Lines 694-698). Considering that the blocking peptides do not necessarily exclude the cross-reactivity of immunostaining, we used knockout animals to

verify the specificity of ATF4 (Fig. 1a) and TRPM3 (Supplementary Fig. 5I) antibodies.

8): Suppl Fig 1D. The legend describes "infection", but the text indicates siRNA treatment. The sequence of "FITC-siRNA" not appear to be described in the ms.

Answer: We appreciate this comment from the reviewer. According to the reviewer's suggestion, we changed the description in the legend (Line 1115) and added the sequence of FITC-labelled ATF4-siRNA in the revised manuscript (Lines 702-703).

9): For ATF4 siRNA treatment, what was the time that blotting, staining and experiments were performed after injection? For experiments such as shown in Fig 2, etc, was the time after treatment comparable to that for the blot and staining? That is, is the knockdown present at the time of experimentation?

Answer: We appreciate this comment from the reviewer. Our data showed that ATF4 expression decreased significantly 48 hours after siRNA injection (Supplementary Fig. 1g, h). All subsequent experiments were performed 48 hours after the siRNA injection (Line 711) so that knockdown was present in the western blotting, staining and behavioural tests.

10): It would have been useful if ATF4 het mice had been used as well as ATF4 knockdown to study TRPM3 localization from surface to cytoplasm.

Answer: Thank you for this comment. According to the reviewer's suggestion, we performed the experiment with ATF4 heterozygous mice. Similar to ATF4 knockdown, a decrease in the level of membrane TRPM3 (Fig. 5a) and an increase in the level of cytoplasm TRPM3 (Fig. 5b) but no change in total TRPM3 levels (Fig. 5c) were observed in DRG tissues from *Atf4*^{+/-} mice compared to those from WT mice. We added these results to the revised manuscript (Lines 172-175).

11): While well-written, the ms needs to be edited for minor errors in English language that sometimes affect clarity.

Answer: We appreciate this comment from the reviewer. According to the reviewer's suggestion, we carefully corrected the spelling and grammatical errors throughout the manuscript.

1. Bai L, *et al.* Upregulation of Chemokine CXCL12 in the Dorsal Root Ganglia and Spinal Cord Contributes to the Development and Maintenance of Neuropathic Pain Following Spared Nerve Injury in Rats. *Neuroscience bulletin* **32**, 27-40 (2016).
2. Daniel SK, Seo YD, Pillarisetty VG. The CXCL12-CXCR4/CXCR7 axis as a mechanism of immune resistance in gastrointestinal malignancies. *Seminars in cancer biology*, (2019).
3. Xu T, *et al.* Epigenetic upregulation of CXCL12 expression mediates antitubulin chemotherapeutics-induced neuropathic pain. *Pain* **158**, 637-648 (2017).
4. Srinivasan M, Lahiri DK. Significance of NF-kappaB as a pivotal therapeutic target in the neurodegenerative pathologies of Alzheimer's disease and multiple sclerosis. *Expert opinion on therapeutic targets* **19**, 471-487 (2015).
5. Niederberger E, Geisslinger G. The IKK-NF-kappaB pathway: a source for novel molecular drug targets in pain therapy? *FASEB journal : official publication of the Federation of American Societies for Experimental Biology* **22**, 3432-3442 (2008).
6. Xie MX, *et al.* Nuclear Factor-kappaB Gates Nav1.7 Channels in DRG Neurons via Protein-Protein Interaction. *iScience* **19**, 623-633 (2019).
7. Ritter B, Zschuntsch J, Kvachnina E, Zhang W, Ponimaskin EG. The GABA(B) receptor subunits R1 and R2 interact differentially with the activation transcription factor ATF4 in mouse brain during the postnatal development. *Brain research Developmental brain research* **149**, 73-77 (2004).
8. Vernon E, *et al.* GABA(B) receptors couple directly to the transcription factor ATF4. *Molecular and cellular neurosciences* **17**, 637-645 (2001).
9. Nehring RB, *et al.* The metabotropic GABAB receptor directly interacts with the activating transcription factor 4. *The Journal of biological chemistry* **275**, 35185-35191 (2000).
10. Corona C, Pasini S, Liu J, Amar F, Greene LA, Shelanski ML. Activating Transcription Factor 4 (ATF4) Regulates Neuronal Activity by Controlling GABABR Trafficking. *J Neurosci* **38**, 6102-6113 (2018).
11. Baleriola J, *et al.* Axonally synthesized ATF4 transmits a neurodegenerative signal across brain regions. *Cell* **158**, 1159-1172 (2014).
12. Dong L, Guarino BB, Jordan-Sciutto KL, Winkelstein BA. Activating transcription factor 4, a mediator of the integrated stress response, is increased in the dorsal root ganglia following painful facet joint distraction. *Neuroscience* **193**, 377-386 (2011).
13. Zhang C, *et al.* ATF4 is directly recruited by TLR4 signaling and positively regulates TLR4-

- triggered cytokine production in human monocytes. *Cellular & molecular immunology* **10**, 84-94 (2013).
14. White JH, *et al.* The GABAB receptor interacts directly with the related transcription factors CREB2 and ATFx. *Proc Natl Acad Sci U S A* **97**, 13967-13972 (2000).
 15. Lai KO, Zhao Y, Ch'ng TH, Martin KC. Importin-mediated retrograde transport of CREB2 from distal processes to the nucleus in neurons. *Proceedings of the National Academy of Sciences* **105**, 17175-17180 (2008).
 16. Suzuki A, Shinoda M, Honda K, Shirakawa T, Iwata K. Regulation of transient receptor potential vanilloid 1 expression in trigeminal ganglion neurons via methyl-CpG binding protein 2 signaling contributes tongue heat sensitivity and inflammatory hyperalgesia in mice. *Molecular pain* **12**, (2016).
 17. Han Q, *et al.* SHANK3 Deficiency Impairs Heat Hyperalgesia and TRPV1 Signaling in Primary Sensory Neurons. *Neuron* **92**, 1279-1293 (2016).
 18. He XH, *et al.* TNF-alpha contributes to up-regulation of Nav1.3 and Nav1.8 in DRG neurons following motor fiber injury. *Pain* **151**, 266-279 (2010).
 19. Heinisch S, Kirby LG. SDF-1 alpha/CXCL12 enhances GABA and glutamate synaptic activity at serotonin neurons in the rat dorsal raphe nucleus. *Neuropharmacology* **58**, 501-514 (2010).
 20. Skrzydelski D, *et al.* The chemokine stromal cell-derived factor-1/CXCL12 activates the nigrostriatal dopamine system. *J Neurochem* **102**, 1175-1183 (2007).
 21. Dubovy P, Klusakova I, Svizenska I, Brazda V. Spatio-temporal changes of SDF1 and its CXCR4 receptor in the dorsal root ganglia following unilateral sciatic nerve injury as a model of neuropathic pain. *Histochem Cell Biol* **133**, 323-337 (2010).
 22. Bai LY, *et al.* Upregulation of Chemokine CXCL12 in the Dorsal Root Ganglia and Spinal Cord Contributes to the Development and Maintenance of Neuropathic Pain Following Spared Nerve Injury in Rats. *Neurosci Bull* **32**, 27-40 (2016).
 23. Xu T, *et al.* Epigenetic upregulation of CXCL12 expression mediates antitubulin chemotherapeutics-induced neuropathic pain. *Pain*, (2016).
 24. Reaux-Le Goazigo A, Rivat C, Kitabgi P, Pohl M, Melik Parsadaniantz S. Cellular and subcellular localization of CXCL12 and CXCR4 in rat nociceptive structures: physiological relevance. *The European journal of neuroscience* **36**, 2619-2631 (2012).
 25. Li M, Hale JS, Rich JN, Ransohoff RM, Lathia JD. Chemokine CXCL12 in neurodegenerative

diseases: an SOS signal for stem cell-based repair. *Trends in neurosciences* **35**, 619-628 (2012).

26. Pathak SS, *et al.* The eIF2alpha Kinase GCN2 Modulates Period and Rhythmicity of the Circadian Clock by Translational Control of Atf4. *Neuron*, (2019).
27. Barragan-Iglesias P, *et al.* Activation of the integrated stress response in nociceptors drives methylglyoxal-induced pain. *Pain* **160**, 160-171 (2019).

Reviewers' Comments:

Reviewer #1:

Remarks to the Author:

The newly generated data substantiate the author's model that heat hyperalgesia is mediated by a mechanism that involves the transcription factor ATF4 recruiting the motor protein KIF17 to redistribute TRPM3 to the plasma membrane. The readability of the manuscript was improved but could still benefit from editing for grammar. The added description of the colocalization method in the materials and methods section is sufficient, but I was not able to find the specified software "IPWIN60" on the internet. Please double check the name of the software and include a link to the company or platform the software is distributed by. In line 54 and following, several cargos of KIF17 are described. Here the most well characterized cargo of KIF17, the NR2B subunit of the ion channel N-methyl-D-aspartate receptor should be mentioned. The paper in which this cargo was discovered, Setou et al., 2000, is already cited here. I apologize for overlooking this in the first round of review. The added model panel (Fig. 9e) illustrates the authors' model well. Overall, the revised manuscript is significantly improved.

Reviewer #2:

Remarks to the Author:

The authors provided a thorough and reasonable response to the critiques, I have no further comments

Reviewer #3:

Remarks to the Author:

The authors have made an earnest attempt to respond to the points raised by this and the other reviewers and although not all is resolved, in my view the work is novel and of overall interest and of sufficient quality to be published.

The main issue I have that is still unresolved is whether or not ATF4 acts in a transcriptional role or not to regulate TRPM trafficking. The main response of the authors has been to add a few non-definitive experiments and to add a lot of hand-waving disclaimers to the discussion. The main new data involve the use of ISRIB to rapidly block ATF4 translation. The authors report that 30 min of ISRIB treatment is sufficient to reduce cytoplasmic (but not nuclear) ATF4 levels as well as to reduce membrane TRPM3 levels. On this basis, they conclude that ATF4 is acting non-transcriptionally. While this is a nice experiment, it must be considered that ISRIB's effects are not specific to ATF4 and will affect translation of additional proteins that may in turn regulate TRPM membrane association by other mechanisms. As originally suggested, a better experiment would have been to knockdown ATF4 and replace it with a transcriptionally inactive form to determine whether this phenocopies the experiments in which the authors did this with WT ATF4. The bottom line is that the issue still remains (which is not a fatal flaw regarding publishing) and the paper would read better if the authors had a more cogent and focused discussion of this rather than the diffuse, hand-waving that it presently possesses.

A few additional points.

Line 118: The authors state that ATF4 KO is lethal. Although ATF4 null mice suffer various defects, they can survive.

Line 368: The authors state "Therefore, despite being a transcription factor, ATF4 may also be involved in non-transcriptional regulation." However, the "therefore" is based on studies that the authors cite that show transcriptional as well as non-transcriptional actions of ATF4.

REVIEWER COMMENTS

Reviewer #1 (Remarks to the Author):

The newly generated data substantiate the author's model that heat hyperalgesia is mediated by a mechanism that involves the transcription factor ATF4 recruiting the motor protein KIF17 to redistribute TRPM3 to the plasma membrane. The readability of the manuscript was improved but could still benefit from editing for grammar. The added description of the colocalization method in the materials and methods section is sufficient, but I was not able to find the specified software "IPWIN60" on the internet. Please double check the name of the software and include a link to the company or platform the software is distributed by. In line 54 and following, several cargos of KIF17 are described. Here the most well characterized cargo of KIF17, the NR2B subunit of the ion channel N-methyl-D-aspartate receptor should be mentioned. The paper in which this cargo was discovered, Setou et al., 2000, is already cited here. I apologize for overlooking this in the first round of review. The added model panel (Fig. 9e) illustrates the authors' model well. Overall, the revised manuscript is significantly improved.

Reviewer #2 (Remarks to the Author):

The authors provided a thorough and reasonable response to the critiques, I have no further comments

Reviewer #3 (Remarks to the Author):

The authors have made an earnest attempt to respond to the points raised by this and the other reviewers and although not all is resolved, in my view the work is novel and of overall interest and of sufficient quality to be published.

The main issue I have that is still unresolved is whether or not ATF4 acts in a transcriptional role or not to regulate TRPM trafficking. The main response of the authors has been to add a few non-definitive experiments and to add a lot of hand-waving disclaimers to the discussion. The main new data involve the use of ISRIB to rapidly block ATF4 translation. The authors report that 30 min of ISRIB treatment is sufficient to reduce cytoplasmic (but not nuclear) ATF4 levels as well as to reduce membrane TRPM3 levels. On this basis, they conclude that ATF4 is acting non-transcriptionally. While this is a nice experiment, it must be considered that ISRIB's effects are not specific to ATF4 and will affect translation of additional proteins that may in turn regulate TRPM membrane association by other mechanisms. As originally suggested, a better experiment would have been to knockdown ATF4 and replace it with a transcriptionally inactive form to determine whether this phenocopies the experiments in which the authors did this with WT ATF4. The bottom line is that the issue still remains (which is not a fatal flaw regarding publishing) and the paper would read better if the authors had a more cogent and focused discussion of this rather than the diffuse, hand-waving that it presently possesses.

A few additional points.

Line 118: The authors state that ATF4 KO is lethal. Although ATF4 null mice suffer various defects, they can survive.

Line 368: The authors state "Therefore, despite being a transcription factor,

ATF4 may also be involved in non-transcriptional regulation.” However, the “therefore” is based on studies that the authors cite that show transcriptional as well as non-transcriptional actions of ATF4.

Reviewer #1:

1): The newly generated data substantiate the author's model that heat hyperalgesia is mediated by a mechanism that involves the transcription factor ATF4 recruiting the motor protein KIF17 to redistribute TRPM3 to the plasma membrane. The readability of the manuscript was improved but could still benefit from editing for grammar. The added description of the colocalization method in the materials and methods section is sufficient, but I was not able to find the specified software "IPWIN60" on the internet. Please double check the name of the software and include a link to the company or platform the software is distributed by. In line 54 and following, several cargos of KIF17 are described. Here the most well characterized cargo of KIF17, the NR2B subunit of the ion channel N-methyl-D-aspartate receptor should be mentioned. The paper in which this cargo was discovered, Setou et al., 2000, is already cited here. I apologize for overlooking this in the first round of review. The added model panel (Fig. 9e) illustrates the authors' model well. Overall, the revised manuscript is significantly improved.

Answer: We appreciate the comments from the reviewer. In order to further improve the readability of the manuscript, we read the whole manuscript carefully and edited the grammar.

The software "IPWIN60" means Image-Pro Plus Version 6.0 for Windows (<https://www.mediacy.com/imageproplus>), we corrected "IPWIN60" to "Image-Pro Plus 6.0" in the revised manuscript (Lines 693 and 728).

According to the reviewer's suggestion, we added the most well characterized cargo of KIF17, the NR2B subunit of N-methyl-D-aspartate receptor, to the revised manuscript (Line 56).

Reviewer #2:

The authors provided a thorough and reasonable response to the critiques, I have no further comments

Answer: We thank the reviewer for her/his great efforts to improve our manuscript.

Reviewer #3:

1): The authors have made an earnest attempt to respond to the points raised by this and the other reviewers and although not all is resolved, in my view the work is novel and of overall interest and of sufficient quality to be published.

The main issue I have that is still unresolved is whether or not ATF4 acts in a transcriptional role or not to regulate TRPM trafficking. The main response of the authors has been to add a few non-definitive experiments and to add a lot of hand-waving disclaimers to the discussion. The main new data involve the use of ISRIB to rapidly block ATF4 translation. The authors report that 30 min of ISRIB treatment is sufficient to reduce cytoplasmic (but not nuclear) ATF4 levels as well as to reduce membrane TRPM3 levels. On this basis, they conclude that ATF4 is acting non-transcriptionally. While this is a nice experiment, it must be considered that ISRIB's effects are not specific to ATF4 and will affect translation of additional proteins that may in turn regulate TRPM membrane association by other mechanisms. As originally suggested, a better experiment would have been to knockdown ATF4 and replace it with a transcriptionally inactive form to determine whether this phenocopies the experiments in which the authors did this with WT ATF4. The bottom line is that the issue still remains (which is not a fatal flaw regarding publishing) and the paper would read better if the authors had a more cogent and focused

discussion of this rather than the diffuse, hand-waving that it presently possesses.

Answer: We appreciate this important comment from the reviewer. To further explore whether ATF4 regulates TRPM3 trafficking in a non-transcriptional manner, we overexpressed a transcriptionally inactive form of ATF4 (rAAV-CMV-*Atf4*- Δ bZIP-2A-EGFP-WPRE-PA) in *Atf4*^{+/-} mice and the membrane expression of TRPM3 in sensory neurons were tested. The transcriptionally inactive form of ATF4 (rAAV-CMV-*Atf4*- Δ bZIP-2A-EGFP-WPRE-PA) was generated by site-directed mutagenesis with 6 amino acid substitutions within the DNA-binding domain (²⁹²R²⁹³YRQ²⁹⁴K²⁹⁵K²⁹⁶R²⁹⁷ to ²⁹²G²⁹³Y²⁹⁴L²⁹⁵E²⁹⁶A²⁹⁷A²⁹⁸)^{1, 2, 3}. The data showed that overexpression of a transcriptionally inactive form of ATF4 in *Atf4*^{+/-} mice increased the membrane expression of TRPM3 but did not alter the total expression of TRPM3 in sensory neurons (Supplementary Fig. 6f, g). And overexpressing the transcriptionally inactive form of ATF4 also significantly enhanced the heat sensitivity of *Atf4* heterozygote mice (Supplementary Fig. 6h). These results suggest that ATF4 regulates TRPM3 trafficking in a non-transcriptional manner. We added these results in the revised manuscript (Supplementary Fig. 6f-h, Lines 212-222). According to the reviewer's suggestion, we deleted the diffuse and hand-waving discussion in the revised manuscript.

2): Line 118: The authors state that ATF4 KO is lethal. Although ATF4 null mice suffer various defects, they can survive.

Answer: Thanks for the comment from the reviewer. In our and peer studies, *Atf4* homozygous pups generally died within the first hour after birth, and even the surviving homozygous mice showed various functional defects, such as abnormal lens development and decreased fertility⁴. According to the reviewer's suggestion, we corrected the relevant description in the revised manuscript (Line 118).

3): Line 368: The authors state “Therefore, despite being a transcription factor, ATF4 may also be involved in non-transcriptional regulation.” However, the “therefore” is based on studies that the authors cite that show transcriptional as well as non-transcriptional actions of ATF4.

Answer: We appreciate the comment from the reviewer. According to the reviewer’s suggestion, we deleted the citation of ATF4 transcription function in the revised manuscript (Lines 369-377).

1. He CH, *et al.* Identification of activating transcription factor 4 (ATF4) as an Nrf2-interacting protein. Implication for heme oxygenase-1 gene regulation. *The Journal of biological chemistry* **276**, 20858-20865 (2001).
2. Siu F, Bain PJ, LeBlanc-Chaffin R, Chen H, Kilberg MS. ATF4 is a mediator of the nutrient-sensing response pathway that activates the human asparagine synthetase gene. *The Journal of biological chemistry* **277**, 24120-24127 (2002).
3. Ebert SM, *et al.* The transcription factor ATF4 promotes skeletal myofiber atrophy during fasting. *Molecular endocrinology* **24**, 790-799 (2010).
4. Masuoka HC, Townes TM. Targeted disruption of the activating transcription factor 4 gene results in severe fetal anemia in mice. *Blood* **99**, 736-745 (2002).

Reviewers' Comments:

Reviewer #3:

Remarks to the Author:

The authors are to be commended for strengthening their manuscript by carrying out the suggested, but optional, experiment with a transcriptionally inactive form of ATF4. Very nice study!

REVIEWER COMMENTS

Reviewer #3 (Remarks to the Author):

The authors are to be commended for strengthening their manuscript by carrying out the suggested, but optional, experiment with a transcriptionally inactive form of ATF4. Very nice study!

Answer: We thank the reviewer for her/his great efforts to improve our manuscript.